# BLOCKWISE ADAPTIVITY: FASTER TRAINING AND BETTER GENERALIZATION IN DEEP LEARNING

## ABSTRACT

Stochastic methods with coordinate-wise adaptive stepsize (such as RMSprop and Adam) have been widely used in training deep neural networks. Despite their fast convergence, they can generalize worse than stochastic gradient descent. In this paper, by revisiting the design of Adagrad, we propose to split the network parameters into blocks, and use a blockwise adaptive stepsize. Intuitively, blockwise adaptivity is less aggressive than adaptivity to individual coordinates, and can have a better balance between adaptivity and generalization. We show theoretically that the proposed blockwise adaptive gradient descent has comparable regret in online convex learning and convergence rate for optimizing nonconvex objective as its counterpart with coordinate-wise adaptive stepsize, but is better up to some constant. We also study its uniform stability and show that blockwise adaptivity can lead to lower generalization error than coordinate-wise adaptivity. Experimental results show that blockwise adaptive gradient descent converges faster and improves generalization performance over Nesterov's accelerated gradient and Adam.

## 1 INTRODUCTION

Deep networks have achieved excellent performance in a variety of domains such as computer vision (He et al., 2016), language modeling (Zaremba et al., 2014), and speech recognition (Graves et al., 2013). The most popular optimizer is stochastic gradient decent (SGD) (Robbins & Monro, 1951), which is simple and has low per-iteration complexity. Its convergence rate is also well-established (Ghadimi & Lan, 2013; Bottou et al., 2018). However, vanilla SGD is sensitive to the choice of stepsize, and requires careful tuning.

To improve the efficiency and robustness of SGD, many variants based on coordinate-wise adaptive stepsize (Almeida et al., 1999; Duchi et al., 2011; Schaul et al., 2013; Kingma & Ba, 2015; Tieleman & Hinton, 2012; Zeiler, 2012; Zheng & Kwok, 2017; Reddi et al., 2018; Zaheer et al., 2018; Zou & Shen, 2018; Chen et al., 2019; Zou et al., 2019) have been proposed. Though this is effective in accelerating convergence, its generalization performance is often worse than SGD (Wilson et al., 2017). Recently, it is shown that coordinate-wise adaptive gradient descent is closely related to sign-based gradient descent (Balles & Hennig, 2018; Bernstein et al., 2018). Theoretical arguments and empirical evidence suggest that the gradient sign impedes generalization (Balles & Hennig, 2018). To contract the generalization gap, a partial adaptive parameter for the second-order momentum is proposed (Chen & Gu, 2018). By using a smaller partial adaptive parameter, the adaptive gradient algorithm behaves less like sign descent and more like SGD. Based on the similar motivation, Luo et al. (2019) proposed to gradually transform Adam Kingma & Ba (2015) to SGD. Moreover, to avoid numerical problems in practical implementation, a small $\epsilon$ ($= 10^{-8}$) parameter is typically used in these methods. This $\epsilon$ parameter controls adaptivity of the algorithm (Zaheer et al., 2018). A larger value (say, $\epsilon = 10^{-3}$) can reduce adaptivity and empirically helps Adam to match its generalization performance with SGD. This implies that coordinate-wise adaptivity may be too aggressive for good generalization performance.

To improve generalization performance, attempts have been made to use a layer-wise stepsize (Singh et al., 2015; You et al., 2017; Yu et al., 2017; Zhou et al., 2018), which assign different stepsizes to different layers or normalize the layer-wise gradient. However, there has been no theoretical analysis for its empirical success.

In this paper, we consider the more general case in which model parameters are partitioned into blocks, instead of simply into layers. By revisiting the derivation of Adagrad, we propose the use of a blockwise stepsize which depends on the corresponding gradient block. Such blockwise adaptivity is less aggressive than coordinate-wise adaptivity, as it adapts to parameter blocks instead of to individual parameters. This can thus have a better balance between adaptivity and generalization. Moreover, unlike coordinate-wise adaptivity, blockwise adaptivity is not sign-based gradient descent, and so does not suffer from its performance deterioration.

As in (Bernstein et al., 2018; Ghadimi & Lan, 2013; Ward et al., 2018; Zaheer et al., 2018; Zou & Shen, 2018; Zou et al., 2019), we will focus on the expected risk minimization problem:

$$\min_{\theta} F(\theta) = \mathbf{E}_z[f(\theta; z)], \tag{1}$$

where $f$ is some possibly nonconvex loss function, $z$ is a random sample, $\theta$ is the model parameter, and the expectation is taken w.r.t. the underlying sample distribution. The expected risk measures the generalization performance on unseen data (Bottou et al., 2018), and reduces to the empirical risk when a finite training set is considered. We show theoretically that the proposed blockwise adaptive gradient descent can be faster than its coordinate-wise counterpart. Using tools on uniform stability (Bousquet & Elisseeff, 2002; Hardt et al., 2016), we also show that blockwise adaptivity has potentially lower generalization error than coordinate-wise adaptivity. Empirically, blockwise adaptive gradient descent converges faster and obtains better generalization performance than coordinate-wise descent (Adam) and Nesterov's accelerated gradient (NAG) (Sutskever et al., 2013).

**Notations**. For an integer $n$, $[n] = \{1, 2, \ldots, n\}$. For a vector $x$, $x^T$ denotes its transpose, $\text{Diag}(x)$ is a diagonal matrix with $x$ on its diagonal, $\sqrt{x}$ is the element-wise square root of $x$, $x^2$ is the coordinate-wise square of $x$, $\|x\|_2 = \sqrt{x^T x}$, $\|x\|_\infty = \max_i |x_i|$, $\|x\|_Q^2 = x^T Q x$, where $Q$ is a positive semidefinite (psd) matrix, and $x \geq 0$ means $x_i \geq 0$ for all $i$. For two vectors $x$ and $y$, $x/y$, and $\langle x, y \rangle$ denote the element-wise division and dot product, respectively. For a square matrix $X$, $X^{-1}$ is its inverse, and $X \succeq 0$ means that $X$ is psd. Moreover, $1_d = [1, 1, \ldots, 1]^T \in \mathbb{R}^d$.

## 2 RELATED WORK

### 2.1 ADAGRAD

Adagrad (shown in Algorithm 1) is an adaptive gradient method in online convex learning with coordinate-wise stepsize (Duchi et al., 2011; McMahan & Streeter, 2010). It is particularly useful for sparse learning, as parameters for the rare features can take large steps. Recently, Ward et al. (2018) established its convergence properties with a global adaptive stepsize in nonconvex optimization. It is shown that Adagrad converges to a stationary point at the optimal $\mathcal{O}(1/\sqrt{T})$ rate (up to a factor $\log(T)$), where $T$ is the total number of iterations.

Recall that the SGD iterate is the solution to the problem: $\theta_{t+1} = \arg\min_{\theta} \langle g_t, \theta \rangle + \frac{1}{2\eta}\|\theta - \theta_t\|_2^2$, where $g_t$ is the gradient of the loss $f_t$ at iteration $t$, and $\theta_t \in \mathbb{R}^d$ is the parameter vector. To incorporate information about the curvature of sequence $\{f_t\}$, the $\ell_2$-norm in the SGD update can be replaced by the Mahalanobis norm, leading to (Duchi et al., 2011):

$$\theta_{t+1} = \arg\min_{\theta} \langle g_t, \theta \rangle + \frac{1}{2\eta}\|\theta - \theta_t\|_{\text{Diag}(s_t)^{-1}}^2, \tag{2}$$

where $s_t \geq 0$. This is an instance of mirror descent (Nemirovski & Yudin, 1983). Its regret bound has a gradient-related term $\sum_{t=1}^{T} \|g_t\|_{\text{Diag}(s_t)^{-1}}^2$. Adagrad's stepsize can be obtained by examining a similar objective (Duchi et al., 2011):

$$\min_{s \in \mathcal{S}} \sum_{t=1}^{T} \|g_t\|_{\text{Diag}(s)^{-1}}^2, \tag{3}$$

where $\mathcal{S} = \{s : s \geq 0, \langle s, 1 \rangle \leq c\}$, and $c$ is some constant. At optimality, $s_{*,i} = c\|g_{1:T,i}\|_2 / \sum_{j=1}^{d} \|g_{1:T,j}\|_2$, where $g_{1:T,i} = [g_{1,i}, \ldots, g_{T,i}]^T$. As $s_t$ cannot depend on $g_j$'s with $j > t$, this suggests $s_{t,i} \propto \|g_{1:t,i}\|_2$. Theoretically, this choice of $s_t$ leads to a regret bound that is competitive with the best post-hoc optimal bound (McMahan & Streeter, 2010).

To solve the expected risk minimization problem in (1), an Adagrad variant called weighted AdaEMA is recently proposed in (Zou et al., 2019). It employs weighted averaging of $g_{t,i}^2$'s for stepsize. Moreover, weighted AdaEMA is a general coordinate-wise adaptive method and includes many Adagrad variants, including Adam and RMSprop, as special cases.

## 2.2 UNIFORM STABILITY

Given $n$ samples $S = \{z_i\}_{i=1}^n$ drawn i.i.d. from an underlying data distribution $\mathcal{D}$, one often learns the model by minimizing the empirical risk: $\min_\theta \Phi_S(\theta) \equiv \frac{1}{n} \sum_{i=1}^n f(\theta; z_i)$. Let $M(S)$ be the output of a possibly randomized algorithm $M$ (e.g., SGD) running on data $S$.

**Definition 1.** *(Hardt et al., 2016) Let $S$ and $S'$ be two data sets of size $n$ that differ in only one sample. Algorithm $M$ is $\epsilon_u$-uniformly stable if $\epsilon_{stab} \equiv \sup_{S,S'} \sup_{z \in \mathcal{D}} \mathbf{E}_M[f(M(S); z) - f(M(S'); z)] \leq \epsilon_u$.*

The generalization error is defined as $\epsilon_{gen} \equiv \mathbf{E}_{S,M}[\Phi_S(M(S)) - F(M(S))]$, where the expectation is taken w.r.t. the set $S$ and randomness of $M$ (Hardt et al., 2016). It is shown that the generalization error is bounded by the uniform stability of $M$, i.e., $|\epsilon_{gen}| \leq \epsilon_{stab}$ (Hardt et al., 2016). In other words, the more uniformly stable an algorithm is, the lower is its generalization error.

## 3 BLOCKWISE ADAPTIVE DESCENT

Consider the under-determined least squares problem:

$$\min_\theta \|X\theta - y\|_2^2, \tag{4}$$

where $X \in \mathbb{R}^{n \times d}$ is the input matrix (with sample size $n$, and dimensionality $d > n$), and output $y \in \mathbb{R}^n$. We assume that $XX^T$ is invertible. As pointed out in (Zhang et al., 2017; Wilson et al., 2017), any stochastic gradient descent method on problem (4) with a global stepsize outputs a trajectory with iterates lying in the span of the rows of $X$. One solution of (4) is $X^T(XX^T)^{-1}y$, which happens to be the solution with minimum $\ell_2$-norm among all possible global minimizers. This minimum-norm solution has the largest margin, and maximizing margin typically leads to lower generalization error (Boser et al., 1992).

It is known that SGD converges to the minimum $\ell_2$-norm solution of problem (4) (Zhang et al., 2017). On the other hand, coordinate-wise adaptive methods (such as Adagrad, RMSprop, and Adam) fail to find the minimum $\ell_2$-norm solution, but converge to solutions with low $\ell_\infty$-norm instead (Wilson et al., 2017). Examples in (Wilson et al., 2017) show that solutions obtained by these adaptive methods can generalize arbitrarily poorly, while the SGD solution makes no error.

In Section 3.1, we first show that blockwise adaptivity, unlike coordinate-wise adaptivity, can find the minimum $\ell_2$-norm solution of a nonlinear least squares problem in layer-wise training of a neural network. This motivates us to further exploit blockwise adaptivity in end-to-end neural network training (Section 3.2). To provide more analysis, Section 3.3 studies the proposed algorithm in the online convex learning setting as in (Duchi et al., 2011; Kingma & Ba, 2015; Reddi et al., 2018).

## 3.1 BLOCKWISE VS COORDINATE-WISE ADAPTIVITY

Consider a $L$-layer neural network with output $\phi_{L-1}(\cdots \phi_2(\phi_1(XW_1)W_2) \cdots W_{L-1})W_L$, where $\{W_l \in \mathbb{R}^{d_{l-1} \times d_l}\}_{l=1}^L$ are weight matrices with $d_0 = d$ and $d_L = m$, the output dimensionality. Assume that the nonlinear activation functions $\{\phi_l\}_{l=1}^{L-1}$ are bijective with nonzero derivatives on $\mathbb{R}$ (e.g., tanh and leaky ReLU). For simplicity, we further assume that $d_l = d = m > n$ for all $l$. Training this neural network with the square loss corresponds to solving the nonlinear optimization problem: $\min_{\{W_l\}_{l=1}^L} \|\phi_{L-1}(\cdots \phi_2(\phi_1(XW_1)W_2) \cdots W_{L-1})W_L - Y\|_2^2$, where $Y \in \mathbb{R}^{n \times m}$ is the label matrix. Consider training the network layer-by-layer, starting from the bottom one. For layer $l$, its optimization subproblem can be rewritten as

$$\min_{W_l} \|\Phi_l(H_{l-1}W_l) - Y\|_2^2, \tag{5}$$

where $\Phi_l(\cdot) = \phi_{L-1}(\cdots \phi_{l+1}(\phi_l(\cdot)W_{l+1}) \cdots W_{L-1})W_L$, $H_{l-1} = \phi_{l-1}(\cdots \phi_1(XW_1) \cdots W_{l-1})$ is the input activation to the $l$th layer, and $H_0 = X$. Note that (4) is a special case of (5) with $L = 1$

and identity mapping. To minimize (5), the weights for this layer are updated as

$$W_{t+1,l} = W_{t,l} - \eta_{t,l} g_{t,l}, \tag{6}$$

where $g_{t,l}$ is a stochastic gradient evaluated at $W_{t,l}$ at time $t$, and $\eta_{t,l}$ is the stepsize which may be adaptive in that it depends on $g_{t,l}$.

**Proposition 1.** *Assume that $W_{l'}$'s (with $l' > l$) are invertible. If $W_l$ is initialized to zero, and $H_{l-1}$ has full row rank, the critical point that (6) converges to is the minimum $\ell_2$-norm solution of (5) in expectation.*

Another benefit of using a blockwise stepsize is that the optimizer's extra memory cost can be reduced. Using a coordinate-wise stepsize requires an additional $\mathcal{O}(d)$ memory for storing estimates of the second moment, while the blockwise stepsize only needs an extra $\mathcal{O}(B)$ memory, where $B$ is the number of blocks. A deep network generally has millions of parameters but only tens of layers. If we set $B$ to be the number of layers, memory reduction can be significant.

## 3.2 BLOCKWISE ADAPTIVE GRADIENT (BAG)

Let the gradient $g_t \in \mathbb{R}^d$ be partitioned to $\{g_{t,\mathcal{G}_b} \in \mathbb{R}^{d_b} : b = 1, \ldots, B\}$, where $\mathcal{G}_b$ is the set of indices in block $b$, and $g_{t,\mathcal{G}_b}$ is the corresponding subvector of $g_t$. Inspired by problem (3) in the derivation of Adagrad, we consider the following variant which imposes a block structure on $s$:

$$\min_{s \in \mathcal{S}'} \sum_{t=1}^{T} \|g_t\|^2_{\text{Diag}(s)^{-1}}, \tag{7}$$

where $\mathcal{S}' = \{s : s = [q_1 1_{d_1}^T, \ldots, q_B 1_{d_B}^T]^T \geq 0, \langle s, 1 \rangle \leq c\}$ for some $q_i \in \mathbb{R}$. We assume the indices in $\mathcal{G}_b$ are consecutive; otherwise, we can simply reorder the elements of the gradient. Note that reordering does not change the result, as the objective is invariant to ordering of the coordinates. It can be easily shown that at optimality of (7), $q_b = c\|g_{1:T,\mathcal{G}_b}\|_2/(\sqrt{d_b} \sum_{i=1}^{B} \sqrt{d_i} \|g_{1:T,\mathcal{G}_i}\|_2)$, where $g_{1:T,\mathcal{G}_b} = [g_{1,\mathcal{G}_b}^T, \ldots, g_{T,\mathcal{G}_b}^T]^T$. The optimal $q_b$ is thus proportional to $\|g_{1:T,\mathcal{G}_b}\|_2/\sqrt{d_b}$. When $s_t$ in (2) is partitioned by the same block structure, the optimal $q_b$ suggests to incorporate $\|g_{1:t,\mathcal{G}_b}\|_2/\sqrt{d_b}$ into $s_t$ for block $b$ at time $t$.

| **Algorithm 1** Adagrad: Adaptive gradient for online convex learning (Duchi et al., 2011). | **Algorithm 2** BAG: Blockwise adaptive gradient for online convex learning. |
|---|---|
| 1: **Input:** $\eta > 0$; $\epsilon > 0$. | 1: **Input:** $\eta > 0$; $\epsilon > 0$. |
| 2: **initialize** $\theta_1$; $v_0 \leftarrow 0$ | 2: **initialize** $\theta_1$; $v_0 \leftarrow 0$ |
| 3: **for** $t = 1, 2, \ldots, T$ **do** | 3: **for** $t = 1, 2, \ldots, T$ **do** |
| 4:     Receive subgradient $g_t \in \partial f_t(\theta_t)$ | 4:     Receive subgradient $g_t \in \partial f_t(\theta_t)$ |
| 5:     **for** $i = 1, 2, \ldots, d$ **do** | 5:     **for** $b = 1, 2, \ldots, B$ **do** |
| 6:       $v_{t,i} = v_{t-1,i} + \|g_{t,i}\|^2_2$ | 6:       $v_{t,b} = v_{t-1,b} + \|g_{t,\mathcal{G}_b}\|^2_2/d_b$ |
| 7:       $\theta_{t+1,i} = \theta_{t,i} - \eta g_{t,i}/(\sqrt{v_{t,i}} + \epsilon)$ | 7:       $\theta_{t+1,\mathcal{G}_b} = \theta_{t,\mathcal{G}_b} - \eta g_{t,\mathcal{G}_b}/(\sqrt{v_{t,b}} + \epsilon)$ |
| 8:     **end for** | 8:     **end for** |
| 9: **end for** | 9: **end for** |

The proposed procedure, which will be called blockwise adaptive gradient (BAG), is shown in Algorithm 2. Compared to Adagrad, each block, instead of each coordinate, has its own learning rate. When $B = d$ (i.e., each block has only one coordinate), BAG reduces to Adagrad. When $B = 1$ (i.e., all coordinates are grouped together), Algorithm 2 produces the update: $\theta_{t+1} = \theta_t - \eta(g_t/(\|g_{1:t}\|_2/\sqrt{d} + \epsilon))$ with a global adaptive learning rate, which is equivalent to AdaGrad-Norm (Ward et al., 2018).

Returning to the underdetermined least squares problem in (4), the following Proposition shows that when $B > 1$, BAG finds the minimum $\ell_2$-norm solution in each subspace induced by the group structure. When $B = 1$, BAG converges to the minimum $\ell_2$-norm solution of (4).

**Proposition 2.** *Assume that for each $b \in [B]$, each submatrix $X_{:,\mathcal{G}_b} \in \mathbb{R}^{n \times d_b}$ has full row rank. BAG (with $\theta_1$ initialized to 0) converges to an optimal solution $\theta_*$ of (4). For each $b \in [B]$, the subvector $\theta_{*,\mathcal{G}_b}$ of $\theta_*$ equals $X_{:,\mathcal{G}_b}^T (X_{:,\mathcal{G}_b} X_{:,\mathcal{G}_b}^T)^{-1} u_b$ for some $u_b \in \mathbb{R}^n$ and $\sum_{b \in [B]} u_b = y$.*

### 3.3 REGRET ANALYSIS

To further illustrate the advantages of blockwise adaptivity over coordinate-wise adaptivity, we consider the online convex learning setting. At round $t$, the learner picks $\theta_t$, and suffers a loss $f_t(\theta_t)$. After $T$ rounds, the learner wants to achieve a low regret w.r.t. an optimal $\theta_* = \arg\min_\theta \sum_{t=1}^T f_t(\theta)$ in hindsight:

$$R(T) \equiv \sum_{t=1}^T f_t(\theta_t) - \sum_{t=1}^T f_t(\theta_*) \equiv \sum_{t=1}^T f_t(\theta_t) - \inf_\theta \sum_{t=1}^T f_t(\theta). \tag{8}$$

We make the following assumptions.

**Assumption 1.** *Each $f_t$ in (8) is convex but possibly nonsmooth. There exists a subgradient $g \in \partial f_t(\theta)$ such that $f_t(\theta') \geq f_t(\theta) + \langle g, \theta' - \theta \rangle$ for all $\theta, \theta'$.*

**Assumption 2.** *Each parameter block is in a ball of the corresponding optimal block throughout the iterations. In other words, for all $b \in [B]$, $\max_t \|\theta_{t,\mathcal{G}_b} - \theta_{*,\mathcal{G}_b}\|_2 \leq D_b$ for some $D_b$, where $\theta_{*,\mathcal{G}_b}$ is the subvector of $\theta_*$ in block $b$.*

When $B = 1$, this reduces to the common assumption in online convex learning Duchi & Singer (2009). When $B > 1$, it naturally encodes the heterogeneity of model parameters.

**Theorem 1.** *Suppose that Assumptions 1 and 2 hold. Then,*

$$R(T) \leq \sum_{b=1}^B \left[ \frac{1}{2\eta\sqrt{d_b}} D_b^2 + \eta\sqrt{d_b} \right] \|g_{1:T,\mathcal{G}_b}\|_2. \tag{9}$$

Assume that $\max_t \|\theta_t - \theta_*\|_\infty \leq D_\infty$ for some constant $D_\infty$. When $B = d$, the above regret bound reduces to that of Adagrad (Theorem 5 of (Duchi et al., 2011)) by setting $D_b = D_\infty$ for all $b \in [B]$.

In the following, we show that when gradient magnitudes for elements in the same block have the same upper bound, blockwise adaptive learning can have lower regret than coordinate-wise adaptive learning. As a deep network can be naturally divided into blocks (examples will be given in Section 5.2) and parameters in the same block are likely to have gradients with similar magnitudes (which is verified empirically in Appendix B), blockwise adaptivity can be more beneficial.

**Corollary 1.** *Assume that $\mathbf{E}[g_{t,i}^2] \leq \sigma_b^2$ for all $i \in \mathcal{G}_b$. The expectation of (9) can be bounded as:*

$$\mathbf{E}[R(T)] \leq \sum_{b=1}^B \sigma_b \left[ \frac{1}{2\eta} D_b^2 + \eta d_b \right] \sqrt{T}. \tag{10}$$

*When $B = d$ (which corresponds to Adagrad), Assumption 2 becomes $\max_t(\theta_{t,i} - \theta_{*,i}) \leq D_i$ for some $D_i$. Expectation of the bound in (9) then reduces to*

$$\mathbf{E}[R(T)] \leq \sum_{b=1}^B \sigma_b \left[ \frac{1}{2\eta} \sum_{i \in \mathcal{G}_b} D_i^2 + \eta d_b \right] \sqrt{T}. \tag{11}$$

Assuming that Assumption 2 is tight in the sense that $D_b^2 \leq \sum_{i \in \mathcal{G}_b} D_i^2$. The bound in (10) is then smaller than that in (11). Intuitively, when gradients in a block have similar magnitudes in expectation, we have from the weak law of large numbers that $v_{t,b} = \sum_{i \in \mathcal{G}_b} v_{t,i}/d_b$ (where $v_{t,b}$ and $v_{t,i}$ are as defined in Algorithms 2 and 1, respectively) is a better estimate of $\mathbf{E}[v_{t,i}]$ than $v_{t,i}$ for a single coordinate $i$. This implies using blockwise adaptivity may lead to better performance.

## 4 BLOCKWISE ADAPTIVE GRADIENT WITH MOMENTUM (BAGM)

In Algorithm 2, $v_{t,b}$'s are increasing w.r.t. $t$. The update suffers from vanishing stepsize, making slow progress on nonconvex problems such as deep network training. To alleviate this problem, many Adagrad variants (such as RMSprop, Adam and weighted AdaEMA (Zou et al., 2019)) use weighted moving average momentum. In this paper, we extend the use of blockwise adaptive stepsize to weighted AdaEMA (Section 4.1), which includes Adam and RMSprop as special cases. The proposed algorithm will be called blockwise adaptive gradient with momentum (BAGM). Sections 4.2 and 4.3 then study its convergence and generalization properties. Note that as BAG is a special case of BAGM, the analysis there also apply to BAG.

### 4.1 PROPOSED ALGORITHM

The proposed BAGM is shown in Algorithm 4. Here, $m_t$ serves as an exponential moving averaged momentum, $\{\beta_t\}$ is a sequence of momentum parameters, and $a_t$'s assign different weights to the past gradients in the accumulation of variance. Note from Algorithm 4 that the variance estimate can be rewritten as

$$\hat{v}_{t,b} = \sum_{i=1}^{t} \frac{a_i}{A_t} \frac{\|g_{i,\mathcal{G}_b}\|_2^2}{d_b} = \frac{1}{\sum_{j=1}^{t} a_j} \sum_{i=1}^{t} a_i \frac{\|g_{i,\mathcal{G}_b}\|_2^2}{d_b}. \tag{12}$$

In particular, we will consider the three weight sequences $\{a_t\}$ introduced in (Zou & Shen, 2018). **S.1**: $a_t = a$ for some $a > 0$; **S.2**: $a_t = t^\tau$ for some $\tau > 0$; The fraction $a_t/A_t$ in (12) then decreases as $\mathcal{O}(1/t)$. **S.3**: $a_t = \alpha^{-t}$ for some $0 < \alpha < 1$: It can be shown that this is equivalent to using the exponential moving average estimate: $v_{t,b} = \alpha v_{t-1,b} + (1-\alpha)\frac{\|g_{t,\mathcal{G}_b}\|_2^2}{d_b}$, and $\hat{v}_{t,b} = \frac{v_{t,b}}{1-\alpha^t}$.

With $\beta_t = 0$, weight sequence **S.1**, and $\epsilon = 0$, BAGM reduces BAG. When $B = d$ and $\epsilon = 0$, BAGM reduces to weighted AdaEMA. As weighted AdaEMA includes many Adagrad variants, the proposed BAGM also covers the corresponding blockwise variants.

---

**Algorithm 3** Weighted AdaEMA for stochastic nonconvex optimization.

1: **Input**: $\{\eta_t\}$; $\{a_t\}$; $\{\beta_t\}$; $\epsilon > 0$.
2: **initialize** $\theta_1$; $v_0 \leftarrow 0$; $m_0 \leftarrow 0$; $A_0 \leftarrow \epsilon$
3: **for** $t = 1, 2, \ldots, T$ **do**
4:     Sample an unbiased stochastic gradient $g_t$
5:     $A_t = A_{t-1} + a_t$
6:     **for** $i = 1, 2, \ldots, d$ **do**
7:         $v_{t,i} = v_{t-1,i} + a_t g_{t,i}^2$
8:         $\hat{v}_{t,i} = v_{t,i}/A_t$
9:         $m_{t,i} = \beta_t m_{t-1,i} + (1-\beta_t)g_{t,i}$
10:        $\theta_{t+1,i} = \theta_{t,i} - \eta_t m_{t,i}/\sqrt{\hat{v}_{t,i}}$
11:     **end for**
12: **end for**

---

**Algorithm 4** BAGM: Blockwise adaptive gradient with momentum for stochastic nonconvex optimization.

1: **Input**: $\{\eta_t\}$; $\{a_t\}$; $\{\beta_t\}$; $\epsilon > 0$.
2: **initialize** $\theta_1$; $v_0 \leftarrow 0$; $m_0 \leftarrow 0$; $A_0 \leftarrow 0$
3: **for** $t = 1, 2, \ldots, T$ **do**
4:     Sample an unbiased stochastic gradient $g_t$
5:     $A_t = A_{t-1} + a_t$
6:     **for** $b = 1, 2, \ldots, B$ **do**
7:         $v_{t,b} = v_{t-1,b} + a_t\|g_{t,\mathcal{G}_b}\|_2^2/d_b$
8:         $\hat{v}_{t,b} = v_{t,b}/A_t$
9:         $m_{t,\mathcal{G}_b} = \beta_t m_{t-1,\mathcal{G}_b} + (1-\beta_t)g_{t,\mathcal{G}_b}$
10:        $\theta_{t+1,\mathcal{G}_b} = \theta_{t,\mathcal{G}_b} - \eta_t m_{t,\mathcal{G}_b}/(\sqrt{\hat{v}_{t,b}}+\epsilon)$
11:     **end for**
12: **end for**

---

### 4.2 CONVERGENCE ANALYSIS ON NONCONVEX PROBLEMS

**Assumption 3.** *$F$ in (1) is lower-bounded (i.e., $F(\theta_*) = \inf_\theta F(\theta) > -\infty$) and $L$-smooth.*

**Assumption 4.** *Each block of stochastic gradient has bounded second moment, i.e., $\mathbf{E}_t[\|g_{t,\mathcal{G}_b}\|_2^2]/d_b \leq \sigma_b^2, \forall b \in [B], \forall t$, where the expectation is taken w.r.t. the random $f_t$.*

Assumption 4 implies the variance of each block of stochastic gradient is upper-bounded by $d_b\sigma_b^2$ (i.e., $\mathbf{E}_t[\|g_{t,\mathcal{G}_b} - \nabla_{\mathcal{G}_b}F(\theta_t)\|_2^2] = \mathbf{E}_t[\|g_{t,\mathcal{G}_b}\|_2^2] - \|\nabla_{\mathcal{G}_b}F(\theta_t)\|_2^2 \leq d_b\sigma_b^2$). This naturally encodes the notion of heterogeneous gradient in a multi-layer neural network. When $B = 1$, this reduces to the usual second moment bound in stochastic approximation (Shamir & Zhang, 2013; Zou et al., 2019).

**Assumption 5.** *$0 \leq \beta_t \leq \beta$ for some $0 \leq \beta < 1$.*

Assumption 5 allows us to use, for example, a constant $\beta_t = \beta$, a decreasing sequence $\beta_t = \beta/t^\tau$, or an increasing sequence $\beta_t = \beta(1 - 1/t^\tau)$.

**Assumption 6.** *(i) $\{a_t\}$ is non-decreasing; (ii) $a_t$ grows slowly such that $\{A_{t-1}/A_t\}$ is non-decreasing and $A_t/(A_{t-1} + a_1) \leq \omega$ for some $\omega \geq 0$; (iii) $p \equiv \lim_{t \to \infty} A_{t-1}/A_t > \beta^2$.*

Assumption 6 is satisfied by the three weight sequences introduced above. Specifically, for **S.1**: $\omega = 1$ and $p = 1$; **S.2**: $\omega = (1 + 2^\tau)/2$ and $p = 1$; **S.3**: $\omega = (1 + 1/\alpha)/2$ and $p = \alpha > \beta^2$.

**Assumption 7.** *(Zou et al., 2019) The stepsize $\eta_t$ is chosen such that $w_t = \eta_t/\sqrt{a_t/A_t}$ is "almost" non-increasing, i.e., there exists a non-increasing sequence $\{z_t\}$ and positive constants $C_1$ and $C_2$ such that $C_1 z_t \leq w_t \leq C_2 z_t$ for all $t$.*

Assumption 7 is satisfied by the weights sequences **S.1**, **S.2**, **S.3** when

$$\eta_t = \eta/\sqrt{t} \tag{13}$$

for some $\eta > 0$. Interested readers are referred to (Zou et al., 2019) for details.

**Proposition 3.** *Suppose that Assumptions 3-7 hold. With probability at least $1 - \delta^{2/3}$, $\min_{1 \leq t \leq T} \|\nabla F(\theta_t)\|_2^2 \leq \frac{1}{\delta} \mathcal{O}(\log(T)/\sqrt{T})$ for* **S.1** *and* **S.2***; and $\min_{1 \leq t \leq T} \|\nabla F(\theta_t)\|_2^2 \leq \frac{1}{\delta} \mathcal{O}(1)$ for* **S.3**.

Note that SGD, with the decreasing stepsize in (13), converges at a rate of $O(\log(T)/T)$ (Ghadimi & Lan, 2013). Thus, the rates for **S.1** and **S.2** are as good as SGD. Though **S.3** only leads to an $\mathcal{O}(1)$ bound, it has good performance in practice (Kingma & Ba, 2015; Zaheer et al., 2018). Moreover, recall that when $B = d$ and $\epsilon = 0$, BAGM reduces to weighted AdaEMA. In this case, Proposition 3 obtains the same convergence rates as in (Zou et al., 2019).

Next, we consider $B = \tilde{B}$ for some $\tilde{B} \neq d$ (blockwise stepsize), and compare it with $B = d$ (coordinate-wise stepsize). This requires the following assumption, which is slightly stronger than Assumption 4 (that only bounds the expectation).

**Assumption 8.** $\|g_{t,\mathcal{G}_b}\|_2^2/d_b \leq G_b^2, \forall b \in [B]$ *and* $\forall t$.

Note that when $B = d$, Assumption 8 becomes $g_{t,i}^2 \leq G_i^2$ for some $G_i$, and Assumption 4 becomes $\mathbf{E}_t[g_{t,i}^2] \leq \sigma_i^2$ for some $\sigma_i$. Let $\tilde{\mathcal{G}}_b$ be the set of indices in block $b$ when $B = \tilde{B}$. The following Corollary shows that BAGM has faster convergence than its coordinate-wise counterpart when $\{\sigma_i^2\}_{i \in \tilde{\mathcal{G}}_b}$ have low variability. This also agrees with our observation in Section 3.3 that blockwise adaptivity can have lower regret under this condition.

**Corollary 2.** *Suppose that Assumptions 3-8 hold. Define $r_1 \equiv \frac{\sum_{b=1}^{\tilde{B}} \sum_{i \in \tilde{\mathcal{G}}_b} \log\left(\sigma_i^2/\epsilon^2+1\right)}{\sum_{b=1}^{\tilde{B}} d_b \log\left(\sigma_b^2/\epsilon^2+1\right)}$, $r_2 \equiv \frac{\sum_{b=1}^{\tilde{B}} \sum_{i \in \tilde{\mathcal{G}}_b} \sigma_i}{\sum_{b=1}^{\tilde{B}} \sigma_b d_b}$ and $r_3 \equiv \frac{\sum_{b=1}^{\tilde{B}} \sum_{i \in \tilde{\mathcal{G}}_b} \sigma_i \log\left(\sigma_i^2/\epsilon^2+1\right)}{\sum_{b=1}^{\tilde{B}} \sigma_b d_b \log\left(\sigma_b^2/\epsilon^2+1\right)}$. Let $\tilde{C}_d(T)/\delta$ (resp. $\tilde{C}_{\tilde{B}}(T)/\delta$) be the high probability upper bound on $\min_{1 \leq t \leq T} \|\nabla F(\theta_t)\|_2^2$ when $B = d$ (resp. $B = \tilde{B}$). If $\min(1, r_1, r_2, r_3)\sqrt{\frac{\max_b \max_{i \in \tilde{\mathcal{G}}_b} G_i^2 + \epsilon^2}{\max_b G_b^2 + \epsilon^2}} \geq 1$, then $\tilde{C}_d(T) \geq \tilde{C}_{\tilde{B}}(T)$.*

Note that $\sqrt{\frac{\max_b \max_{i \in \tilde{\mathcal{G}}_b} G_i^2 + \epsilon^2}{\max_b G_b^2 + \epsilon^2}} \geq 1$ when Assumption 8 is tight. By comparing the denominator and numerator in $r_1, r_2, r_3$, it can be seen that $\min(r_1, r_2, r_3)$ is close to or greater than 1 when $\{\sigma_i^2\}_{i \in \tilde{\mathcal{G}}_b}$ have low variability. These will be verified empirically in Appendix B.

### 4.3 UNIFORM STABILITY AND GENERALIZATION ERROR

As in Definition 1, let $S, S'$ be two data sets of size $n$ that differ in only one sample, and the $t$th iterates of BAGM on $S$ and $S'$ by $\theta_t$ and $\theta_t'$, respectively. Let $\Delta_t = \|\theta_t - \theta_t'\|_2$, and $\tilde{\Delta}_t(z) = |f(\theta_t; z) - f(\theta_t'; z)|$. The following Proposition allows us to study how $B$ affects the growth of $\mathbf{E}[\tilde{\Delta}_t(z)]$, where the expectation is taken w.r.t. randomness of the algorithm.

**Proposition 4.** *Suppose that Assumptions 3-7 hold. Assume that $f$ is $\tilde{\gamma}$-Lipschitz[1], $\beta_t = 0$, and the initial $\theta_1, \theta_1'$ values are the same. We have $\sup_{S,S'} \sup_z \mathbf{E}[\tilde{\Delta}_{t+1}(z)] \leq \frac{2\tilde{\gamma} C_2}{nC_1} \sqrt{\left[w_1^2 \sum_{b=1}^{B} d_b \log\left(\sigma_b^2/\epsilon^2 + 1\right) + d\omega \sum_{k=1}^{t} \eta_k^2\right] t} + \left(1 - \frac{1}{n}\right) \tilde{\gamma} W_t$, where $W_t = \tilde{\gamma} \sum_{k=1}^{t} \eta_k \mathbf{E}\left[\max_b \left|1/(\sqrt{\hat{v}_{k,b}} + \epsilon) - 1/(\sqrt{\hat{v}_{k,b}'} + \epsilon)\right|\right] + L \sum_{k=1}^{t} \eta_k \mathbf{E}\left[\Delta_k/(\sqrt{\min_b \hat{v}_{k,b}} + \epsilon)\right]$.*

Recall that $\sigma_b^2 \leq \frac{1}{d_b} \sum_{i \in \tilde{\mathcal{G}}_b} \sigma_i^2$. If $\sigma_b^2 = \frac{1}{d_b} \sum_{i \in \tilde{\mathcal{G}}_b} \sigma_i^2$, the first term on the RHS of the above bound is smallest when $B = d$; otherwise, some $B < d$ will make this term smallest. For the $\min_b \hat{v}_{k,b}$ term inside $W_t$, this reduces to $\frac{1}{\tilde{B}} \sum_{b=1}^{\tilde{B}} \hat{v}_{k,b}$ when $B = 1$, and to $\min_b \min_{i \in \tilde{G}_b} \hat{v}_{k,i}$ when $B = d$.

---

[1] In other words, $|f(\theta; z) - f(\theta'; z)| \leq \tilde{\gamma}\|\theta - \theta'\|_2$ for any $z$.

As $\frac{1}{\tilde{B}}\sum_{b=1}^{\tilde{B}}\hat{v}_{k,b}\geq\min_b\hat{v}_{k,b}\geq\min_b\min_{i\in\tilde{G}_b}\hat{v}_{k,i}$, this $\min_b\hat{v}_{k,b}$ term is the smallest when $B=d$, and is largest when $B=1$. As for the other terms in $W_t$, the first term is small when $B$ is close to $d$, and the second term is small when $B$ approaches 1. Hence, for $B$ equals some $1<\tilde{B}<d$, $\sup_{S,S'}\sup_z \mathbf{E}[\tilde{\Delta}_{t+1}(z)]$, and thus the generalization error, grows slower than those of $B=d$ and $B=1$. Besides, Proposition 4 also indicates that a larger $\epsilon$ makes the bound smaller.

Note that the uniform stability bound of SGD (Theorem 3.8 in [12]) is not directly comparable with our Proposition 4. However, as SGD and $B=1$ both depend on the same second moment upper bound (Assumption 4), by showing $B=\tilde{B}$ is better than $B=1$, we expect blockwise adaptivity to be also better than SGD.

## 5    EXPERIMENTS

In Section 5.1, we first empirically validate the regret analysis results of BAG (Section 3.3) on a linear model. In Section 5.2, we run deep networks on a number of standard benchmark data sets including CIFAR-10 (Section 5.2.1), ImageNet (Section 5.3), and WikiText-2 (Section 5.4). As the focus is on deep learning, we only use BAGM (instead of BAG) in this section.

### 5.1    ILLUSTRATION OF THE REGRET ANALYSIS RESULTS

In this section, we use BAG on the linear model $f_t(\theta_t)=\max(0,1-y_t\langle\theta_t,x_t\rangle)$, with input $x_t\in\mathbb{R}^d$ generated in a blockwise manner and $y_t\in\{-1,1\}$ is the label for $x_t$. Assume that $x_t$ is partitioned into $\tilde{B}$ blocks, whose structure may be different from that of the $B$ gradient blocks. With probability $p_b$, each element $x_{t,i}$ in input block $b$ is sampled from $\mathcal{N}(c_b y_t,\gamma_b^2)$ for some scaling factor $c_b$ and variance $\gamma_b^2$, and $x_{t,i}=0$ otherwise. It can be easily shown that for elements in the same input block $b$, their expected gradient magnitudes have the same upper bound ($\mathbf{E}[g_{t,i}^2]\leq p_b(c_b^2+\gamma_b^2)$).

We generate a synthetic data set, with $d=100$, using the above procedure. The first 50 features of $x_t$ are sampled independently from $\mathcal{N}(10y_t,100)$ with probability 0.5, and zero otherwise. The last 50 features are sampled independently from $\mathcal{N}(-5y_t,25)$ with probability 0.4, and zero otherwise. The class label $y_t\in\{-1,1\}$ are sampled randomly with equal probabilities.

We study BAG with $B=1,2,3,4$ and 100. For $B=2$, gradient $g_t$ is partitioned in the same way as the input. For $B=3$, we form the first block using the first 35 coordinates, the second block with the next 30 coordinates, and the third block with the remaining 35 elements. For $B=4$, $g_t$ is divided into four blocks each of 25 elements. We initialize $\theta_1$ to zero, fix $\epsilon=10^{-8}$ and $\eta=0.01$.

Figure 1 compares the expected regret, which is estimated by averaging the regrets over 100 repetitions. BAG with $B=2$ and 4 achieve lower regrets than the others (as $B=4$ covers the case for $B=2$). BAG with $B=3$ is a little worse as its block structure is different from the input block structure, but still performs better than $B=d$. For $B=1$, the mismatch in block structures is severe and its performance is worst.

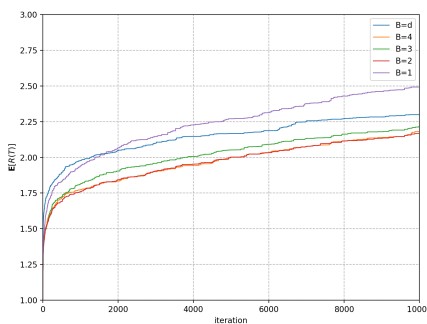

Figure 1: Expected regret with different $B$'s (note that curves for $B=2$ and 4 overlap).

### 5.2    REAL-WORLD DATA SETS

We introduce four block construction strategies: **B.1**: Use a single adaptive stepsize for each parameter tensor/matrix/vector. A parameter tensor can be the kernel tensor in a convolution layer, a parameter matrix can be the weight matrix in a fully-connected layer, and a parameter vector can be a bias vector; **B.2**: Use an adaptive stepsize for each output dimension of the parameter matrix/vector in a fully connected layer, and an adaptive stepsize for each output channel in the convolution layer; **B.3**: Use an adaptive stepsize for each output dimension of the parameter matrix/vector in a fully

connected layer, and an adaptive stepsize for each kernel in the convolution layer; **B.4**: Use an adaptive stepsize for each input dimension of the parameter tensor/matrix, and an adaptive stepsize for each parameter vector. More details on the implementation can be found in Appendix A

We compare the proposed BAGM (with block construction approaches **B.1**, **B.2**, **B.3**, **B.4**) with the following baselines: (i) Nesterov's accelerated gradient (NAG) (Sutskever et al., 2013); and (ii) Adam (Kingma & Ba, 2015). These two algorithms are widely used in deep networks. NAG provides a strong baseline with good generalization performance, while Adam serves as a fast counterpart with coordinate-wise adaptive stepsize.

As grid search for all hyper-parameters is very computationally expensive, we only tune the most important ones using a validation set and fix the rest. We use a constant $\beta_t = \beta$ (momentum parameter) and exponential increasing sequence **S.3** with $\alpha = 0.999$ for BAGM. For Adam, we fix its second moment parameter to 0.999 and tune its momentum parameter. Note that with such configurations, Adam is a special case of BAGM with $B = d$ (i.e., weighted AdaEMA). For all the adaptive methods, we use $\epsilon = 10^{-3}$ as suggested in (Zaheer et al., 2018). All the experiments are run on a AWS p3.16 instance with 8 NVIDIA V100 GPUs.

### 5.2.1 CIFAR-10

We train a deep residual network from the MXNet Gluon CV model zoo Guo et al. (2019) on the CIFAR-10 data set. We use the 56-layer and 110-layer networks as in (He et al., 2016). 10% of the training data are carved out as validation set. We perform grid search using the validation set for the initial stepsize $\eta$ and momentum parameter $\beta$ on ResNet56. The obtained hyperparameters are then also used on ResNet110. We follow a similar setup as in (He et al., 2016). Details are in Appendix A.2. To reduce statistical variance, results are averaged over 5 repetitions.

| | ResNet56 | ResNet110 | | top-1 error (%) | top-5 error (%) |
|---|---|---|---|---|---|
| NAG | $6.91 \pm 0.15$ | $6.28 \pm 0.23$ | NAG | 20.94 | 5.51 |
| Adam | $6.64 \pm 0.30$ | $6.35 \pm 0.18$ | Adam | 21.04 | 5.47 |
| BAGM-**B.1** | $\mathbf{6.26 \pm 0.12}$ | $\mathbf{5.94 \pm 0.09}$ | BAGM-**B.1** | **20.79** | 5.43 |
| BAGM-**B.2** | $6.51 \pm 0.14$ | $6.27 \pm 0.18$ | BAGM-**B.2** | 20.90 | **5.39** |
| BAGM-**B.3** | $6.52 \pm 0.33$ | $6.31 \pm 0.06$ | BAGM-**B.3** | 20.88 | 5.52 |
| BAGM-**B.4** | $6.38 \pm 0.40$ | $6.02 \pm 0.15$ | BAGM-**B.4** | 20.82 | 5.48 |

Table 1: Testing errors (%) on CIFAR-10. The best results are bolded.

Table 2: Validation set errors on ImageNet. The best results are bolded.

Table 1 shows the testing errors of the various methods. With a large $\epsilon = 10^{-3}$, the testing performance of Adam matches that of NAG. This agrees with (Zaheer et al., 2018) that a larger $\epsilon$ reduces adaptivity and improves generalization performance. It also agrees with Proposition 4 that the bound is smaller when $\epsilon$ is larger. As for BAGM, it outperforms Adam for all block construction schemes used. It also outperforms NAG with schemes **B.1**, **B.2** and **B.4**.

Convergence of the training, testing, and generalization errors (absolute difference between training error and testing error) are shown in Figure 2.[2] As can be seen, on both ResNet models, BAGM-**B.1** converges to a lower training error rate than Adam. This agrees with Corollary 2 that blockwise adaptive methods can have faster convergence than their counterparts with element-wise adaptivity. Moreover, the generalization error of BAGM-**B.1** is smaller than Adam, which agrees with Proposition 4 that blockwise adaptivity can have a slower growth of generalization error. On both models, BAGM-**B.1** gives the smallest generalization error, while NAG has the highest generalization error on ResNet56. Hence, the proposed methods can accelerate convergence and improve generalization.

### 5.3 IMAGENET

In this experiment, we train a 50-layer ResNet model on ImageNet (Russakovsky et al., 2015). The data set has 1000 classes, 1.28M training samples, and 50,000 validation images. As the data set does not come with labels for its test set, we evaluate its generalization performance on the validation

---

[2]To reduce clutterness, we only show results of the block construction scheme BAGM-**B.1**, which gives the lowest testing error among the proposed block schemes. The full results are shown in Figure 3.

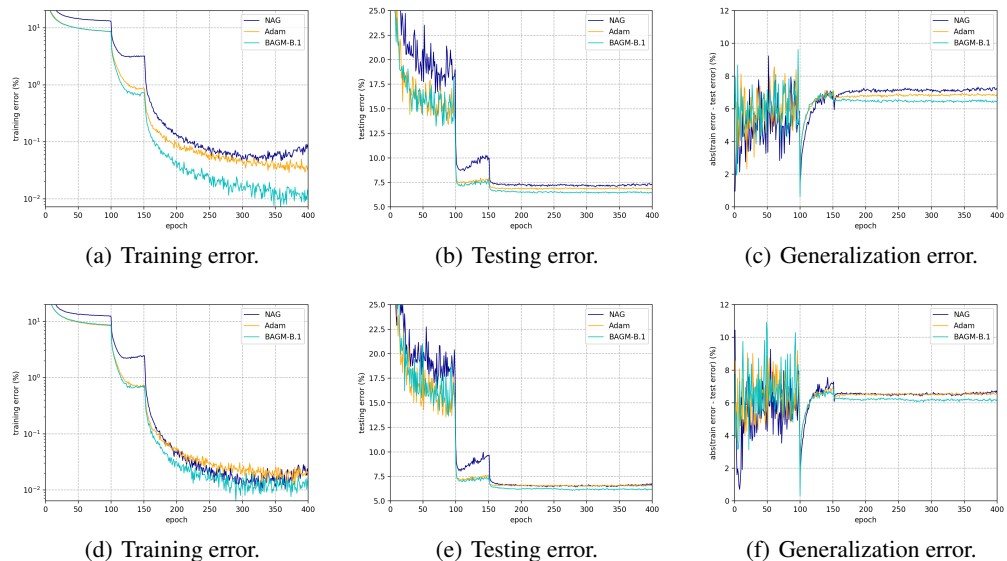

(a) Training error.    (b) Testing error.    (c) Generalization error.

(d) Training error.    (e) Testing error.    (f) Generalization error.

Figure 2: Results on CIFAR-10. Top: ResNet56; Bottom: ResNet110. Note that the training error (%) is plotted in log scale.

set. We use the ResNet50_v1d network from the MXNet Gluon CV model zoo. We train the FP16 (half precision) model on 8 GPUs, each of which processes 128 images in each iteration. More details are in Appendix A.3. As it takes long time to train on ImageNet, we only run each algorithm once.

Performance on the validation set is shown in Table 2. As can be seen, BAGM with all the block schemes (particularly BAGM-**B.1**) achieve lower top-1 errors than Adam and NAG. As for the top-5 error, BAGM-**B.2** obtains the lowest, which is then followed by BAGM-**B.1**. Overall, BAGM-**B.1** has the best performance on both CIFAR-10 and ImageNet.

## 5.4 WORD-LEVEL LANGUAGE MODELING

In this section, we train the AWD-LSTM word-level language model (Merity et al., 2018) on the WikiText-2 (WT2) data set (Merity et al., 2017). We use the publicly available implementation in the Gluon NLP toolkit Guo et al. (2019). We perform grid search on the initial learning rate and momentum parameter as in Section 5.2.1, and set the weight decay to $1.2 \times 10^{-6}$ as in (Merity et al., 2018). Results are averaged over 3 repetitions. More details on the setup are in Appendix A.4. As there is no convolutional layer, **B.2** and **B.3** are the same.

Table 3 shows the testing perplexities, the lower the better. As can be seen, all adaptive methods achieve lower test perplexities than NAG, and BAGM-**B.2** obtains the best result.

| NAG | Adam | BAGM-**B.1** | BAGM-**B.2** | BAGM-**B.4** |
|---|---|---|---|---|
| $65.75 \pm 0.10$ | $65.40 \pm 0.13$ | $65.42 \pm 0.10$ | $\mathbf{65.29} \pm 0.14$ | $65.55 \pm 0.07$ |

Table 3: Testing perplexities on the WikiText-2 data set. The best results are bolded.

## 6 CONCLUSION

In this paper, we proposed adapting the stepsize for each parameter block, instead of for each individual parameter as in Adam and RMSprop. Regret, convergence and uniform stability analyses show that it can have lower regret, faster convergence and lower generalization error than its counterpart with coordinate-wise adaptive stepsize. Experiments on synthetic dataset, image classification and language modeling confirm these theoretical results.

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

# A  Experimental Setup

## A.1  Implementation Details

As $\{a_t\}$ is non-decreasing, the accumulated sum $A_t$ can grow significantly, which may potentially cause some numerical issue. In practice, using steps 7 and 8 in Algorithms 4, we equivalently rewrite the update in (12) as the following exponentially moving update:

$$\hat{v}_{t,b} = \alpha_t \hat{v}_{t-1,b} + (1 - \alpha_t) \frac{\|g_{t,\mathcal{G}_b}\|_2^2}{d_b},$$

where $\alpha_t = 1 - a_t/A_t$. If $a_t = \alpha^{-t}$, then $\alpha_t = \alpha(1 - \alpha^{t-1})/(1 - \alpha^t)$. Based on Proposition 3, this setting leads to an $\mathcal{O}(1)$ bound. On the other hand, if $a_t = t^\tau$, then we have $a_t/A_t = \mathcal{O}(1/t)$. This suggests that we can use polynomial-decay averaging $\alpha_t = 1 - (c+1)/(t+c)$ for some $c \geq 0$ (Shamir & Zhang, 2013), whereas $c > 0$ reduces the weight of earlier iterates compared to later ones. The larger $c$ corresponds to the larger $\tau$. In this case, as $\sum_{t=1}^{T} a_t = \mathcal{O}(T^\gamma)$ for some $\gamma > 0$, we have a convergence rate of $\mathcal{O}(\log(T)/T)$.

## A.2  CIFAR-10

The CIFAR-10 data set has 50,000 training images and 10,000 testing images. As in (He et al., 2016), we employ data augmentation for training. We first pad the input picture by adding 4 pixels on each side of the image. Then, a $32 \times 32$ crop is randomly sampled from the padded image with random horizontal flipping. A mini-batch size of 128 is used. The stepsize is divided by 10 at the 39k-th and 59k-th iterations. We use a weight decay of 0.0001.

For NAG, the initial stepsize $\eta$ is chosen from $\{0.01, 0.05, 0.1, 0.5, 1\}$. For the adaptive methods, we have $\eta \in \{0.0001, 0.0005, 0.001, 0.005, 0.01\}$. The momentum parameter is searched over $\{0, 0.5, 0.9\}$. The learning rate is multiplied by 0.1 at the 100th and 150th epochs. We perform grid search on the hyper-parameters by running each algorithm for 200 epochs on ResNet56. The hyper-parameters that give the highest accuracy on the validation set are employed. The testing performance is obtained by running each algorithm with its best hyper-parameters on full training set for 400 epochs. The same obtained hyperparameters are then used on training ResNet110. When NAG is applied to ResNet110, we use a smaller learning rate at the beginning to warm up the training. Specifically, the obtained learning rate is divided by 10 in the first 4000 iterations, and then go back to the original one and continue training. The grid search results are shown in Table 4.

Figure 3 shows that, on ResNet56, BAGM converges to a lower training error rate than Adam for all schemes used. For the deeper ResNet100 model, BAGM-**B.1** and **B.4** have faster convergence than Adam, while BAGM-**B.2** and **B.3** show the same convergence speed with Adam.

|       | $\eta$ | $\beta$ |
|-------|--------|---------|
| NAG   | 0.5    | 0.9     |
| Adam  | 0.005  | 0       |
| BAGM-**B.1** | 0.005 | 0 |
| BAGM-**B.2** | 0.005 | 0 |
| BAGM-**B.3** | 0.005 | 0 |
| BAGM-**B.4** | 0.005 | 0 |

Table 4: The best $\eta$ and $\beta$ obtained by grid search on CIFAR-10.

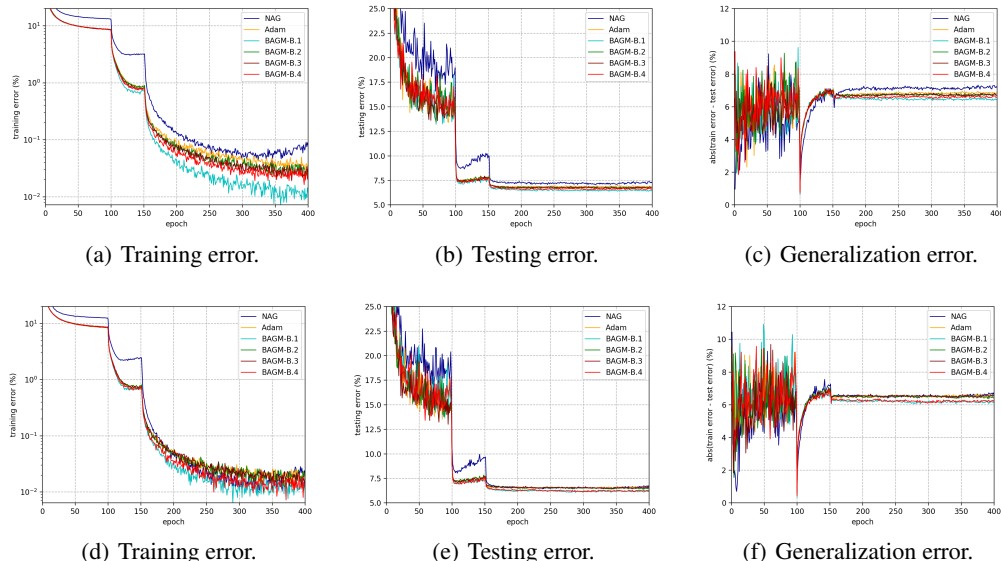

| (a) Training error. | (b) Testing error. | (c) Generalization error. |

| (d) Training error. | (e) Testing error. | (f) Generalization error. |

Figure 3: Results on CIFAR-10 for the proposed method with all four block construction schemes and the baselines. Top: ResNet56; Bottom: ResNet110. The training error (%) is plotted on a logarithmic scale.

### A.3    IMAGENET

In this experiment, we employ label smoothing and mixup (Zhang et al., 2018). The cosine schedule (Loshchilov & Hutter, 2017) for learning rate is used. A warmup of 5 epochs is applied. During validation, we use the center crop. Hyperparameter tuning is based on the obtained results in Section A.2. Specifically, for NAG, the initial learning rate is chosen from $\{0.4, 0.5\}$, and momentum parameter is fixed to 0.9. For Adam and BAGM, we have the initial learning rate $\eta \in \{0.004, 0.005\}$, and we use momentum parameter $\beta = 0$. A weight decay of 0.0001 is used (weight decay is not applied to bias vectors, and parameters for batch normalization layers) [3]. The best learning rates for each method are shown in Table 5.

### A.4    WORD LANGUAGE MODELING

In this experiment, we follow the same setting in (Merity et al., 2018). A 3-layer AWD-LSTM is considered. The model is unrolled for 70 steps, and a mini-batch of size 80 is used. We clip the norm of the gradients at 0.25. The details of the configuration used in this experiment can be found in `https://github.com/dmlc/gluon-nlp/blob/master/scripts/`

---

[3]The example script for running NAG with $\eta = 0.4$ can be found in `https://raw.githubusercontent.com/dmlc/web-data/master/gluoncv/logs/classification/imagenet/resnet50_v1d-mixup.sh`. Details of the data augmentation can be found in `https://github.com/dmlc/gluon-cv/blob/master/scripts/classification/imagenet/train_imagenet.py`.

| | $\eta$ |
|---|---|
| NAG | 0.4 |
| Adam | 0.004 |
| BAGM-**B.1** | 0.004 |
| BAGM-**B.2** | 0.004 |
| BAGM-**B.3** | 0.004 |
| BAGM-**B.4** | 0.004 |

Table 5: The best $\eta$ obtained by grid search on ImageNet.

`language_model/word_language_model.py`. For completeness, we show the model configuration in Table 6.

| | dimensionality/dropout rate |
|---|---|
| embedding size | 400 |
| hidden size | 1150 |
| dropout | 0.4 |
| dropout for RNN layers | 0.2 |
| dropout for input embedding layers | 0.65 |
| dropout to remove words from embedding layer | 0.1 |
| weight dropout | 0.5 |

Table 6: Model configuration of AWD-LSTM model.

As the WikiText-2 data set comes with a validation set, we perform grid search by evaluating the performance on the validation set. For NAG, the initial stepsize is chosen from $\{1, 3, 10, 30\}$. For the adaptive methods, we select $\eta \in \{0.1, 0.03, 0.01, 0.003\}$. The momentum parameter varies in $\{0, 0.5, 0.9\}$. The learning rate is multiplied by 0.1 when the validation performance does not improve for 30 consecutive epochs. We tie the word embeddings and softmax weights. For each algorithm, we employ the iterate averaging scheme proposed in (Merity et al., 2018). The model is trained for 750 epochs. The hyper-parameters obtained by grid search are shown in Table 7. In general, **B.1** and **B.4** are not suitable for updating the word embedding matrix as word frequency varies a lot and thus the gradient is highly sparse. However, the gradient becomes dense when we use weight tying. In modern toolkits such as Tensorflow, MXNet, and Pytorch, the weight matrices of the LSTM gates are concatenated to speed up matrix-vector multiplication. We need to apply **B.1** and **B.4** to these weight matrices separately.

| | $\eta$ | $\beta$ |
|---|---|---|
| NAG | 30 | 0 |
| Adam | 0.03 | 0.5 |
| BAGM-**B.1** | 0.03 | 0.9 |
| BAGM-**B.2** | 0.03 | 0.5 |
| BAGM-**B.4** | 0.03 | 0.5 |

Table 7: The best $\eta$ and $\beta$ obtained by grid search on the word language modeling experiment.

# B    GRADIENTS IN PARAMETER BLOCK

As discussed in Corollary 2, BAGM can have faster convergence than its coordinate-wise counterpart when $\{\sigma_i^2\}_{i \in \tilde{\mathcal{G}}_b}$ have low variability. In this section, we verify this experimentally. Using the setup in Section 5.2.1, we focus on BAGM-**B.1**, which shows the fastest convergence. At the end of each epoch, we perform 10 full data passes with random shuffle and data augmentation (as described in Appendix A.2) to compute $\mathbf{E}[g_i^2]$ and $\mathbf{E}[\|g_{\tilde{\mathcal{G}}_b}\|_2^2]/d_b$. We then approximate $\sigma_i^2$ and $\sigma_b^2$ by their empirical maxima over all epochs. Figure 4 shows the coefficient of variation of $\{\sigma_i^2\}_{i \in \tilde{\mathcal{G}}_b}$, which is defined as the ratio of the standard deviation to the mean (Everitt, 2006). As can be seen, around 86% (resp. 75%) of all blocks for ResNet56 (resp. ResNet110) have coefficient of variation smaller than 1, indicating that $\{\sigma_i^2\}_{i \in \tilde{\mathcal{G}}_b}$ have low variance and concentrate around the mean.

We also compute the empirical values of $r_{\min} \equiv \min(r_1, r_2, r_3)$ and $\sqrt{\frac{\max_b \max_{i \in \tilde{\mathcal{G}}_b} G_i^2 + \epsilon^2}{\max_b G_b^2 + \epsilon^2}}$ in Corollary 2. We use the two symbols ($\bar{v}_{T,B}$ and $C(T)$) that are defined in Appendix F for the corresponding main theorem. Moreover, we estimate $\sqrt{\frac{\bar{v}_{T,d} + \epsilon^2}{\bar{v}_{T,\tilde{B}} + \epsilon^2}}$, where $\bar{v}_{T,d} = \bar{v}_{T,B=d}$ and $\bar{v}_{T,\tilde{B}} = \bar{v}_{T,B=\tilde{B}}$, instead of $\sqrt{\frac{\max_b \max_{i \in \tilde{\mathcal{G}}_b} G_i^2 + \epsilon^2}{\max_b G_b^2 + \epsilon^2}}$, as $C(T)$ is tighter than $\tilde{C}(T)$. We estimate $\bar{v}_{T,\tilde{B}}$ using $\max_{1 \le t \le T} \max_b \hat{v}_{t,b}$. The results are shown in Table B.

| | ResNet56 | | ResNet110 | |
|---|---|---|---|---|
| | $\sqrt{\frac{\bar{v}_{T,d} + \epsilon^2}{\bar{v}_{T,\tilde{B}} + \epsilon^2}}$ | $\min(r_1, r_2, r_3)$ | $\sqrt{\frac{\bar{v}_{T,d} + \epsilon^2}{\bar{v}_{T,\tilde{B}} + \epsilon^2}}$ | $\min(r_1, r_2, r_3)$ |
| B.1 | 3.70 | 1.02 | 3.30 | 1.01 |
| B.2 | 1.50 | 1.06 | 1.73 | 1.05 |
| B.3 | 1.47 | 1.01 | 1.39 | 1.01 |
| B.4 | 2.27 | 1.17 | 2.47 | 1.13 |

Table 8: Empirical estimate of $\sqrt{\frac{\bar{v}_{T,d} + \epsilon^2}{\bar{v}_{T,\tilde{B}} + \epsilon^2}}$ and $\min(r_1, r_2, r_3)$.

As can been seen, all $\min(r_1, r_2, r_3)$ are larger than 1, indicating $\{\sigma_i^2\}_{i \in \tilde{\mathcal{G}}_b}$ have low variability for all the proposed blocks, which agree with Corollary 2 and explain why the proposed blockwise adaptivity leads to faster convergence. Moreover, B.1 has largest $\sqrt{\frac{\bar{v}_{T,d} + \epsilon^2}{\bar{v}_{T,\tilde{B}} + \epsilon^2}}$. Theoretically, it means B.1 should lead to fastest convergence, which is empirically verified in Figure 3. Similarly, B.2 has second largest $\sqrt{\frac{\bar{v}_{T,d} + \epsilon^2}{\bar{v}_{T,\tilde{B}} + \epsilon^2}}$ and is therefore the second fastest.

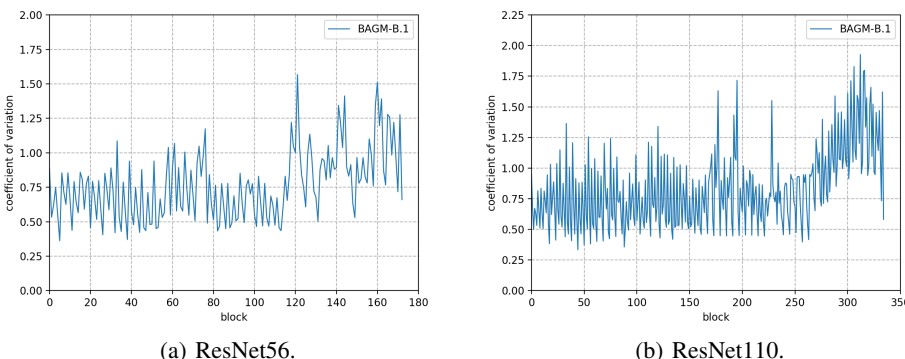

(a) ResNet56.  (b) ResNet110.

Figure 4: Coefficient of variation of $\{\sigma_i^2\}_{i \in \tilde{\mathcal{G}}_b}$ for all the blocks with **B.1**. The blocks with higher indices in the abscissa belong to deeper layers.

## C  PROOF OF PROPOSITION 1

*Proof.* In this proof, we use denominator layout for matrix calculus. As all the activation functions are bijective and $\{W_k\}_{k=l+1}^L$ are invertible, $\Phi_l$ is bijective and has an inverse function $\Phi_l^{-1}$. Specifically, $\Phi_l^{-1}$ is given by

$$\Phi_l^{-1}(Y) = \phi_l^{-1}(\cdots \phi_{L-2}^{-1}(\phi_{L-1}^{-1}(YW_L^{-1})W_{L-1}^{-1}) \cdots W_{l+1}^{-1}).$$

Then, with the assumption that $H_{l-1}$ has full row rank, the nonconvex objective (5) can be reformulated as the following convex problem:

$$\min_{W_l} \|H_{l-1}W_l - \Phi_l^{-1}(Y)\|_2^2. \tag{14}$$

It is obvious that its large margin solution is $H_{l-1}^T(H_{l-1}H_{l-1}^T)^{-1}\Phi_l^{-1}(Y)$. In the sequel, we will see that every critical point of (5) is a global optimal solution. Let $h_{i,l-1}$ denotes a column vector

that is the $i$-th row of $H_{l-1}$ and $Z_{:,i}$ be the $i$-th column of matrix $Z$. The gradient of (5) is

$$2H_{l-1}^T \sum_{k=1}^{d} \text{Diag}(\Phi_l(H_{l-1}W_l)_{:,k} - Y_{:,k})G_{k,l} = H_{l-1}^T E_l,$$

where $G_{k,l} = [\nabla_{x=h_{1,l-1}^T W_l} \Phi_l(x)_k; \cdots ; \nabla_{x=h_{n,l-1}^T W_l} \Phi_l(x)_k] \in \mathbb{R}^{n \times d}$ and $E_l = 2\sum_{k=1}^{d} \text{Diag}(\Phi_l(H_{l-1}W_l)_{:,k} - Y_{:,k})G_{k,l}$ to be the error matrix. As $H_{l-1}$ has full row rank, then clearly gradient is zero if only and if $E_l = 0$. By the definition of $G_{k,l}$, we can see that $E_l = 0$ if only and if $\Phi_l(H_{l-1}W_l) = Y$ when $\nabla_{x=h_{i,l-1}^T W_l} \Phi_l(x)$ has full row rank for all $i \in [n]$. Note that the gradient $\nabla_{x=h_{i,l-1}^T W_l} \Phi_l(x)$ is of the following form:

$$\nabla_{x=h_{i,l-1}^T W_l} \Phi_l(x) = (W_L \circ \phi_{L-1}'(h_{i,L-2}^T W_{L-1})^T 1_d^T)^T \cdots (W_{l+1} \circ \phi_l'(h_{i,l-1}^T W_l)^T 1_d^T)^T,$$

where $\circ$ is the Hadamard product. For all $k \in \{l, \ldots, L-1\}$, as $W_{k+1}$ has full rank and $\phi_k'(z) \neq 0$ for any $z \in \mathbb{R}$, we have that $W_{k+1} \circ \phi_k'(h_{i,k-1}^T W_k)^T 1_d^T$ has full rank. Applying the fact that the multiplication of a number of invertible matrices preserves full rank, we obtain that $\nabla_{x=h_{i,l-1}^T W_l} \Phi_l(x)$ has full rank. Therefore, every critical point satisfies $\Phi_l(H_{l-1}W_l) = Y$ and every critical point is a global optimal solution.

Let $i_t$ be the index chosen at iteration $t$ and $y_{i_t}$ be the $i_t$-th row of $Y$. Let us define $e_{t,l} = 2\sum_{k=1}^{d} (\Phi_l(h_{i_t,l-1}^T W_{t,l})_k - y_{i_t,k}) \nabla_{x=h_{i_t,l-1}^T W_{t,l}} \Phi_l(x)_k$. Now, we prove that if the following update rule applied on (5) finds a critical point, then the iterate converges to the largest margin solution.

$$W_{t+1,l} = W_{t,l} - \eta_{t,l} h_{i_t,l-1} e_{t,l} = H_{l-1}^T \left( -\sum_{j=1}^{t} \eta_{j,l} \tilde{E}_{j,l} \right), \tag{15}$$

where we use $W_{l,1} = 0$, $\eta_{t,l}$ is the stepsize for $l$-th layer at iteration $t$, and $\tilde{E}_{j,l}$ is a matrix in which its $i_k$-th row is $e_{j,l}$ and all the other rows are zeros. Then, the solution found by (15) lies in the span of rows of $H_{l-1}$. In other words, the solution has the following parametric form:

$$W_l = H_{l-1}^T \alpha_l$$

for some $\alpha_l \in \mathbb{R}^n$. Thus, if (15) is converging to a critical point in expectation, then we have $W_{t,l} \to W_{*,l}$ as $t \to \infty$, where $W_{*,l} = H_{l-1}^T \alpha_{*,l}$ for some optimal $\alpha_{*,l}$. Since every critical point is an optimal solution, then $W_{*,l}$ is also a solution to (14), and we have

$$\Phi_l^{-1}(Y) = H_{l-1}W_{*,l} = H_{l-1}H_{l-1}^T \alpha_{*,l}.$$

We solve for $\alpha_{*,l}$ and obtain

$$\alpha_{*,l} = (H_{l-1}H_{l-1}^T)^{-1}\Phi_l^{-1}(Y).$$

Therefore, $W_{*,l} = H_{l-1}^T (H_{l-1}H_{l-1}^T)^{-1}\Phi_l^{-1}(Y)$. $\qquad \square$

## D  PROOF OF PROPOSITION 2

*Proof.* Let $(x_{i_t}, y_{i_t})$ be the pair of sample selected at iteration $t$. The stochastic gradient of least square problem (4) at the $t$-th iteration is

$$2(x_{i_t}^T \theta_t - y_{i_t})x_{i_t} = X^T e_t,$$

where we define $e_t$ to be the error vector with value $2(x_{i_t}^T \theta_t - y_{i_t})$ in the $i_t$-th coordinate and zeros elsewhere. For each block $b$, BAG with $\theta_1 = 0$ uses the following update rule:

$$\theta_{t+1,\mathcal{G}_b} = \theta_{t,\mathcal{G}_b} - \eta_{t,b} X_{:,\mathcal{G}_b}^T e_t = X_{:,\mathcal{G}_b}^T \left( -\sum_{i=1}^{t} \eta_{i,b} e_i \right),$$

where $\eta_{t,b} = \eta/(\sqrt{\sum_{i=1}^{t} \|X_{:,\mathcal{G}_b}^T e_i\|_2^2/d_b} + \epsilon)$. Then, each subvector of the solution found by BAG lies in the span of rows of $X_{:,\mathcal{G}_b}$. In other words, each subvector of the solution is of the following parametric form:

$$\theta_{\mathcal{G}_b} = X_{:,\mathcal{G}_b}^T \alpha_b$$

for some $\alpha_b \in \mathbb{R}^n$. Combining with Theorem 1 and online-to-batch conversion (Cesa-Bianchi et al., 2004), BAG is converging in expectation $\frac{1}{t} \sum_{i=1}^{t} \theta_i \to \theta_*$ as $t \to \infty$, where $\theta_{*,\mathcal{G}_b} = X_{:,\mathcal{G}_b}^T \alpha_{*,b}$ for some optimal $\alpha_{*,b}$. Since $\theta_*$ is a solution to (4), we have

$$y = X\theta_* = \sum_{b=1}^{B} X_{:,\mathcal{G}_b} X_{:,\mathcal{G}_b}^T \alpha_{*,b}.$$

Assume that each submatrix $X_{:,\mathcal{G}_b}$ has full row rank, then $X_{:,\mathcal{G}_b} X_{:,\mathcal{G}_b}^T$ is invertible, we can solve for $\alpha_{*,b}$'s and obtain

$$\alpha_{*,b} = (X_{:,\mathcal{G}_b} X_{:,\mathcal{G}_b}^T)^{-1} u_b$$

for some $u_b \in \mathbb{R}^n$ and $\sum_{b=1}^{B} u_b = y$. $\qquad\square$

## E    PROOF OF THEOREM 1

**Lemma 1.** *Let $\{\theta_t\}$ be the sequence generated by the Algorithm 2. Define $s_t = [(\sqrt{v_{t,1}} + \epsilon)1_{d_1}^T, \dots, (\sqrt{v_{t,B}} + \epsilon)1_{d_B}^T]^T$. Let $H_t = Diag(s_t)$. Then, for any $\theta$, we have*

$$f_t(\theta_t) - f_t(\theta) \le \frac{1}{2\eta}\|\theta_t - \theta\|_{H_t}^2 - \frac{1}{2\eta}\|\theta_{t+1} - \theta\|_{H_t}^2 + \frac{\eta}{2}\|g_t\|_{H_t^{-1}}^2.$$

*Proof.* For any $\theta$, the convexity of $f_t$ indicates that

$$
\begin{aligned}
&f_t(\theta_t) - f_t(\theta) \\
&\le \quad \langle g_t, \theta_t - \theta \rangle \\
&= \quad \langle g_t, \theta_{t+1} - \theta \rangle + \langle g_t, \theta_t - \theta_{t+1} \rangle \\
&= \quad \frac{1}{\eta}\langle \theta_{t+1} - \theta, H_t(\theta_t - \theta_{t+1}) \rangle + \langle g_t, \theta_t - \theta_{t+1} \rangle \\
&= \quad \frac{1}{2\eta}\|\theta_t - \theta\|_{H_t}^2 - \frac{1}{2\eta}\|\theta_{t+1} - \theta\|_{H_t}^2 - \frac{1}{2\eta}\|\theta_{t+1} - \theta_t\|_{H_t}^2 + \langle g_t, \theta_t - \theta_{t+1} \rangle \\
&\le \quad \frac{1}{2\eta}\|\theta_t - \theta\|_{H_t}^2 - \frac{1}{2\eta}\|\theta_{t+1} - \theta\|_{H_t}^2 - \frac{1}{2\eta}\|\theta_{t+1} - \theta_t\|_{H_t}^2 + \frac{1}{2\eta}\|\theta_{t+1} - \theta_t\|_{H_t}^2 + \frac{\eta}{2}\|g_t\|_{H_t^{-1}}^2 \\
&= \quad \frac{1}{2\eta}\|\theta_t - \theta\|_{H_t}^2 - \frac{1}{2\eta}\|\theta_{t+1} - \theta\|_{H_t}^2 + \frac{\eta}{2}\|g_t\|_{H_t^{-1}}^2,
\end{aligned}
$$

where the second to last inequality follows from Fenchel's inequality applied to the conjugate functions $\frac{1}{2\eta}\|\cdot\|_{H_t}^2$ and $\frac{\eta}{2}\|\cdot\|_{H_t^{-1}}^2$. $\qquad\square$

**Lemma 2.** *Considering an arbitrary R-valued sequence $\{a_i\}$ and its vector representation $a_{1:t} = [a_1, \dots, a_t]$, we have*

$$\sum_{t=1}^{T} \frac{a_t^2}{\|a_{1:t}\|_2} \le 2\|a_{1:T}\|_2.$$

*Proof.* The lemma can be proved by induction. The lemma trivially holds when $T = 1$. Assume the lemma holds for $T - 1$, we get

$$
\begin{aligned}
\sum_{t=1}^{T} \frac{a_t^2}{\|a_{1:t}\|_2} &\le \quad 2\|a_{1:T-1}\|_2 + \frac{a_T^2}{\|a_{1:T}\|_2} \\
&= \quad 2\sqrt{Z - x} + \frac{x}{\sqrt{Z}},
\end{aligned}
$$

where we define $Z = \|a_{1:T}\|_2^2$ and $x = a_T^2$. As the RHS is non-increasing for $x \geq 0$. We can set $x = 0$ to maximize the bound and obtain $2\sqrt{Z}$. $\qquad\square$

**Lemma 3.** *Let $H_t$ be defined as in Lemma 1. Denote $g_{1:t,\mathcal{G}_b} = [g_{1,\mathcal{G}_b}^T, \ldots, g_{t,\mathcal{G}_b}^T]^T$. We have*

$$\sum_{t=1}^{T} \|g_t\|_{H_t^{-1}}^2 \leq 2\sum_{b=1}^{B} \sqrt{d_b}\|g_{1:T,\mathcal{G}_b}\|_2.$$

*Proof.*

$$\begin{aligned}
\sum_{t=1}^{T} \|g_t\|_{H_t^{-1}}^2 &\leq \sum_{t=1}^{T} \langle g_t, \mathrm{Diag}(s_t)^{-1}g_t\rangle \\
&= \sum_{t=1}^{T}\sum_{b=1}^{B} \frac{\sqrt{d_b}\|g_{t,\mathcal{G}_b}\|_2^2}{\|g_{1:t,\mathcal{G}_b}\|_2} \\
&= \sum_{b=1}^{B} \sqrt{d_b} \sum_{t=1}^{T} \frac{\|g_{t,\mathcal{G}_b}\|_2^2}{\|g_{1:t,\mathcal{G}_b}\|_2} \\
&\leq 2\sum_{b=1}^{B} \sqrt{d_b}\|g_{1:T,\mathcal{G}_b}\|_2.
\end{aligned}$$

where the last inequality follows from the Lemma 2 by setting $a_i = \|g_{i,\mathcal{G}_b}\|_2^2$. $\qquad\square$

### E.1 PROOF OF THEOREM 1

*Proof.* By summing up the equation in Lemma 1 with $\theta = \theta_*$, we obtain

$$\sum_{t=1}^{T} f_t(\theta_t) - f_t(\theta_*) \leq \frac{1}{2\eta}\|\theta_1 - \theta_*\|_{H_1}^2 + \frac{1}{2\eta}\sum_{t=1}^{T-1}\left[\|\theta_{t+1} - \theta_*\|_{H_{t+1}}^2 - \|\theta_{t+1} - \theta_*\|_{H_t}^2\right] + \frac{\eta}{2}\sum_{t=1}^{T} \|g_t\|_{H_t^{-1}}^2.$$

By the construction of $H_t$, we have that $H_{t+1} \succeq H_t$. Then, we get

$$\begin{aligned}
\|\theta_{t+1} &- \theta_*\|_{H_{t+1}}^2 - \|\theta_{t+1} - \theta_*\|_{H_t}^2 \\
&= \langle \theta_{t+1} - \theta_*, \mathrm{Diag}(s_{t+1} - s_t)(\theta_{t+1} - \theta_*)\rangle \\
&= \sum_{b=1}^{B} \|\theta_{t+1,\mathcal{G}_b} - \theta_{*,\mathcal{G}_b}\|_2^2(\sqrt{v_{t+1,b}} - \sqrt{v_{t,b}}).
\end{aligned}$$

Given the above result, we have

$$\begin{aligned}
\sum_{t=1}^{T-1}&\left[\|\theta_{t+1} - \theta_*\|_{H_{t+1}}^2 - \|\theta_{t+1} - \theta_*\|_{H_t}^2\right] \\
&= \sum_{b=1}^{B}\sum_{t=1}^{T-1} \|\theta_{t+1,\mathcal{G}_b} - \theta_{*,\mathcal{G}_b}\|_2^2(\sqrt{v_{t+1,b}} - \sqrt{v_{t,b}}) \\
&= \sum_{b=1}^{B}\sum_{t=1}^{T-1} \|\theta_{t+1,\mathcal{G}_b} - \theta_{*,\mathcal{G}_b}\|_2^2(\sqrt{v_{t+1,b}} - \sqrt{v_{t,b}}) + \sum_{b=1}^{B} \|\theta_{1,\mathcal{G}_b} - \theta_{*,\mathcal{G}_b}\|_2^2(\sqrt{v_{1,b}} - \sqrt{v_{1,b}}) \\
&\leq \sum_{b=1}^{B} D_b^2\sqrt{v_{T,b}} - \sum_{b=1}^{B} \|\theta_{1,\mathcal{G}_b} - \theta_{*,\mathcal{G}_b}\|_2^2\sqrt{v_{1,b}}.
\end{aligned}$$

Recall that $v_{T,b} = \|g_{1:T,\mathcal{G}_b}\|_2^2/d_b$. Let $\epsilon = 0$. Combining Lemma 3 with the fact that $\|\theta_1 - \theta_*\|_{H_1}^2 = \sum_{b=1}^{B} \|\theta_{1,\mathcal{G}_b} - \theta_{*,\mathcal{G}_b}\|_2^2\sqrt{v_{1,b}}$, we have

$$\sum_{t=1}^{T} f_t(\theta_t) - f_t(\theta_*) \leq \frac{1}{2\eta}\sum_{b=1}^{B} \frac{D_b^2}{\sqrt{d_b}}\|g_{1:T,\mathcal{G}_b}\|_2 + \eta\sum_{b=1}^{B} \sqrt{d_b}\|g_{1:T,\mathcal{G}_b}\|_2.$$

$\qquad\square$

### E.2 PROOF OF COROLLARY 1

*Proof.* Taking expectation of the gradient terms in (9), we have, for all $b$'s,

$$\mathbf{E}[\|g_{1:T,\mathcal{G}_b}\|_2] \leq \sqrt{\sum_{i\in\mathcal{G}_b}\sum_{t=1}^{T}\mathbf{E}[g_{t,i}^2]} \leq \sigma_b\sqrt{d_b T}.$$

Let $B = \tilde{B}$. Then, (9) reduces to

$$\mathbf{E}[R(T)] \leq \sum_{b=1}^{\tilde{B}} \sigma_b \left[\frac{1}{2\eta}D_b^2 + \eta d_b\right]\sqrt{T}.$$

$\square$

## F MAIN THEOREM

Let $\sigma_{t,b} = \sqrt{\mathbf{E}_t[\|g_{t,\mathcal{G}_b}\|_2^2]}$. We define a sequence of virtual estimates of the second moment:

$$
\begin{aligned}
\tilde{v}_{t,b} &= \frac{1}{A_t}(v_{t-1,b} + \frac{a_t}{d_b}\mathbf{E}_t[\|g_{t,\mathcal{G}_b}\|_2^2]) \\
&= \frac{1}{A_t}\left(\sum_{i=1}^{t-1} a_i \frac{\|g_{i,\mathcal{G}_b}\|_2^2}{d_b} + a_t \frac{\sigma_{t,b}^2}{d_b}\right) \forall b \in [B].
\end{aligned}
\tag{16}
$$

Let $\hat{v}_{T,B} \equiv \max_{1\leq t\leq T}\mathbf{E}[\max_b \tilde{v}_{t,b}]$

**Theorem 2.** *Suppose that Assumptions 3-7 hold. Let $\rho = \beta^2/\tilde{p}$. We have*

$$\min_{1\leq t\leq T}(\mathbf{E}[\|\nabla F(\theta_t)\|_2^{4/3}])^{3/2} \leq C(T),$$

*where*[4]

$$
\begin{aligned}
C(T) &= \frac{\sqrt{2(\bar{v}_{T,B}+\epsilon^2)}}{\eta_T T}\left[\frac{2C_2}{(1-\beta)C_1}C_0 + C_4\left[\frac{\beta}{\sqrt{C_a(1-\rho)}}\sum_{b=1}^{B}\sigma_b d_b \sum_{t=2}^{T}w_t\left(\sqrt{\frac{A_t}{A_{t-1}}}-1\right)\right.\right. \\
&\quad \left.\left. + \sum_{b=1}^{B}C_b'\left[w_1\log\left(\frac{\sigma_b^2}{\epsilon^2}+1\right) + \omega\sum_{t=1}^{T}\eta_t\sqrt{\frac{a_t}{A_t}}\right]\right]\right],
\end{aligned}
\tag{17}
$$

$C_0 = F(\theta_1) - F(\theta_*)$, $C_4 = \frac{2C_2^2}{C_1^2\sqrt{C_a}(1-\sqrt{\rho})(1-\beta)}$, $C_b' = \frac{LC_2^3 w_1 d_b}{C_1^3 C_a(1-\sqrt{\rho})^2} + \frac{2C_3^2 C_2 \sigma_b d_b}{C_1}$, *and* $C_3 = \frac{\beta/(1-\beta)}{\sqrt{C_a A_1/A_2(1-\rho)}} + 1$.

When $B = d$, the bound here is tighter than that in (Zou et al., 2019), as we exploit heterogeneous second-order upper bound (Assumption 4). Note that constants $C_0, C_3, C_4$ are not related to block partition. They only depend on initial optimality gap and sequences $\{\eta_t\}, \{a_t\}, \{\beta_t\}$. In the following, we introduce several lemmas to prove the Theorem 2.

In the sequel, we define $H_t$ as

$$H_t = \text{Diag}(s_t),$$

where

$$s_t = [(\sqrt{\hat{v}_{t,1}}+\epsilon)1_{d_1}^T, \ldots, (\sqrt{\hat{v}_{t,B}}+\epsilon)1_{d_B}^T]^T.$$

Let $\delta_t = \theta_{t+1} - \theta_t = -\eta_t m_t/s_t$. We introduce $\tilde{H}_t$ as

$$\tilde{H}_t = \text{Diag}(\tilde{s}_t),$$

where $\tilde{v}_{t,b}$ is defined in (16) and

$$\tilde{s}_t = [(\sqrt{\tilde{v}_{t,1}}+\epsilon)1_{d_1}^T, \ldots, (\sqrt{\tilde{v}_{t,B}}+\epsilon)1_{d_B}^T]^T.$$

Assume that $\sigma_{t,b}/\sqrt{d_b} \leq \sigma_b$ for all $t$ and let $\Sigma = \text{Diag}([\sigma_1^2 1_{d_1}^T, \ldots, \sigma_B^2 1_{d_B}^T]^T)$.

---

[4]When $T = 1$, the second term in $C(T)$ (involving summation from $t = 2$ to $T$) disappears.

**Lemma 4.** *Let $S_t = S_0 + \sum_{i=1}^{t} a_i$, where $\{a_t\}$ is a non-negative sequence and $S_0 > 0$. We have*

$$\sum_{t=1}^{T} \frac{a_t}{S_t} \leq \log(S_T) - \log(S_0)$$

*Proof.* The concavity of log leads to $\log(b) \leq \log(a) + \frac{1}{a}(b - a)$ for all $a, b > 0$. This suggests that

$$\frac{a - b}{a} \leq \log(a) - \log(b) = \log\left(\frac{a}{b}\right).$$

Hence, we have

$$\sum_{t=1}^{T} \frac{a_t}{S_t} = \sum_{t=1}^{T} \frac{S_t - S_{t-1}}{S_t} \leq \sum_{t=1}^{T} \log\left(\frac{S_t}{S_{t-1}}\right) = \log(S_T) - \log(S_0).$$

$\square$

**Lemma 5.** *Let $\{a_t\}$ and $\{s_t\}$ be two real number sequences, and let $S_t = \sum_{i=1}^{t} s_i$. Then, we have*

$$\sum_{t=1}^{T} a_t s_t = \sum_{t=1}^{T-1} (a_t - a_{t+1}) S_t + a_T S_T.$$

*Proof.* Let $S_0 = 0$. Expanding the summation, we obtain

$$
\begin{aligned}
\sum_{t=1}^{T} a_t s_t &= \sum_{t=1}^{T} a_t (S_t - S_{t-1}) \\
&= \sum_{t=1}^{T-1} a_t S_t - \sum_{t=1}^{T-1} a_{t+1} S_t + a_T S_T \\
&= \sum_{t=1}^{T-1} (a_t - a_{t+1}) S_t + a_t S_T
\end{aligned}
$$

$\square$

**Lemma 6.** *Assume $\{a_t\}$ is non-decreasing such that $\{A_{t-1}/A_t\}$ is non-decreasing. Define $w_t = \eta_t / \sqrt{\frac{a_t}{A_t}}$. Assume $w_t$ is "almost" non-increasing. This means there exists another non-increasing sequence $\{z_t\}$ and positive constants $C_1$ and $C_2$ such that $C_1 z_t \leq w_t \leq C_2 z_t$. Then,*

$$w_t \leq C_2/C_1 w_i \quad \text{and} \quad \eta_t \leq C_2/C_1 \eta_i$$

*for all $i < t$.*

*Proof.* For any $i < t$,

$$w_t \leq C_2 z_t \leq C_2 z_i \leq C_2/C_1 w_i.$$

Then,

$$\eta_t \leq \frac{C_2 \sqrt{a_t/A_t}}{C_1 \sqrt{a_i/A_i}} \eta_i = \frac{C_2 \sqrt{1 - A_{t-1}/A_t}}{C_1 \sqrt{1 - A_{i-1}/A_i}} \eta_i \leq C_2/C_1 \eta_i.$$

$\square$

**Lemma 7.** *Assume that $\{a_t\}$ is non-decreasing. For any block diagonal matrix $C = Diag([c_1 1_{d_1}^T, \ldots, c_B 1_{d_B}^T]^T)$ with $c_b \geq 0$ for all $b$, we have*

$$\sum_{t=1}^{T} \mathbf{E}\left[\frac{a_t}{A_t} \|H_t^{-1} g_t\|_C^2\right] \leq \sum_{b=1}^{B} c_b d_b \left[\log\left(\frac{\sigma_b^2}{\epsilon^2} + 1\right) + \log\left(\frac{1}{a_1} \sum_{i=1}^{T} a_i + 1\right)\right].$$

*Proof.*

$$
\begin{aligned}
\sum_{t=1}^{T} \frac{a_t}{A_t} \|H_t^{-1} g_t\|_C^2 &= \sum_{t=1}^{T} \sum_{b=1}^{B} \frac{c_b \frac{a_t}{A_t} \|g_{t,\mathcal{G}_b}\|_2^2}{(\sqrt{\hat{v}_{t,b}} + \epsilon)^2} \\
&\leq \sum_{t=1}^{T} \sum_{b=1}^{B} \frac{c_b \frac{a_t}{A_t} \|g_{t,\mathcal{G}_b}\|_2^2}{\hat{v}_{t,b} + \epsilon^2} \\
&= \sum_{b=1}^{B} \sum_{t=1}^{T} \frac{c_b a_t \|g_{t,\mathcal{G}_b}\|_2^2}{\sum_{i=1}^{t} a_i \|g_{i,\mathcal{G}_b}\|_2^2 / d_b + A_t \epsilon^2} \\
&= \sum_{b=1}^{B} c_b d_b \sum_{t=1}^{T} \frac{a_t \|g_{t,\mathcal{G}_b}\|_2^2}{\sum_{i=1}^{t} a_i \|g_{i,\mathcal{G}_b}\|_2^2 + d_b A_t \epsilon^2} \\
&\leq \sum_{b=1}^{B} c_b d_b \sum_{t=1}^{T} \frac{a_t \|g_{t,\mathcal{G}_b}\|_2^2}{\sum_{i=1}^{t} a_i \|g_{i,\mathcal{G}_b}\|_2^2 + d_b a_1 \epsilon^2}.
\end{aligned}
$$

Hence,

$$
\sum_{t=1}^{T} \frac{a_t}{A_t} \|H_t^{-1} g_t\|_C^2 \leq \sum_{b=1}^{B} c_b d_b \left[ \log \left( \sum_{i=1}^{T} a_i \|g_{i,\mathcal{G}_b}\|_2^2 + d_b a_1 \epsilon^2 \right) - \log(d_b a_1 \epsilon^2) \right],
$$

where the inequality follows from Lemma 4. Using Jensen's inequality, we get

$$
\begin{aligned}
\sum_{t=1}^{T} \mathbf{E} \left[ \frac{a_t}{A_t} \|H_t^{-1} g_t\|_C^2 \right] &\leq \sum_{b=1}^{B} c_b d_b \mathbf{E} \left[ \log \left( \sum_{i=1}^{T} a_i \|g_{i,\mathcal{G}_b}\|_2^2 + d_b a_1 \epsilon^2 \right) - \log(d_b a_1 \epsilon^2) \right] \\
&\leq \sum_{b=1}^{B} c_b d_b \left[ \log \left( \sum_{i=1}^{T} a_i \mathbf{E}[\|g_{i,\mathcal{G}_b}\|_2^2] + d_b a_1 \epsilon^2 \right) - \log(d_b a_1 \epsilon^2) \right] \\
&\leq \sum_{b=1}^{B} c_b d_b \left[ \log \left( d_b \sigma_b^2 \sum_{i=1}^{T} a_i + d_b a_1 \epsilon^2 \right) - \log(d_b a_1 \epsilon^2) \right] \\
&= \sum_{b=1}^{B} c_b d_b \log \left( \frac{\sigma_b^2}{a_1 \epsilon^2} \sum_{i=1}^{T} a_i + 1 \right).
\end{aligned}
$$

Using the inequality $\log(1 + ab) \leq \log(1 + a + b + ab) = \log(1 + a) + \log(1 + b)$ for $a, b \geq 0$, we have

$$
\sum_{t=1}^{T} \mathbf{E} \left[ \frac{a_t}{A_t} \|H_t^{-1} g_t\|_C^2 \right] \leq \sum_{b=1}^{B} c_b d_b \left[ \log \left( \frac{\sigma_b^2}{\epsilon^2} + 1 \right) + \log \left( \frac{1}{a_1} \sum_{i=1}^{T} a_i + 1 \right) \right].
$$

$\square$

**Lemma 8.** *Assume that $\{a_t\}$ is non-decreasing. Define $w_t = \eta_t / \sqrt{\frac{a_t}{A_t}}$. Assume $w_t$ is "almost" non-increasing. This means there exists another non-increasing sequence $\{z_t\}$ and positive constants $C_1$ and $C_2$ such that $C_1 z_t \leq w_t \leq C_2 z_t$ for all $t$. For any block diagonal matrix $C = Diag([c_1 1_{d_1}^T, \ldots, c_B 1_{d_B}^T]^T)$ with $c_b \geq 0$ for all $b$, we have*

$$
\sum_{t=1}^{T} \eta_t \mathbf{E} \left[ \sqrt{\frac{a_t}{A_t}} \|H_t^{-1} g_t\|_C^2 \right] \leq \frac{C_2}{C_1} \left[ w_1 \sum_{b=1}^{B} c_b d_b \log \left( \frac{\sigma_b^2}{\epsilon^2} + 1 \right) + \sum_{b=1}^{B} c_b d_b \sum_{t=1}^{T} \eta_t \sqrt{\frac{a_t}{A_t}} \frac{A_t}{A_{t-1} + a_1} \right].
$$

*Proof.* Let $\xi_t = \frac{a_t}{A_t}\|H_t^{-1}g_t\|_C^2$, then $\zeta_t = \sum_{i=1}^t \xi_i$. Lemma 5 indicates that we have

$$
\begin{aligned}
\sum_{t=1}^T \eta_t \sqrt{\frac{a_t}{A_t}}\|H_t^{-1}g_t\|_C^2 &= \sum_{t=1}^T w_t \xi_t \\
&\le C_2 \sum_{t=1}^T z_t \xi_t \\
&= C_2 \left[ \sum_{t=1}^{T-1}(z_t - z_{t+1})\zeta_t + z_T \zeta_T \right].
\end{aligned}
$$

Define $M_t = \sum_{b=1}^B c_b d_b \left[\log\left(\frac{\sigma_b^2}{\epsilon^2} + 1\right) + \log\left(\frac{1}{a_1}\sum_{i=1}^t a_i + 1\right)\right]$. By Lemma 7, we have $\mathbf{E}[\zeta_t] \le M_t$. Then,

$$
\begin{aligned}
\sum_{t=1}^T \eta_t \mathbf{E}\left[\sqrt{\frac{a_t}{A_t}}\|H_t^{-1}g_t\|_C^2\right] &\le C_2 \left[\sum_{t=1}^{T-1}(z_t - z_{t+1})\mathbf{E}[\zeta_t] + z_T \mathbf{E}[\zeta_T]\right] \\
&\le C_2 \left[\sum_{t=1}^{T-1}(z_t - z_{t+1})M_t + z_T M_T\right],
\end{aligned}
$$

where the last inequality follows from the assumption that $z_t \ge z_{t+1}$. Then,

$$
\begin{aligned}
\sum_{t=1}^T \eta_t \mathbf{E}\left[\sqrt{\frac{a_t}{A_t}}\|H_t^{-1}g_t\|_C^2\right] &\le C_2\left[\sum_{t=1}^{T-1}(z_t - z_{t+1})M_t + z_T M_T\right] \\
&= C_2\left[\sum_{t=1}^T z_t(M_t - M_{t-1}) + z_1 M_0\right] \\
&= C_2\left[z_1 \sum_{b=1}^B c_b d_b \log\left(\frac{\sigma_b^2}{\epsilon^2}+1\right) + \sum_{t=1}^T z_t \sum_{b=1}^B c_b d_b \log\left(\frac{A_t + a_1}{A_{t-1} + a_1}\right)\right] \\
&\le \frac{C_2}{C_1}\left[w_1 \sum_{b=1}^B c_b d_b \log\left(\frac{\sigma_b^2}{\epsilon^2}+1\right) + \sum_{t=1}^T w_t \sum_{b=1}^B c_b d_b \log\left(\frac{A_t + a_1}{A_{t-1} + a_1}\right)\right].
\end{aligned}
$$

As $\log(1+x) \le x$ for $x > -1$ and the fact that $A_t \ge A_{t-1}$, we get

$$
\log\left(\frac{A_t + a_1}{A_{t-1} + a_1}\right) = \log\left(1 + \frac{A_t + a_1}{A_{t-1} + a_1} - 1\right) \le \frac{A_t + a_1}{A_{t-1} + a_1} - 1 = \frac{a_t}{A_{t-1} + a_1}.
$$

Hence,

$$
\begin{aligned}
\sum_{t=1}^T \eta_t \mathbf{E}\left[\sqrt{\frac{a_t}{A_t}}\|H_t^{-1}g_t\|_C^2\right] &\le \frac{C_2}{C_1}\left[w_1 \sum_{b=1}^B c_b d_b \log\left(\frac{\sigma_b^2}{\epsilon^2}+1\right) + \sum_{t=1}^T w_t \sum_{b=1}^B c_b d_b \log\left(\frac{A_t + a_1}{A_{t-1} + a_1}\right)\right] \\
&\le \frac{C_2}{C_1}\left[w_1 \sum_{b=1}^B c_b d_b \log\left(\frac{\sigma_b^2}{\epsilon^2}+1\right) + \sum_{b=1}^B c_b d_b \sum_{t=1}^T \eta_t \sqrt{\frac{a_t}{A_t}}\frac{A_t}{A_{t-1} + a_1}\right].
\end{aligned}
$$

$\square$

**Lemma 9.** *Let* $\tilde{\eta}_{t,b} = \frac{\eta_t}{\sqrt{\tilde{v}_{t,b}}+\epsilon}$. *For each block $b$ and $t \ge 2$, we have*

$$
\left(\delta_t - \frac{\beta_t \eta_t}{\sqrt{1 - a_t/A_t}\eta_{t-1}}\delta_{t-1}\right)_{\mathcal{G}_b} = -(1-\beta_t)\tilde{\eta}_{t,b}g_{t,\mathcal{G}_b} + \tilde{\eta}_{t,b}\frac{\frac{a_t}{A_t d_b}\|g_{t,\mathcal{G}_b}\|_2^2}{\sqrt{\hat{v}_{t,b}}+\epsilon}X_{t,b} + \tilde{\eta}_{t,b}\frac{\sigma_{t,b}}{\sqrt{d_b}}Y_{t,b} + Z_{t,b},
$$

*where*

$$X_{t,b} = \frac{\beta_t m_{t-1,\mathcal{G}_b}}{\sqrt{\hat{v}_{t,b}} + \sqrt{A_{t-1}\hat{v}_{t-1,b}/A_t}} + \frac{(1-\beta_t)g_{t,\mathcal{G}_b}}{\sqrt{\tilde{v}_{t,b}} + \sqrt{\hat{v}_{t,b}}},$$

$$Y_{t,b} = \frac{\frac{a_t}{A_t\sqrt{d_b}}\|g_{t,\mathcal{G}_b}\|_2}{\sqrt{\hat{v}_{t,b}} + \epsilon} \frac{\beta_t m_{t-1,\mathcal{G}_b}}{\sqrt{A_{t-1}\hat{v}_{t-1,b}/A_t} + \epsilon} \frac{\sqrt{\frac{a_t}{A_t d_b}}\|g_{t,\mathcal{G}_b}\|_2}{\sqrt{\hat{v}_{t,b}} + \sqrt{A_{t-1}\hat{v}_{t-1,b}/A_t}} \frac{\sqrt{\frac{a_t}{A_t d_b}}\sigma_{t,b}}{\sqrt{\tilde{v}_{t,b}} + \sqrt{A_{t-1}\hat{v}_{t-1,b}/A_t}}$$
$$- \frac{\frac{a_t}{A_t}g_{t,\mathcal{G}_b}}{\sqrt{\hat{v}_{t,b}} + \epsilon} \frac{(1-\beta_t)\frac{\sigma_{t,b}}{\sqrt{d_b}}}{\sqrt{\tilde{v}_{t,b}} + \sqrt{\hat{v}_{t,b}}},$$

$$Z_{t,b} = \beta_t\eta_t m_{t-1,\mathcal{G}_b} \frac{\left(1 - \sqrt{A_{t-1}/A_t}\right)\epsilon}{(\sqrt{A_{t-1}\hat{v}_{t-1,b}/A_t} + \epsilon)(\sqrt{A_{t-1}\hat{v}_{t-1,b}/A_t} + \sqrt{A_{t-1}/A_t}\epsilon)}.$$

*Proof.* For any $t \geq 2$,

$$\delta_t - \frac{\beta_t\eta_t}{\sqrt{A_{t-1}/A_t}\eta_{t-1}}\delta_{t-1}$$
$$= -\frac{\eta_t m_t}{s_t} + \frac{\beta_t\eta_t m_{t-1}}{\sqrt{A_{t-1}/A_t}s_{t-1}}$$
$$= -\eta_t\left[\frac{m_t}{s_t} - \frac{\beta_t m_{t-1}}{\sqrt{A_{t-1}/A_t}s_{t-1}}\right]$$
$$= -\frac{(1-\beta_t)\eta_t g_t}{\sqrt{v_t} + \epsilon} - \beta_t\eta_t m_{t-1}\left[\frac{1}{\sqrt{v_t} + \epsilon} - \frac{1}{\sqrt{A_{t-1}v_{t-1}/A_t} + \epsilon}\right]$$
$$\quad - \beta_t\eta_t m_{t-1}\left[\frac{1}{\sqrt{A_{t-1}s_{t-1}/A_t} + \epsilon} - \frac{1}{\sqrt{A_{t-1}s_{t-1}/A_t} + \sqrt{A_{t-1}/A_t}\epsilon}\right]$$
$$= -\frac{(1-\beta_t)\eta_t g_t}{\sqrt{v_t} + \epsilon} - \beta_t\eta_t m_{t-1}\left[\frac{1}{\sqrt{v_t} + \epsilon} - \frac{1}{\sqrt{A_{t-1}v_{t-1}/A_t} + \epsilon}\right] \qquad (18)$$
$$\quad + \beta_t\eta_t m_{t-1}\frac{\left(1 - \sqrt{A_{t-1}/A_t}\right)\epsilon}{(\sqrt{A_{t-1}v_{t-1}/A_t} + \epsilon)(\sqrt{A_{t-1}v_{t-1}/A_t} + \sqrt{A_{t-1}/A_t}\epsilon)}$$

Let expand the first term of (18) as

$$\frac{(1-\beta_t)\eta_t g_t}{\sqrt{v_t} + \epsilon} = \frac{(1-\beta_t)\eta_t g_t}{\sqrt{\tilde{v}_t} + \epsilon} + (1-\beta_t)\eta_t g_t\left[\frac{1}{\sqrt{v_t} + \epsilon} - \frac{1}{\sqrt{\tilde{v}_t} + \epsilon}\right]$$
$$= \frac{(1-\beta_t)\eta_t g_t}{\sqrt{\tilde{v}_t} + \epsilon} + (1-\beta_t)\eta_t g_t\frac{\tilde{v}_t - v_t}{(\sqrt{v_t} + \epsilon)(\sqrt{\tilde{v}_t} + \epsilon)(\sqrt{\tilde{v}_t} + \sqrt{v_t})}.$$

For each block $b$, we have

$$\frac{(1-\beta_t)\eta_t g_{t,\mathcal{G}_b}}{\sqrt{\hat{v}_{t,b}} + \epsilon}$$
$$= \frac{(1-\beta_t)\eta_t g_{t,\mathcal{G}_b}}{\sqrt{\tilde{v}_{t,b}} + \epsilon} + (1-\beta_t)\eta_t g_{t,\mathcal{G}_b}\frac{\frac{a_t}{A_t d_b}(\sigma_{t,b}^2 - \|g_{t,\mathcal{G}_b}\|_2^2)}{(\sqrt{\hat{v}_{t,b}} + \epsilon)(\sqrt{\tilde{v}_{t,b}} + \epsilon)(\sqrt{\tilde{v}_{t,b}} + \sqrt{\hat{v}_{t,b}})}$$
$$= (1-\beta_t)\tilde{\eta}_{t,b}g_{t,\mathcal{G}_b} + \tilde{\eta}_{t,b}\frac{\sigma_{t,b}}{\sqrt{d_b}}\frac{\frac{a_t}{A_t}g_{t,\mathcal{G}_b}}{\sqrt{\hat{v}_{t,b}} + \epsilon}\frac{(1-\beta_t)\frac{\sigma_{t,b}}{\sqrt{d_b}}}{\sqrt{\tilde{v}_{t,b}} + \sqrt{\hat{v}_{t,b}}} - \tilde{\eta}_{t,b}\frac{\frac{a_t}{A_t d_b}\|g_{t,\mathcal{G}_b}\|_2^2}{\sqrt{\hat{v}_{t,b}} + \epsilon}\frac{(1-\beta_t)g_{t,\mathcal{G}_b}}{\sqrt{\tilde{v}_{t,b}} + \sqrt{\hat{v}_{t,b}}} \qquad (19)$$

Then, we expand the second term of (18):

$$\beta_t \eta_t m_{t-1} \left[ \frac{1}{\sqrt{v_t} + \epsilon} - \frac{1}{\sqrt{A_{t-1} v_{t-1}/A_t} + \epsilon} \right]$$

$$= \beta_t \eta_t m_{t-1} \frac{\sqrt{A_{t-1} v_{t-1}/A_t} - \sqrt{v_t}}{(\sqrt{v_t} + \epsilon)(\sqrt{A_{t-1} v_{t-1}/A_t} + \epsilon)}$$

$$= \beta_t \eta_t m_{t-1} \frac{A_{t-1} v_{t-1}/A_t - v_t}{(\sqrt{v_t} + \epsilon)(\sqrt{A_{t-1} v_t/A_t} + \epsilon)(\sqrt{v_t} + \sqrt{A_{t-1} v_{t-1}/A_t})}.$$

Similarly, for each block $b$, we have

$$\beta_t \eta_t m_{t-1,\mathcal{G}_b} \left[ \frac{1}{\sqrt{\hat{v}_{t,b}} + \epsilon} - \frac{1}{\sqrt{A_{t-1} \hat{v}_{t-1,b}/A_t} + \epsilon} \right]$$

$$= -\beta_t \eta_t m_{t-1,\mathcal{G}_b} \frac{a_t \|g_{t,\mathcal{G}_b}\|_2^2/(A_t d_b)}{(\sqrt{\hat{v}_{t,b}} + \epsilon)(\sqrt{A_{t-1} \hat{v}_{t-1,b}/A_t} + \epsilon)(\sqrt{\hat{v}_{t,b}} + \sqrt{A_{t-1} \hat{v}_{t-1,b}/A_t})}$$

$$= -\beta_t \eta_t m_{t-1,\mathcal{G}_b} \left[ \frac{a_t \|g_{t,\mathcal{G}_b}\|_2^2/(A_t d_b)}{(\sqrt{\hat{v}_{t,b}} + \epsilon)(\sqrt{\tilde{v}_{t,b}} + \epsilon)(\sqrt{\hat{v}_{t,b}} + \sqrt{A_{t-1} \hat{v}_{t-1,b}/A_t})} \right.$$

$$\left. + \frac{a_t \|g_{t,\mathcal{G}_b}\|_2^2/(A_t d_b)}{(\sqrt{\hat{v}_{t,b}} + \epsilon)(\sqrt{\hat{v}_{t,b}} + \sqrt{A_{t-1} \hat{v}_{t-1,b}/A_t})} \left[ \frac{1}{\sqrt{A_{t-1} \hat{v}_{t-1,b}/A_t} + \epsilon} - \frac{1}{\sqrt{\tilde{v}_{t,b}} + \epsilon} \right] \right]$$

$$= -\tilde{\eta}_{t,b} \frac{\frac{a_t}{A_t d_b} \|g_{t,\mathcal{G}_b}\|_2^2}{\sqrt{\hat{v}_{t,b}} + \epsilon} \frac{\beta_t m_{t-1,\mathcal{G}_b}}{\sqrt{\hat{v}_{t,b}} + \sqrt{A_{t-1} \hat{v}_{t-1,b}/A_t}}$$

$$- \tilde{\eta}_{t,b} \frac{\sigma_{t,b}}{\sqrt{d_b}} \left[ \frac{\frac{a_t}{A_t \sqrt{d_b}} \|g_{t,\mathcal{G}_b}\|_2}{\sqrt{\hat{v}_{t,b}} + \epsilon} \frac{\beta_t m_{t-1,\mathcal{G}_b}}{\sqrt{A_{t-1} \hat{v}_{t-1,b}/A_t} + \epsilon} \frac{\sqrt{\frac{a_t}{A_t d_b}} \|g_{t,\mathcal{G}_b}\|_2}{\sqrt{\hat{v}_{t,b}} + \sqrt{A_{t-1} \hat{v}_{t-1,b}/A_t}} \frac{\sqrt{\frac{a_t}{A_t d_b}} \sigma_{t,b}}{\sqrt{\tilde{v}_{t,b}} + \sqrt{A_{t-1} \hat{v}_{t-1,b}/A_t}} \right] \tag{20}$$

Combining (19) and (20) into (18), we obtain the result. $\qquad\square$

**Lemma 10.** *Suppose that $\{a_t\}$ is a non-decreasing sequence and $A_t = \sum_{i=1}^t a_t$ such that $\{A_t/A_{t+1}\}$ is non-decreasing and $\lim_{t\to\infty} \frac{A_t}{A_{t+1}} = p > 0$. Let $\hat{A}_{t,i} = \prod_{j=i+1}^t \frac{A_{j-1}}{A_j}$ for $1 \le i < t$ and $\hat{A}_{t,t} = 1$. For a fixed constant $\tilde{p}$ such that $\beta^2 < \tilde{p} < p$, we have*

$$\hat{A}_{t,i} \ge C_a \tilde{p}^{t-i} ,$$

*where $C_a = \left( \prod_{j=2}^N \frac{A_{j-1}}{A_j \tilde{p}} \right)$ and N is the maximum of the indices for which $A_{j-1}/A_j < \tilde{p}$. When there are no such indices, i.e., $A_1/A_2 \ge \tilde{p}$, we use $C_a = 1$ by convention.*

*Proof.*

$$\hat{A}_{t,i} = \prod_{j=i+1}^t \frac{A_{j-1}}{A_j} \ge \left( \prod_{j=i+1}^N \frac{A_{j-1}}{A_j} \right) \tilde{p}^{t-N} = \left( \prod_{j=i+1}^N \frac{A_{j-1}}{A_j \tilde{p}} \right) \tilde{p}^{t-i} \ge \left( \prod_{j=2}^N \frac{A_{j-1}}{A_j \tilde{p}} \right) \tilde{p}^{t-i}.$$

$$\square$$

**Lemma 11.** *Suppose that $0 \le \beta_t \le \beta < 1$ for all t. Let $\rho := \frac{\beta^2}{\tilde{p}}$, where $\tilde{p}$ is defined in Lemma 10. Then, for all t, we have*

$$\|m_{t,\mathcal{G}_b}\|_2^2 \le \frac{1}{C_a a_t/(A_t d_b)(1-\rho)} \hat{v}_{t,b},$$

*where $C_a$ is defined in Lemma 10.*

*Proof.* Let $\hat{\beta}_{t,i} = \prod_{j=i+1}^{t} \beta_j$ for $i < t$ and $\hat{\beta}_{t,t} = 1$

$$
\begin{aligned}
\|m_{t,\mathcal{G}_b}\|_2^2 &= \left\| \sum_{i=1}^{t} (1-\beta_i)\hat{\beta}_{t,i} g_{i,\mathcal{G}_b} \right\|_2^2 \\
&= \left\| \sum_{i=1}^{t} \frac{(1-\beta_i)\hat{\beta}_{t,i}}{\sqrt{\frac{a_i}{A_t d_b}}} \sqrt{\frac{a_i}{A_t d_b}} g_{i,\mathcal{G}_b} \right\|_2^2 \\
&\leq \left( \sum_{i=1}^{t} \frac{(1-\beta_i)^2 \hat{\beta}_{t,i}^2}{\frac{a_i}{A_t d_b}} \right) \left( \sum_{i=1}^{t} \frac{a_i}{A_t d_b} \|g_{i,\mathcal{G}_b}\|_2^2 \right) \\
&= \left( \sum_{i=1}^{t} \frac{(1-\beta_i)^2 \hat{\beta}_{t,i}^2}{\frac{a_i}{A_t d_b}} \right) \hat{v}_{t,b}.
\end{aligned}
\tag{21}
$$

Then, with Lemma 10, we get

$$
\sum_{i=1}^{t} \frac{(1-\beta_i)^2 \hat{\beta}_{t,i}^2}{\frac{a_i}{A_t d_b}} = \sum_{i=1}^{t} \frac{(1-\beta_i)^2 \hat{\beta}_{t,i}^2}{\frac{a_i}{A_i d_b} \hat{A}_{t,i}} \leq \frac{1}{C_a a_t/(A_t d_b)} \sum_{i=1}^{t} \left( \frac{\beta^2}{\tilde{p}} \right)^{t-i} \leq \frac{1}{C_a a_t/(A_t d_b)(1-\rho)}. \tag{22}
$$

Then, combining (21) and (22), we obtain the result. $\qquad\square$

**Lemma 12.** *Assume $F$ is $L$-smooth, $\{a_t\}$ is non-decreasing such that $\{A_{t-1}/A_t\}$ is non-decreasing and $\lim_{t\to\infty} \frac{A_t}{A_{t+1}} = p > 0$. Let $\tilde{p}$ be a constant such that $\beta^2 < \tilde{p} < p$. Assume $\mathbf{E}_t[\|g_{t,\mathcal{G}_b}\|_2^2] = \sigma_{t,b}^2 \leq d_b \sigma_b^2$. Define $w_t = \eta_t/\sqrt{\frac{a_t}{A_t}}$. Assume $w_t$ is "almost" non-increasing. This means there exists another non-increasing sequence $\{z_t\}$ and positive constants $C_1$ and $C_2$ such that $C_1 z_t \leq w_t \leq C_2 z_t$ for all $t$. Assume $0 \leq \beta_t \leq \beta < 1$ for all $t$. Define following Lyapunov function:*

$$
M_t = \mathbf{E}[\langle \nabla F(\theta_t), \delta_t \rangle + L\|\delta_t\|_2^2].
$$

*Let $C_3 \equiv \left[ \frac{\beta/(1-\beta)}{\sqrt{C_a A_1/A_2(1-\rho)}} + 1 \right]$, where $\rho := \frac{\beta^2}{\tilde{p}}$. Then, for any $t \geq 2$, we have*

$$
\begin{aligned}
M_t \leq{}& \frac{\beta_t \eta_t}{\sqrt{A_{t-1}/A_t} \eta_{t-1}} M_{t-1} - \frac{1-\beta_t}{2} \eta_t \mathbf{E}\left[ \|\nabla F(\theta_t)\|_{\tilde{H}_t^{-1}}^2 \right] + 2w_t C_3^2 \mathbf{E}\left[ \frac{a_t}{A_t} \|H_t^{-1} g_t\|_{\Sigma^{1/2}}^2 \right] \\
&+ L\mathbf{E}[\|\delta_t\|_2^2] + \frac{\beta w_t}{\sqrt{C_a(1-\rho)}} \left( \sqrt{\frac{A_t}{A_{t-1}}} - 1 \right) \sum_{b=1}^{B} \sigma_b d_b,
\end{aligned}
\tag{23}
$$

*and for $t = 1$, we have*

$$
M_1 \leq -\frac{1-\beta_1}{2} \eta_1 \mathbf{E}\left[ \|\nabla F(\theta_1)\|_{\tilde{H}_1^{-1}}^2 \right] + 2w_1 C_3^2 \mathbf{E}\left[ \frac{a_1}{A_1} \|H_1^{-1} g_1\|_{\Sigma^{1/2}}^2 \right] + L\mathbf{E}[\|\delta_1\|_2^2]. \tag{24}
$$

*Proof.* For any $t \geq 2$,

$$
\mathbf{E}[\langle \nabla F(\theta_t), \delta_t \rangle] = \frac{\beta_t \eta_t}{\sqrt{A_{t-1}/A_t} \eta_{t-1}} \mathbf{E}[\langle \nabla F(\theta_t), \delta_{t-1} \rangle] + \mathbf{E}\left[ \left\langle \nabla F(\theta_t), \delta_t - \frac{\beta_t \eta_t}{\sqrt{A_{t-1}/A_t} \eta_{t-1}} \delta_{t-1} \right\rangle \right]. \tag{25}
$$

Then, for the first term of (25), we have

$$
\begin{aligned}
\langle \nabla F(\theta_t), \delta_{t-1} \rangle &= \langle \nabla F(\theta_{t-1}), \delta_{t-1} \rangle + \langle \nabla F(\theta_t) - \nabla F(\theta_{t-1}), \delta_{t-1} \rangle \\
&\leq \langle \nabla F(\theta_{t-1}), \delta_{t-1} \rangle + L\|\theta_t - \theta_{t-1}\|_2 \|\delta_{t-1}\|_2 \\
&= \langle \nabla F(\theta_{t-1}), \delta_{t-1} \rangle + L\|\delta_{t-1}\|_2^2,
\end{aligned}
$$

where the first inequality follows from Schwartz inequality and the smoothness of the function $F$. Hence, we have

$$
\begin{aligned}
\frac{\beta_t \eta_t}{\sqrt{A_{t-1}/A_t} \eta_{t-1}} \mathbf{E}[\langle \nabla F(\theta_t), \delta_{t-1} \rangle] &\leq \frac{\beta_t \eta_t}{\sqrt{A_{t-1}/A_t} \eta_{t-1}} \mathbf{E}\left[ \langle \nabla F(\theta_{t-1}), \delta_{t-1} \rangle + L\|\delta_{t-1}\|_2^2 \right] \\
&= \frac{\beta_t \eta_t}{\sqrt{A_{t-1}/A_t} \eta_{t-1}} M_{t-1}.
\end{aligned}
$$

Now, we estimate the second term of (25). By Lemma 9, for each block $b$, we get

$$
\mathbf{E}\left[\left\langle \nabla_{\mathcal{G}_b} F(\theta_t), \delta_{t,\mathcal{G}_b} - \frac{\beta_t \eta_t}{\sqrt{A_{t-1}/A_t} \eta_{t-1}} \delta_{t-1,\mathcal{G}_b} \right\rangle\right]
$$

$$
= -(1-\beta_t)\mathbf{E}[\langle \nabla_{\mathcal{G}_b} F(\theta_t), \tilde{\eta}_{t,b} g_{t,\mathcal{G}_b}\rangle] + \mathbf{E}\left[\left\langle \nabla_{\mathcal{G}_b} F(\theta_t), \tilde{\eta}_{t,b} \frac{\frac{a_t}{A_t d_b}\|g_{t,\mathcal{G}_b}\|_2^2}{\sqrt{\hat{v}_{t,b}} + \epsilon} X_{t,b} \right\rangle\right]
$$

$$
+ \mathbf{E}\left[\left\langle \nabla_{\mathcal{G}_b} F(\theta_t), \tilde{\eta}_{t,b} \frac{\sigma_{t,b}}{\sqrt{d_b}} Y_{t,b} \right\rangle\right] + \mathbf{E}[\langle \nabla_{\mathcal{G}_b} F(\theta_t), Z_{t,b}\rangle]. \tag{26}
$$

For the first term of (26), we have

$$
-(1-\beta_t)\mathbf{E}[\langle \nabla_{\mathcal{G}_b} F(\theta_t), \tilde{\eta}_{t,b} g_{t,\mathcal{G}_b}\rangle] = -(1-\beta_t)\mathbf{E}[\langle \nabla_{\mathcal{G}_b} F(\theta_t), \tilde{\eta}_{t,b} \nabla_{\mathcal{G}_b} F(\theta_t)\rangle]
$$

$$
= -(1-\beta_t)\tilde{\eta}_{t,b}\mathbf{E}[\|\nabla_{\mathcal{G}_b} F(\theta_t)\|_2^2]. \tag{27}
$$

For the second term of (26), we have

$$
\mathbf{E}\left[\left\langle \nabla_{\mathcal{G}_b} F(\theta_t), \tilde{\eta}_{t,b} \frac{\frac{a_t}{A_t d_b}\|g_{t,\mathcal{G}_b}\|_2^2}{\sqrt{\hat{v}_{t,b}} + \epsilon} X_{t,b} \right\rangle\right]
$$

$$
\leq \mathbf{E}\left[\frac{\sqrt{\tilde{\eta}_{t,b}}\|\nabla_{\mathcal{G}_b} F(\theta_t)\|_2 \|g_{t,\mathcal{G}_b}\|_2/\sqrt{d_b}}{\sigma_{t,b}/\sqrt{d_b}} \frac{\sqrt{\tilde{\eta}_{t,b}}\frac{a_t}{A_t}\|g_{t,\mathcal{G}_b}\|_2/\sqrt{d_b}\sigma_{t,b}/\sqrt{d_b}\|X_{t,b}\|_2}{\sqrt{\hat{v}_{t,b}} + \epsilon}\right]. \tag{28}
$$

Note that

$$
\sqrt{\tilde{\eta}_{t,b}}\sigma_{t,b}/\sqrt{d_b} = \sqrt{\frac{\eta_t \sigma_{t,b}^2/d_b}{\sqrt{\tilde{v}_{t,b}} + \epsilon}} \leq \sqrt{\frac{\eta_t \sigma_{t,b}^2/d_b}{\sqrt{a_t/A_t \sigma_{t,b}^2/d_b}}} \leq \sqrt{\frac{\eta_t \sigma_b}{\sqrt{a_t/A_t}}} = \sqrt{w_t \sigma_b}. \tag{29}
$$

Besides, we have

$$
\|X_{t,b}\|_2 = \left\|\frac{\beta_t m_{t-1,\mathcal{G}_b}}{\sqrt{\hat{v}_{t,b}} + \sqrt{A_{t-1}\hat{v}_{t-1,b}/A_t}} + \frac{(1-\beta_t)g_{t,\mathcal{G}_b}}{\sqrt{\tilde{v}_{t,b}} + \sqrt{\hat{v}_{t,b}}}\right\|_2
$$

$$
\leq \left\|\frac{\beta_t m_{t-1,\mathcal{G}_b}}{\sqrt{\hat{v}_{t,b}} + \sqrt{A_{t-1}\hat{v}_{t-1,b}/A_t}}\right\|_2 + \left\|\frac{(1-\beta_t)g_{t,\mathcal{G}_b}}{\sqrt{\tilde{v}_{t,b}} + \sqrt{\hat{v}_{t,b}}}\right\|_2.
$$

With Lemma 11, we have

$$
\left\|\frac{m_{t-1,\mathcal{G}_b}}{\sqrt{\hat{v}_{t,b}} + \sqrt{A_{t-1}\hat{v}_{t-1,b}/A_t}}\right\|_2 \leq \left\|\frac{m_{t-1,\mathcal{G}_b}}{\sqrt{A_{t-1}\hat{v}_{t-1,b}/A_t}}\right\|_2 \leq \frac{1}{\sqrt{C_a A_{t-1}/A_t a_t/(A_t d_b)(1-\rho)}}, \tag{30}
$$

$$
\left\|\frac{g_{t,\mathcal{G}_b}}{\sqrt{\tilde{v}_{t,b}} + \sqrt{\hat{v}_{t,b}}}\right\|_2 \leq \left\|\frac{g_{t,\mathcal{G}_b}}{\sqrt{\hat{v}_{t,b}}}\right\|_2 \leq \left\|\frac{g_{t,\mathcal{G}_b}}{\sqrt{a_t/A_t\|g_{t,\mathcal{G}_b}\|_2^2/d_b}}\right\|_2 = \frac{\sqrt{d_b}}{\sqrt{a_t/A_t}}. \tag{31}
$$

Then, we get

$$
\|X_{t,b}\|_2 \leq \frac{\beta_t}{\sqrt{C_a A_{t-1}/A_t a_t/(A_t d_b)(1-\rho)}} + \frac{(1-\beta_t)\sqrt{d_b}}{\sqrt{a_t/A_t}}
$$

$$
= \left[\frac{\beta_t/(1-\beta_t)}{\sqrt{C_a A_{t-1}/A_t(1-\rho)}} + 1\right]\frac{(1-\beta_t)\sqrt{d_b}}{\sqrt{a_t/A_t}}
$$

$$
\leq \left[\frac{\beta/(1-\beta)}{\sqrt{C_a A_{t-1}/A_t(1-\rho)}} + 1\right]\frac{(1-\beta_t)\sqrt{d_b}}{\sqrt{a_t/A_t}}
$$

$$
\leq \left[\frac{\beta/(1-\beta)}{\sqrt{C_a A_1/A_2(1-\rho)}} + 1\right]\frac{(1-\beta_t)\sqrt{d_b}}{\sqrt{a_t/A_t}} := C_3 \frac{(1-\beta_t)\sqrt{d_b}}{\sqrt{a_t/A_t}},
$$

where the last-to-second inequality follows from the assumption that $\beta_t \leq \beta$, and the last inequality holds as we assume $\{a_t\}$ is chosen such that $\{A_{t-1}/A_t\}$ is non-decreasing for all $t$. Hence, combining the above result with (29) and (28), we have

$$
\mathbf{E}\left[\left\langle \nabla_{\mathcal{G}_b} F(\theta_t), \tilde{\eta}_{t,b} \frac{\frac{a_t}{A_t d_b}\|g_{t,\mathcal{G}_b}\|_2^2}{\sqrt{\hat{v}_{t,b}}+\epsilon} X_{t,b} \right\rangle\right]
$$

$$
\leq \mathbf{E}\left[\frac{\sqrt{\tilde{\eta}_{t,b}}\|\nabla_{\mathcal{G}_b} F(\theta_t)\|_2\|g_{t,\mathcal{G}_b}\|_2/\sqrt{d_b}}{\sigma_{t,b}/\sqrt{d_b}} \sqrt{w_t \sigma_b} C_3(1-\beta_t) \frac{\sqrt{\frac{a_t}{A_t}}\|g_{t,\mathcal{G}_b}\|_2}{\sqrt{\hat{v}_{t,b}}+\epsilon}\right]
$$

$$
\leq \mathbf{E}\left[\frac{1-\beta_t}{4} \frac{\tilde{\eta}_{t,b}\|\nabla_{\mathcal{G}_b} F(\theta_t)\|_2^2\|g_{t,\mathcal{G}_b}\|_2^2/d_b}{\sigma_{t,b}^2/d_b} + w_t \sigma_b C_3^2(1-\beta_t) \frac{\frac{a_t}{A_t}\|g_{t,\mathcal{G}_b}\|_2^2}{(\sqrt{\hat{v}_{t,b}}+\epsilon)^2}\right]
$$

$$
\leq \mathbf{E}\left[\frac{1-\beta_t}{4} \frac{\tilde{\eta}_{t,b}\|\nabla_{\mathcal{G}_b} F(\theta_t)\|_2^2 \mathbf{E}_t[\|g_{t,\mathcal{G}_b}\|_2]^2/d_b}{\sigma_{t,b}^2/d_b} + w_t \sigma_b C_3^2(1-\beta_t) \frac{\frac{a_t}{A_t}\|g_{t,\mathcal{G}_b}\|_2^2}{(\sqrt{\hat{v}_{t,b}}+\epsilon)^2}\right]
$$

$$
\leq \mathbf{E}\left[\frac{1-\beta_t}{4}\tilde{\eta}_{t,b}\|\nabla_{\mathcal{G}_b} F(\theta_t)\|_2^2 + w_t \sigma_b C_3^2 \frac{\frac{a_t}{A_t}\|g_{t,\mathcal{G}_b}\|_2^2}{(\sqrt{\hat{v}_{t,b}}+\epsilon)^2}\right], \tag{32}
$$

where the second inequality follows from $ab \leq \frac{a^2}{2c} + \frac{cb^2}{2}$ for any $c > 0$. Now, we estimate the third term of (26):

$$
\mathbf{E}\left[\left\langle \nabla_{\mathcal{G}_b} F(\theta_t), \tilde{\eta}_{t,b}\frac{\sigma_{t,b}}{\sqrt{d_b}}Y_{t,b}\right\rangle\right] \leq \mathbf{E}\left[\sqrt{\tilde{\eta}_{t,b}}\|\nabla_{\mathcal{G}_b}F(\theta_t)\|_2 \sqrt{\tilde{\eta}_{t,b}}\frac{\sigma_{t,b}}{\sqrt{d_b}}\|Y_{t,b}\|_2\right].
$$

Similarly, with (30) and (31), by expanding $\|Y_{t,b}\|_2$, we have

$$
\|Y_{t,b}\|_2 \leq \frac{\frac{a_t}{A_t\sqrt{d_b}}\|g_{t,\mathcal{G}_b}\|_2}{\sqrt{\hat{v}_{t,b}}+\epsilon} \frac{\beta_t\|m_{t-1,\mathcal{G}_b}\|_2}{\sqrt{A_{t-1}\hat{v}_{t-1,b}/A_t}+\epsilon} \frac{\sqrt{\frac{a_t}{A_t d_b}}\|g_{t,\mathcal{G}_b}\|_2}{\sqrt{\hat{v}_{t,b}}+\sqrt{A_{t-1}\hat{v}_{t-1,b}/A_t}} \frac{\sqrt{\frac{a_t}{A_t d_b}}\sigma_{t,b}}{\sqrt{\tilde{v}_{t,b}}+\sqrt{A_{t-1}\hat{v}_{t-1,b}/A_t}}
$$

$$
+\frac{\frac{a_t}{A_t}\|g_{t,\mathcal{G}_b}\|_2}{\sqrt{\hat{v}_{t,b}}+\epsilon}\frac{(1-\beta_t)\frac{\sigma_{t,b}}{\sqrt{d_b}}}{\sqrt{\tilde{v}_{t,b}}+\sqrt{\hat{v}_{t,b}}}
$$

$$
\leq \frac{\frac{a_t}{A_t\sqrt{d_b}}\|g_{t,\mathcal{G}_b}\|_2}{\sqrt{\hat{v}_{t,b}}+\epsilon}\frac{\beta_t}{\sqrt{C_a A_{t-1}/A_t a_t/(A_t d_b)(1-\rho)}}+\frac{\frac{a_t}{A_t}\|g_{t,\mathcal{G}_b}\|_2}{\sqrt{\hat{v}_{t,b}}+\epsilon}\frac{1-\beta_t}{\sqrt{a_t/A_t}}
$$

$$
= \frac{\sqrt{\frac{a_t}{A_t}}\|g_{t,\mathcal{G}_b}\|_2}{\sqrt{\hat{v}_{t,b}}+\epsilon}\frac{\beta_t}{\sqrt{C_a A_{t-1}/A_t(1-\rho)}}+\frac{\sqrt{\frac{a_t}{A_t}}\|g_{t,\mathcal{G}_b}\|_2}{\sqrt{\hat{v}_{t,b}}+\epsilon}(1-\beta_t)
$$

$$
= \frac{\sqrt{\frac{a_t}{A_t}}\|g_{t,\mathcal{G}_b}\|_2}{\sqrt{\hat{v}_{t,b}}+\epsilon}\left[\frac{\beta_t/(1-\beta_t)}{\sqrt{C_a A_{t-1}/A_t(1-\rho)}}+1\right](1-\beta_t)
$$

$$
\leq \frac{\sqrt{\frac{a_t}{A_t}}\|g_{t,\mathcal{G}_b}\|_2}{\sqrt{\hat{v}_{t,b}}+\epsilon}C_3(1-\beta_t),
$$

where $C_3$ is the constant defined above. Hence, together with (29), we obtain

$$
\mathbf{E}\left[\left\langle \nabla_{\mathcal{G}_b} F(\theta_t), \tilde{\eta}_{t,b}\frac{\sigma_{t,b}}{\sqrt{d_b}}Y_{t,b}\right\rangle\right] \leq \mathbf{E}\left[\sqrt{\tilde{\eta}_{t,b}}\|\nabla_{\mathcal{G}_b}F(\theta_t)\|_2 \sqrt{\tilde{\eta}_{t,b}}\frac{\sigma_{t,b}}{\sqrt{d_b}}\frac{\sqrt{\frac{a_t}{A_t}}\|g_{t,\mathcal{G}_b}\|_2}{\sqrt{\hat{v}_{t,b}}+\epsilon}C_3(1-\beta_t)\right]
$$

$$
\leq \frac{1-\beta_t}{4}\tilde{\eta}_{t,b}\mathbf{E}\left[\|\nabla_{\mathcal{G}_b}F(\theta_t)\|_2^2\right]+w_t\sigma_b C_3^2(1-\beta_t)\mathbf{E}\left[\frac{\frac{a_t}{A_t}\|g_{t,\mathcal{G}_b}\|_2^2}{(\sqrt{\hat{v}_{t,b}}+\epsilon)^2}\right]
$$

$$
\leq \frac{1-\beta_t}{4}\tilde{\eta}_{t,b}\mathbf{E}\left[\|\nabla_{\mathcal{G}_b}F(\theta_t)\|_2^2\right]+w_t\sigma_b C_3^2\mathbf{E}\left[\frac{\frac{a_t}{A_t}\|g_{t,\mathcal{G}_b}\|_2^2}{(\sqrt{\hat{v}_{t,b}}+\epsilon)^2}\right]. \tag{33}
$$

The last term of (26) can be bounded as follows

$$\mathbf{E}[\langle \nabla_{\mathcal{G}_b} F(\theta_t), Z_{t,b}\rangle] \leq \mathbf{E}[\|\nabla_{\mathcal{G}_b} F(\theta_t)\|_2 \|Z_{t,b}\|_2] \leq \mathbf{E}[\sigma_b \sqrt{d_b}\|Z_{t,b}\|_2],$$

and with (30), we get

$$
\begin{aligned}
\|Z_{t,b}\|_2 &\leq \frac{\beta_t \eta_t \|m_{t-1,\mathcal{G}_b}\|_2}{\sqrt{A_{t-1}\hat{v}_{t-1,b}/A_t} + \sqrt{A_{t-1}/A_t}\epsilon} \frac{\left(1 - \sqrt{A_{t-1}/A_t}\right)\epsilon}{\left(\sqrt{A_{t-1}\hat{v}_{t-1,b}/A_t} + \epsilon\right)} \\
&\leq \frac{\beta_t \eta_t}{\sqrt{C_a A_{t-1}/A_t a_t/(A_t d_b)(1-\rho)}} \left(1 - \sqrt{A_{t-1}/A_t}\right) \\
&\leq \frac{\beta \eta_t \sqrt{A_t d_b/a_t}}{\sqrt{C_a(1-\rho)}} \left(\sqrt{\frac{A_t}{A_{t-1}}} - 1\right) \\
&= \frac{\beta w_t \sqrt{d_b}}{\sqrt{C_a(1-\rho)}} \left(\sqrt{\frac{A_t}{A_{t-1}}} - 1\right).
\end{aligned}
$$

Hence,

$$\mathbf{E}[\langle \nabla_{\mathcal{G}_b} F(\theta_t), Z_{t,b}\rangle] \leq \frac{\beta \sigma_b d_b w_t}{\sqrt{C_a(1-\rho)}} \left(\sqrt{\frac{A_t}{A_{t-1}}} - 1\right). \tag{34}$$

Combining (26), (27), (32), (33), and (34), we get

$$
\mathbf{E}\left[\left\langle \nabla_{\mathcal{G}_b} F(\theta_t), \delta_{t,\mathcal{G}_b} - \frac{\beta_t \eta_t}{\sqrt{A_{t-1}/A_t}\eta_{t-1}}\delta_{t-1,\mathcal{G}_b}\right\rangle\right]
$$
$$
\leq -\frac{1-\beta_t}{2}\tilde{\eta}_{t,b}\mathbf{E}\left[\|\nabla_{\mathcal{G}_b} F(\theta_t)\|_2^2\right] + 2w_t\sigma_b C_3^2 \mathbf{E}\left[\frac{\frac{a_t}{A_t}\|g_{t,\mathcal{G}_b}\|_2^2}{(\sqrt{\hat{v}_{t,b}} + \epsilon)^2}\right] + \frac{\beta \sigma_b d_b w_t}{\sqrt{C_a(1-\rho)}}\left(\sqrt{\frac{A_t}{A_{t-1}}} - 1\right).
$$

Summing from $b = 1$ to $B$, we obtain

$$
\mathbf{E}\left[\left\langle \nabla F(\theta_t), \delta_t - \frac{\beta_t \eta_t}{\sqrt{A_{t-1}/A_t}\eta_{t-1}}\delta_{t-1}\right\rangle\right]
$$
$$
\leq -\frac{1-\beta_t}{2}\eta_t \mathbf{E}\left[\|\nabla F(\theta_t)\|_{\tilde{H}_t^{-1}}^2\right] + 2w_t C_3^2 \mathbf{E}\left[\frac{a_t}{A_t}\|H_t^{-1}g_t\|_{\Sigma^{1/2}}^2\right]
$$
$$
+ \frac{\beta w_t}{\sqrt{C_a(1-\rho)}}\left(\sqrt{\frac{A_t}{A_{t-1}}} - 1\right)\sum_{b=1}^{B}\sigma_b d_b.
$$

Then, with (25), we have

$$
\mathbf{E}[\langle \nabla F(\theta_t), \delta_t\rangle]
$$
$$
\leq \frac{\beta_t \eta_t}{\sqrt{A_{t-1}/A_t}\eta_{t-1}}M_{t-1} - \frac{1-\beta_t}{2}\eta_t \mathbf{E}\left[\|\nabla F(\theta_t)\|_{\tilde{H}_t^{-1}}^2\right] + 2w_t C_3^2 \mathbf{E}\left[\frac{a_t}{A_t}\|H_t^{-1}g_t\|_{\Sigma^{1/2}}^2\right]
$$
$$
+ \frac{\beta w_t}{\sqrt{C_a(1-\rho)}}\left(\sqrt{\frac{A_t}{A_{t-1}}} - 1\right)\sum_{b=1}^{B}\sigma_b d_b.
$$

We obtain (23) by adding the term $L\mathbf{E}[\|\delta_t\|_2^2]$ to both sides of the above equation. When $t = 1$, we have

$$
\begin{aligned}
M_1 &= \mathbf{E}[-\langle \nabla F(\theta_1), \eta_1 m_1/(\sqrt{\hat{v}_1} + \epsilon)\rangle + L\|\delta_1\|_2^2] \\
&= \mathbf{E}[-\langle \nabla F(\theta_1), \eta_1(1-\beta_1)g_1/(\sqrt{\hat{v}_1} + \epsilon)\rangle + L\|\delta_1\|_2^2]. \tag{35}
\end{aligned}
$$

Then, following the derivation of (19), for each block $b$, we have

$$
\begin{aligned}
\frac{(1-\beta_1)\eta_1 g_{1,\mathcal{G}_b}}{\sqrt{\hat{v}_{1,b}} + \epsilon} &= (1-\beta_1)\tilde{\eta}_{1,b}g_{1,\mathcal{G}_b} + \tilde{\eta}_{1,b}\frac{\sigma_{1,b}}{\sqrt{d_b}}\frac{\frac{a_1}{A_1}g_{1,\mathcal{G}_b}}{\sqrt{\hat{v}_{1,b}} + \epsilon}\frac{(1-\beta_1)\frac{\sigma_{1,b}}{\sqrt{d_b}}}{\sqrt{\tilde{v}_{1,b}} + \sqrt{\hat{v}_{1,b}}} \\
&\quad - \tilde{\eta}_{1,b}\frac{\frac{a_1}{A_1 d_b}\|g_{1,\mathcal{G}_b}\|_2^2}{\sqrt{\hat{v}_{1,b}} + \epsilon}\frac{(1-\beta_1)g_{1,\mathcal{G}_b}}{\sqrt{\tilde{v}_{1,b}} + \sqrt{\hat{v}_{1,b}}}.
\end{aligned}
$$

Hence, with similar argument, we get

$$\mathbf{E}\left[-\left\langle \nabla F(\theta_1), \eta_1 \frac{(1-\beta_1)g_1}{\sqrt{\hat{v}_1}+\epsilon} \right\rangle\right] \leq -\frac{1-\beta_1}{2}\eta_1 \mathbf{E}\left[\|\nabla F(\theta_1)\|^2_{\tilde{H}_1^{-1}}\right] + 2w_1 C_3^2 \mathbf{E}\left[\frac{a_1}{A_1}\|H_1^{-1}g_1\|^2_{\Sigma^{1/2}}\right].$$

Combining above with (35), and adding $L\mathbf{E}[\|\delta\|_2^2]$, we obtain (24). □

**Lemma 13.** *With the same assumptions in Lemma 12, we have*

$$\sum_{t=1}^{T}\|\delta_t\|_2^2 \leq \frac{C_2^2/C_1^2 w_1}{C_a(1-\sqrt{\rho})^2}\sum_{t=1}^{T}w_t \frac{a_t}{A_t}\|H_t^{-1}g_t\|_2^2.$$

*Proof.* For each block $b$,

$$\|m_{t,\mathcal{G}_b}\|_2 = \left\|\sum_{i=1}^{t}\left(\prod_{j=i+1}^{t}\beta_j\right)(1-\beta_i)g_{i,\mathcal{G}_b}\right\|_2$$

$$\leq \sum_{i=1}^{t}\left(\prod_{j=i+1}^{t}\beta_j(1-\beta_i)\right)\|g_{i,\mathcal{G}_b}\|_2$$

$$\leq \sum_{i=1}^{t}\beta^{t-i}\|g_{i,\mathcal{G}_b}\|_2.$$

Then,

$$\frac{\|m_{t,\mathcal{G}_b}\|_2}{\sqrt{\hat{v}_{t,b}}+\epsilon} \leq \sum_{i=1}^{t}\frac{\beta^{t-i}\|g_{i,\mathcal{G}_b}\|_2}{\sqrt{\hat{v}_{t,b}}+\epsilon}.$$

Since $\hat{v}_{t,b} \geq A_{t-1}\hat{v}_{t-1,b}/A_t$, we have $\hat{v}_{t,b} \geq \left(\prod_{j=i+1}^{t}A_{j-1}/A_j\right)\hat{v}_{i,b} = \hat{A}_{t,i}\hat{v}_{i,b} \geq C_a\tilde{p}^{t-i}\hat{v}_{i,b}$ by Lemma 10. It follows that

$$\frac{\|m_{t,\mathcal{G}_b}\|_2}{\sqrt{\hat{v}_{t,b}}+\epsilon} \leq \sum_{i=1}^{t}\frac{\beta^{t-i}\|g_{i,\mathcal{G}_b}\|_2}{\sqrt{\hat{v}_{t,b}}+\epsilon} \leq \frac{1}{\sqrt{C_a}}\sum_{i=1}^{t}\left(\frac{\beta}{\sqrt{\tilde{p}}}\right)^{t-i}\frac{\|g_{i,\mathcal{G}_b}\|_2}{\sqrt{\hat{v}_{i,b}}+\epsilon} = \frac{1}{\sqrt{C_a}}\sum_{i=1}^{t}\sqrt{\rho}^{t-i}\frac{\|g_{i,\mathcal{G}_b}\|_2}{\sqrt{\hat{v}_{i,b}}+\epsilon}.$$

Then, as $a_t/A_t = 1 - A_{t-1}/A_t$ is non-decreasing, we have

$$\|\delta_t\|_2^2 = \sum_{b=1}^{B}\left\|\frac{\eta_t m_{t,\mathcal{G}_b}}{\sqrt{\hat{v}_{t,b}}+\epsilon}\right\|_2^2 = \sum_{b=1}^{B}\left\|\frac{w_t\sqrt{a_t/A_t}m_{t,\mathcal{G}_b}}{\sqrt{\hat{v}_{t,b}}+\epsilon}\right\|_2^2 \leq \frac{w_t^2}{C_a}\sum_{b=1}^{B}\left(\sum_{i=1}^{t}\sqrt{\rho}^{t-i}\frac{\sqrt{a_t/A_t}\|g_{i,\mathcal{G}_b}\|_2}{\sqrt{\hat{v}_{i,b}}+\epsilon}\right)^2$$

$$\leq \frac{w_t^2}{C_a}\sum_{b=1}^{B}\left(\sum_{i=1}^{t}\sqrt{\rho}^{t-i}\frac{\sqrt{a_i/A_i}\|g_{i,\mathcal{G}_b}\|_2}{\sqrt{\hat{v}_{i,b}}+\epsilon}\right)^2$$

$$\leq \frac{w_t^2}{C_a}\sum_{b=1}^{B}\left(\sum_{j=1}^{t}\sqrt{\rho}^{t-j}\right)^2\left(\sum_{i=1}^{t}\frac{\sqrt{\rho}^{t-i}}{\left(\sum_{j=1}^{t}\sqrt{\rho}^{t-j}\right)}\frac{\sqrt{a_i/A_i}\|g_{i,\mathcal{G}_b}\|_2}{\sqrt{\hat{v}_{i,b}}+\epsilon}\right)^2$$

$$\leq \frac{w_t^2}{C_a}\sum_{b=1}^{B}\left(\sum_{j=1}^{t}\sqrt{\rho}^{t-j}\right)\sum_{i=1}^{t}\sqrt{\rho}^{t-i}\frac{a_i/A_i\|g_{i,\mathcal{G}_b}\|_2^2}{(\sqrt{\hat{v}_{i,b}}+\epsilon)^2} \leq \frac{w_t^2}{C_a(1-\sqrt{\rho})}\sum_{b=1}^{B}\sum_{i=1}^{t}\sqrt{\rho}^{t-i}\frac{a_i/A_i\|g_{i,\mathcal{G}_b}\|_2^2}{(\sqrt{\hat{v}_{i,b}}+\epsilon)^2}$$

$$= \frac{w_t^2}{C_a(1-\sqrt{\rho})}\sum_{i=1}^{t}\sqrt{\rho}^{t-i}\frac{a_i}{A_i}\|H_i^{-1}g_i\|_2^2.$$

As $w_t \leq C_2/C_1 w_i$ for any $i \leq t$ by Lemma 6, then we have

$$\|\delta_t\|_2^2 \leq \frac{C_2^2/C_1^2 w_1}{C_a(1-\sqrt{\rho})}\sum_{i=1}^{t}\sqrt{\rho}^{t-i}w_i\frac{a_i}{A_i}\|H_i^{-1}g_i\|_2^2.$$

Hence,

$$
\begin{aligned}
\sum_{t=1}^{T} \|\delta_t\|_2^2 &\leq \frac{C_2^2/C_1^2 w_1}{C_a(1-\sqrt{\rho})} \sum_{t=1}^{T}\sum_{i=1}^{t} \sqrt{\rho}^{t-i} w_i \frac{a_i}{A_i} \|H_i^{-1} g_i\|_2^2 \\
&= \frac{C_2^2/C_1^2 w_1}{C_a(1-\sqrt{\rho})} \sum_{i=1}^{T}\sum_{t=i}^{T} \sqrt{\rho}^{t-i} w_i \frac{a_i}{A_i} \|H_i^{-1} g_i\|_2^2 \\
&\leq \frac{C_2^2/C_1^2 w_1}{C_a(1-\sqrt{\rho})^2} \sum_{i=1}^{T} w_i \frac{a_i}{A_i} \|H_i^{-1} g_i\|_2^2.
\end{aligned}
$$

$\square$

**Lemma 14.** *With the same assumptions in Lemma 12, let* $M_t = \mathbf{E}[\langle \nabla F(\theta_t), \delta_t\rangle + L\|\delta_t\|_2^2]$, *we have*

$$
\begin{aligned}
\sum_{t=1}^{T} M_t &\leq \frac{C_2}{C_1\sqrt{C_a}(1-\sqrt{\rho})}\left[2C_3^2\sum_{t=1}^{T} w_t\mathbf{E}\left[\frac{a_t}{A_t}\|H_t^{-1} g_t\|_{\Sigma^{1/2}}^2\right] + \frac{LC_2^2/C_1^2 w_1}{C_a(1-\sqrt{\rho})^2}\sum_{t=1}^{T} w_t\mathbf{E}\left[\frac{a_t}{A_t}\|H_t^{-1} g_t\|_2^2\right]\right. \\
&\quad \left. + \frac{\beta}{\sqrt{C_a}(1-\rho)}\sum_{b=1}^{B}\sigma_b d_b \sum_{t=2}^{T} w_t\left(\sqrt{\frac{A_t}{A_{t-1}}}-1\right)\right] - \frac{1-\beta}{2}\sum_{t=1}^{T}\eta_t\mathbf{E}\left[\|\nabla F(\theta_t)\|_{\tilde{H}_t^{-1}}^2\right].
\end{aligned}
$$

*Proof.* Let define following quantity

$$
\begin{aligned}
N_t &= 2w_t C_3^2\mathbf{E}\left[\frac{a_t}{A_t}\|H_t^{-1} g_t\|_{\Sigma^{1/2}}^2\right] + L\mathbf{E}[\|\delta_t\|_2^2] + \frac{\beta w_t}{\sqrt{C_a}(1-\rho)}\left(\sqrt{\frac{A_t}{A_{t-1}}}-1\right)\sum_{b=1}^{B}\sigma_b d_b, \ \forall t \geq 2, \\
N_1 &= 2w_1 C_3^2\mathbf{E}\left[\frac{a_1}{A_1}\|H_1^{-1} g_1\|_{\Sigma^{1/2}}^2\right] + L\mathbf{E}[\|\delta_1\|_2^2].
\end{aligned}
$$

Then, by Lemma 12, for any $t \geq 2$, we have

$$
\begin{aligned}
M_t &\leq \frac{\beta_t\eta_t}{\sqrt{A_{t-1}/A_t}\eta_{t-1}}M_{t-1} - \frac{1-\beta_t}{2}\eta_t\mathbf{E}\left[\|\nabla F(\theta_t)\|_{\tilde{H}_t^{-1}}^2\right] + N_t \\
&\leq \frac{\beta_t\eta_t}{\sqrt{A_{t-1}/A_t}\eta_{t-1}}M_{t-1} + N_t
\end{aligned}
$$

and $M_1 \leq N_1$. Then, by recursively applying above relation, we get

$$
\begin{aligned}
M_t &\leq \frac{\hat{\beta}_{t,1}\eta_t}{\sqrt{\hat{A}_{t,1}}\eta_1}M_1 + \sum_{i=2}^{t}\frac{\hat{\beta}_{t,i}\eta_t}{\sqrt{\hat{A}_{t,i}}\eta_i}N_i - \frac{1-\beta_t}{2}\eta_t\mathbf{E}\left[\|\nabla F(\theta_t)\|_{\tilde{H}_t^{-1}}^2\right] \\
&\leq \sum_{i=1}^{t}\frac{\hat{\beta}_{t,i}\eta_t}{\sqrt{\hat{A}_{t,i}}\eta_i}N_i - \frac{1-\beta_t}{2}\eta_t\mathbf{E}\left[\|\nabla F(\theta_t)\|_{\tilde{H}_t^{-1}}^2\right],
\end{aligned}
$$

where $\hat{\beta}_{t,i} = \prod_{j=i+1}^{t}\beta_j$ for $i < t$ and $\hat{\beta}_{t,t} = 1$ and $\hat{A}_{t,i} = \prod_{j=i+1}^{t}\frac{A_{j-1}}{A_j}$ for $i < t$ and $\hat{A}_{t,t} = 1$. Note that $\hat{\beta}_{t,i} \leq \beta^{t-i}$, and $\eta_t \leq C_2/C_1\eta_i$. By Lemma 10, we have $\hat{A}_{t,i} \geq C_a\tilde{p}^{t-i}$ . Then,

$$
\begin{aligned}
M_t &\leq \frac{C_2}{C_1\sqrt{C_a}}\sum_{i=1}^{t}\left(\frac{\beta}{\sqrt{\tilde{p}}}\right)^{t-i} N_i - \frac{1-\beta_t}{2}\eta_t\mathbf{E}\left[\|\nabla F(\theta_t)\|_{\tilde{H}_t^{-1}}^2\right] \\
&= \frac{C_2}{C_1\sqrt{C_a}}\sum_{i=1}^{t}\sqrt{\rho}^{t-i} N_i - \frac{1-\beta_t}{2}\eta_t\mathbf{E}\left[\|\nabla F(\theta_t)\|_{\tilde{H}_t^{-1}}^2\right].
\end{aligned}
$$

It can be verified that the above inequality holds for $t = 1$ as $C_2/(C_1\sqrt{C_a}) \geq 1$. Then, summing from $t = 1$ to $t = T$, we obtain

$$
\begin{aligned}
\sum_{t=1}^{T} M_t &\leq \frac{C_2}{C_1\sqrt{C_a}} \sum_{t=1}^{T}\sum_{i=1}^{t} \sqrt{\rho}^{t-i} N_i - \sum_{t=1}^{T} \frac{1-\beta_t}{2} \eta_t \mathbf{E}\left[\|\nabla F(\theta_t)\|^2_{\tilde{H}_t^{-1}}\right] \\
&= \frac{C_2}{C_1\sqrt{C_a}} \sum_{i=1}^{T}\sum_{t=i}^{T} \sqrt{\rho}^{t-i} N_i - \sum_{t=1}^{T} \frac{1-\beta_t}{2} \eta_t \mathbf{E}\left[\|\nabla F(\theta_t)\|^2_{\tilde{H}_t^{-1}}\right] \\
&\leq \frac{C_2}{C_1\sqrt{C_a}(1-\sqrt{\rho})} \sum_{t=1}^{T} N_t - \frac{1-\beta}{2} \sum_{t=1}^{T} \eta_t \mathbf{E}\left[\|\nabla F(\theta_t)\|^2_{\tilde{H}_t^{-1}}\right]. \qquad (36)
\end{aligned}
$$

With Lemma 13, we get

$$
\begin{aligned}
\sum_{t=1}^{T} N_t &= \sum_{t=1}^{T}\left[2w_t C_3^2 \mathbf{E}\left[\frac{a_t}{A_t}\|H_t^{-1}g_t\|^2_{\Sigma^{1/2}}\right] + L\mathbf{E}[\|\delta_t\|^2_2]\right] + \sum_{t=2}^{T} \frac{\beta w_t}{\sqrt{C_a}(1-\rho)}\left(\sqrt{\frac{A_t}{A_{t-1}}}-1\right)\sum_{b=1}^{B}\sigma_b d_b \\
&\leq 2C_3^2 \sum_{t=1}^{T} w_t \mathbf{E}\left[\frac{a_t}{A_t}\|H_t^{-1}g_t\|^2_{\Sigma^{1/2}}\right] + \frac{LC_2^2/C_1^2 w_1}{C_a(1-\sqrt{\rho})^2}\sum_{t=1}^{T} w_t \mathbf{E}\left[\frac{a_t}{A_t}\|H_t^{-1}g_t\|^2_2\right] \\
&\quad + \frac{\beta}{\sqrt{C_a}(1-\rho)}\sum_{b=1}^{B}\sigma_b d_b \sum_{t=2}^{T} w_t\left(\sqrt{\frac{A_t}{A_{t-1}}}-1\right).
\end{aligned}
$$

Combining the above with (36), we obtain the result. $\qquad\square$

**Lemma 15.** *Assume $\{a_t\}$ is non-decreasing such that $\{A_{t-1}/A_t\}$ is non-decreasing. Define $w_t = \eta_t/\sqrt{\frac{a_t}{A_t}}$. Assume $w_t$ is "almost" non-increasing. This means there exists another non-increasing sequence $\{z_t\}$ and positive constants $C_1$ and $C_2$ such that $C_1 z_t \leq w_t \leq C_2 z_t$. We have*

$$
\frac{1}{T}\sum_{t=1}^{T}\left(\mathbf{E}[\|\nabla F(\theta_t)\|^{4/3}_2]\right)^{3/2} \leq \frac{\sqrt{2\left(\max_{1\leq t\leq T}\mathbf{E}\left[\max_b \tilde{v}_{t,b}\right]+\epsilon^2\right)}}{C_1/C_2 \eta_T T}\sum_{t=1}^{T}\eta_t\mathbf{E}\left[\|\nabla F(\theta_t)\|^2_{\tilde{H}_t^{-1}}\right].
$$

*Proof.* By Hölder's inequality, we have $\mathbf{E}[|XY|] \leq (\mathbf{E}[|X|^p])^{1/p}(\mathbf{E}[|Y|^q])^{1/q}$ for any $0 < p, q < 1$ with $1/p + 1/q = 1$. Taking $p = 3/2$, $q = 3$, and

$$
X = \left(\frac{\|\nabla F(\theta_t)\|^2_2}{\sqrt{\max_b \tilde{v}_{t,b}}+\epsilon}\right)^{2/3}, \quad Y = \left(\sqrt{\max_b \tilde{v}_{t,b}}+\epsilon\right)^{2/3},
$$

we obtain

$$
\mathbf{E}[\|\nabla F(\theta_t)\|^{4/3}_2] \leq \left(\mathbf{E}\left[\frac{\|\nabla F(\theta_t)\|^2_2}{\sqrt{\max_b \tilde{v}_{t,b}}+\epsilon}\right]\right)^{2/3}\left(\mathbf{E}\left[\left(\sqrt{\max_b \tilde{v}_{t,b}}+\epsilon\right)^2\right]\right)^{1/3}.
$$

Hence,

$$
\left(\mathbf{E}[\|\nabla F(\theta_t)\|^{4/3}_2]\right)^{3/2} \leq \left(\mathbf{E}\left[\frac{\|\nabla F(\theta_t)\|^2_2}{\sqrt{\max_b \tilde{v}_{t,b}}+\epsilon}\right]\right)\left(\mathbf{E}\left[\left(\sqrt{\max_b \tilde{v}_{t,b}}+\epsilon\right)^2\right]\right)^{1/2}.
$$

Note that

$$
\begin{aligned}
\frac{\|\nabla F(\theta_t)\|^2_2}{\sqrt{\max_b \tilde{v}_{t,b}}+\epsilon} &= \sum_{i=1}^{B}\frac{\|\nabla_{\mathcal{G}_i}F(\theta_t)\|^2_2}{\sqrt{\max_b \tilde{v}_{t,b}}+\epsilon} \\
&\leq \sum_{b=1}^{B}\frac{\|\nabla_{\mathcal{G}_b}F(\theta_t)\|^2_2}{\sqrt{\tilde{v}_{t,b}}+\epsilon} \\
&= \|\nabla F(\theta_t)\|^2_{\tilde{H}_t^{-1}}.
\end{aligned}
$$

We also have

$$
\begin{aligned}
\mathbf{E}\left[\left(\sqrt{\max_b \tilde{v}_{t,b}} + \epsilon\right)^2\right] &\leq 2\mathbf{E}\left[\left(\max_b \tilde{v}_{t,b} + \epsilon^2\right)\right] \\
&= 2\left(\mathbf{E}\left[\max_b \tilde{v}_{t,b}\right] + \epsilon^2\right).
\end{aligned}
$$

Then, for any $t \leq T$, we get

$$
\begin{aligned}
\left(\mathbf{E}[\|\nabla F(\theta_t)\|_2^{4/3}]\right)^{3/2} &\leq \sqrt{2\left(\mathbf{E}\left[\max_b \tilde{v}_{t,b}\right] + \epsilon^2\right)\mathbf{E}\left[\|\nabla F(\theta_t)\|_{\tilde{H}_t^{-1}}^2\right]} \\
&= \frac{\sqrt{2\left(\mathbf{E}\left[\max_b \tilde{v}_{t,b}\right] + \epsilon^2\right)}}{\eta_t}\eta_t\mathbf{E}\left[\|\nabla F(\theta_t)\|_{\tilde{H}_t^{-1}}^2\right] \\
&\leq \frac{\sqrt{2\left(\max_{1\leq t\leq T}\mathbf{E}\left[\max_b \tilde{v}_{t,b}\right] + \epsilon^2\right)}}{C_1/C_2\eta_T}\eta_t\mathbf{E}\left[\|\nabla F(\theta_t)\|_{\tilde{H}_t^{-1}}^2\right],
\end{aligned}
$$

where the last inequality follows from Lemma 6. Taking average from $t = 1$ to $T$, we get

$$
\frac{1}{T}\sum_{t=1}^{T}\left(\mathbf{E}[\|\nabla F(\theta_t)\|_2^{4/3}]\right)^{3/2} \leq \frac{\sqrt{2\left(\max_{1\leq t\leq T}\mathbf{E}\left[\max_b \tilde{v}_{t,b}\right] + \epsilon^2\right)}}{C_1/C_2\eta_T T}\sum_{t=1}^{T}\eta_t\mathbf{E}\left[\|\nabla F(\theta_t)\|_{\tilde{H}_t^{-1}}^2\right].
$$

$\square$

### F.1 PROOF OF THEOREM 2

*Proof.* As $F$ is $L$-smooth, then we have

$$
F(\theta_{t+1}) \leq F(\theta_t) + \langle\nabla F(\theta_t), \theta_{t+1} - \theta_t\rangle + \frac{L}{2}\|\theta_{t+1} - \theta_t\|^2.
$$

Recursively applying the above relation, we get

$$
F(\theta_*) \leq \mathbf{E}[F(\theta_{T+1})] \leq F(\theta_1) + \sum_{t=1}^{T} M_t,
$$

where $M_t = \mathbf{E}[\langle\nabla F(\theta_t), \delta_t\rangle + L\|\delta_t\|_2^2]$. By Lemma 14, we have

$$
\begin{aligned}
&\frac{1-\beta}{2}\sum_{t=1}^{T}\eta_t\mathbf{E}\left[\|\nabla F(\theta_t)\|_{\tilde{H}_t^{-1}}^2\right] \\
&\leq F(\theta_1) - F(\theta_*) + \frac{C_2}{C_1\sqrt{C_a}(1-\sqrt{\rho})}\left[2C_3^2\sum_{t=1}^{T}w_t\mathbf{E}\left[\frac{a_t}{A_t}\|H_t^{-1}g_t\|_{\Sigma^{1/2}}^2\right]\right. \\
&\quad\left. + \frac{LC_2^2/C_1^2 w_1}{C_a(1-\sqrt{\rho})^2}\sum_{t=1}^{T}w_t\mathbf{E}\left[\frac{a_t}{A_t}\|H_t^{-1}g_t\|_2^2\right] + \frac{\beta}{\sqrt{C_a}(1-\rho)}\sum_{b=1}^{B}\sigma_b d_b\sum_{t=2}^{T}w_t\left(\sqrt{\frac{A_t}{A_{t-1}}} - 1\right)\right] \\
&= F(\theta_1) - F(\theta_*) + \frac{C_2}{C_1\sqrt{C_a}(1-\sqrt{\rho})}\left[2C_3^2\sum_{t=1}^{T}\eta_t\mathbf{E}\left[\sqrt{\frac{a_t}{A_t}}\|H_t^{-1}g_t\|_{\Sigma^{1/2}}^2\right]\right. \\
&\quad\left. + \frac{LC_2^2/C_1^2 w_1}{C_a(1-\sqrt{\rho})^2}\sum_{t=1}^{T}\eta_t\mathbf{E}\left[\sqrt{\frac{a_t}{A_t}}\|H_t^{-1}g_t\|_2^2\right] + \frac{\beta}{\sqrt{C_a}(1-\rho)}\sum_{b=1}^{B}\sigma_b d_b\sum_{t=2}^{T}w_t\left(\sqrt{\frac{A_t}{A_{t-1}}} - 1\right)\right].
\end{aligned}
$$

Applying Lemma 8, we have

$$
\begin{aligned}
\frac{1-\beta}{2} & \sum_{t=1}^{T} \eta_t \mathbf{E}\left[\|\nabla F(\theta_t)\|_{\hat{H}_t^{-1}}^2\right] \\
\leq\ & F(\theta_1) - F(\theta_*) \\
& + \frac{C_2}{C_1\sqrt{C_a}(1-\sqrt{\rho})}\left[2C_3^2\frac{C_2}{C_1}\left[w_1\sum_{b=1}^{B}\sigma_b d_b \log\left(\frac{\sigma_b^2}{\epsilon^2}+1\right) + \sum_{b=1}^{B}\sigma_b d_b \sum_{t=1}^{T}\eta_t\sqrt{\frac{a_t}{A_t}}\frac{A_t}{A_{t-1}+a_1}\right] \right. \\
& + \frac{LC_2^3/C_1^3 w_1}{C_a(1-\sqrt{\rho})^2}\left[w_1\sum_{b=1}^{B}d_b \log\left(\frac{\sigma_b^2}{\epsilon^2}+1\right) + \sum_{b=1}^{B}d_b \sum_{t=1}^{T}\eta_t\sqrt{\frac{a_t}{A_t}}\frac{A_t}{A_{t-1}+a_1}\right] \\
& \left. + \frac{\beta}{\sqrt{C_a}(1-\rho)}\sum_{b=1}^{B}\sigma_b d_b \sum_{t=2}^{T}w_t\left(\sqrt{\frac{A_t}{A_{t-1}}}-1\right)\right] \\
\leq\ & F(\theta_1) - F(\theta_*) \\
& + \frac{C_2}{C_1\sqrt{C_a}(1-\sqrt{\rho})}\left[2C_3^2\frac{C_2}{C_1}\left[w_1\sum_{b=1}^{B}\sigma_b d_b \log\left(\frac{\sigma_b^2}{\epsilon^2}+1\right) + \omega\sum_{b=1}^{B}\sigma_b d_b \sum_{t=1}^{T}\eta_t\sqrt{\frac{a_t}{A_t}}\right] \right. \\
& + \frac{LC_2^3/C_1^3 w_1}{C_a(1-\sqrt{\rho})^2}\left[w_1\sum_{b=1}^{B}d_b \log\left(\frac{\sigma_b^2}{\epsilon^2}+1\right) + \omega\sum_{b=1}^{B}d_b \sum_{t=1}^{T}\eta_t\sqrt{\frac{a_t}{A_t}}\right] \\
& \left. + \frac{\beta}{\sqrt{C_a}(1-\rho)}\sum_{b=1}^{B}\sigma_b d_b \sum_{t=2}^{T}w_t\left(\sqrt{\frac{A_t}{A_{t-1}}}-1\right)\right] \\
=\ & F(\theta_1) - F(\theta_*) + \frac{C_2}{C_1\sqrt{C_a}(1-\sqrt{\rho})}\left[\frac{\beta}{\sqrt{C_a}(1-\rho)}\sum_{b=1}^{B}\sigma_b d_b \sum_{t=2}^{T}w_t\left(\sqrt{\frac{A_t}{A_{t-1}}}-1\right)\right. \\
& \left. + \sum_{b=1}^{B}\left[\frac{LC_3^3/C_1^3 w_1 d_b}{C_a(1-\sqrt{\rho})^2} + \frac{2C_3^2 C_2 \sigma_b d_b}{C_1}\right]\left[w_1\log\left(\frac{\sigma_b^2}{\epsilon^2}+1\right) + \omega\sum_{t=1}^{T}\eta_t\sqrt{\frac{a_t}{A_t}}\right]\right].
\end{aligned}
$$

Combining above with Lemma 15, we have

$$
\begin{aligned}
\min_{1\leq t\leq T}& \left(\mathbf{E}[\|\nabla F(\theta_t)\|_2^{4/3}]\right)^{3/2} \leq \frac{1}{T}\sum_{t=1}^{T}\left(\mathbf{E}[\|\nabla F(\theta_t)\|_2^{4/3}]\right)^{3/2} \\
\leq\ & \frac{2\sqrt{2\left(\max_{1\leq t\leq T}\mathbf{E}\left[\max_b \tilde{v}_{t,b}\right]+\epsilon^2\right)}}{C_1/C_2(1-\beta)\eta_T T}\left[F(\theta_1)-F(\theta_*)\right. \\
& + \frac{C_2}{C_1\sqrt{C_a}(1-\sqrt{\rho})}\left[\frac{\beta}{\sqrt{C_a}(1-\rho)}\sum_{b=1}^{B}\sigma_b d_b \sum_{t=2}^{T}w_t\left(\sqrt{\frac{A_t}{A_{t-1}}}-1\right)\right. \\
& \left.\left. + \sum_{b=1}^{B}\left[\frac{LC_2^3/C_1^3 w_1 d_b}{C_a(1-\sqrt{\rho})^2} + \frac{2C_3^2 C_2 \sigma_b d_b}{C_1}\right]\left[w_1\log\left(\frac{\sigma_b^2}{\epsilon^2}+1\right) + \omega\sum_{t=1}^{T}\eta_t\sqrt{\frac{a_t}{A_t}}\right]\right]\right] \\
=\ & \frac{\sqrt{2\left(\max_{1\leq t\leq T}\mathbf{E}\left[\max_b \tilde{v}_{t,b}\right]+\epsilon^2\right)}}{\eta_T T}\left[\frac{2C_2}{(1-\beta)C_1}\left[F(\theta_1)-F(\theta_*)\right]\right. \\
& + \frac{2C_2^2}{C_1^2\sqrt{C_a}(1-\sqrt{\rho})(1-\beta)}\left[\frac{\beta}{\sqrt{C_a}(1-\rho)}\sum_{b=1}^{B}\sigma_b d_b \sum_{t=2}^{T}w_t\left(\sqrt{\frac{A_t}{A_{t-1}}}-1\right)\right. \quad\quad (37) \\
& \left.\left. + \sum_{b=1}^{B}\left[\frac{LC_2^3 w_1 d_b}{C_1^3 C_a(1-\sqrt{\rho})^2} + \frac{2C_3^2 C_2 \sigma_b d_b}{C_1}\right]\left[w_1\log\left(\frac{\sigma_b^2}{\epsilon^2}+1\right) + \omega\sum_{t=1}^{T}\eta_t\sqrt{\frac{a_t}{A_t}}\right]\right]\right] \quad (38)
\end{aligned}
$$

□

### F.2 PROOF OF PROPOSITION 3

To prove the result, we use the following high probability bound.

**Proposition 5.** *With probability at least* $1 - \delta^{2/3}$, $\min_{1 \le t \le T} \|\nabla F(\theta_t)\|_2^2 \le C(T)/\delta$.

*Proof.* By the concavity of the minimum, we have

$$\mathbf{E}\left[\min_{1 \le t \le T} \|\nabla F(\theta_t)\|_2^{4/3}\right]^{3/2} \le \min_{1 \le t \le T} \left(\mathbf{E}[\|\nabla F(\theta_t)\|_2^{4/3}]\right)^{3/2}.$$

Let $X = \min_{1 \le t \le T} \|\nabla F(\theta_t)\|_2^2$. The Theorem 2 suggests that we have $\mathbf{E}[X^{2/3}] \le C(T)^{2/3}$. By Markov's inequality, we get

$$P\left(X^{2/3} > \frac{C(T)^{2/3}}{\delta^{2/3}}\right) \le \frac{\mathbf{E}[X^{2/3}]}{C(T)^{2/3}}\delta^{2/3} \le \delta^{2/3}.$$

Hence, $P\left(X > \frac{C(T)}{\delta}\right) \le \delta^{2/3}$, and we have $P(X \le \frac{C(T)}{\delta}) \ge 1 - \delta^{2/3}$. $\qquad\square$

*Proof.* (of Proposition 3) Recall the definition of $C(T)$ in (17). When $a_t = at^\tau$, we have $A_t = \mathcal{O}(t^{1+\tau})$. This suggests that

$$\eta_t \sqrt{\frac{a_t}{A_t}} \le \frac{\eta}{1-\beta}\sqrt{\frac{a_t}{tA_t}} = \mathcal{O}\left(\frac{1}{t}\right),$$

$$w_t = \frac{\eta}{1-\tilde{\beta}_t}\sqrt{\frac{A_t}{ta_t}} \le \frac{\eta}{1-\beta}\sqrt{\frac{A_t}{ta_t}} = \mathcal{O}(1),$$

and

$$\sqrt{\frac{A_t}{A_{t-1}}} - 1 = \frac{\sqrt{A_t} - \sqrt{A_{t-1}}}{\sqrt{A_{t-1}}} = \mathcal{O}\left(\frac{1}{t}\right).$$

Hence,

$$\sum_{t=1}^{T} \eta_t \sqrt{\frac{a_t}{A_t}} = \mathcal{O}(\log(T)), \quad \sum_{t=1}^{T} w_t \left(\sqrt{\frac{A_t}{A_{t-1}}} - 1\right) = \mathcal{O}(\log(T)), \quad \text{and} \quad C(T) = \mathcal{O}\left(\frac{\log(T)}{\sqrt{T}}\right).$$

On the other hand, when $a_t = \alpha^{-t}$, we have

$$\eta_t \sqrt{\frac{a_t}{A_t}} \le \frac{\eta}{1-\beta}\sqrt{\frac{1-\alpha}{(1-\alpha^t)t}} \le \frac{\eta}{\sqrt{t}},$$

$$w_t = \frac{\eta}{1-\tilde{\beta}_t}\sqrt{\frac{A_t}{a_t t}} = \frac{\eta}{1-\tilde{\beta}_t}\sqrt{\frac{1-\alpha^t}{(1-\alpha)t}} \le \frac{\eta}{(1-\beta)\sqrt{(1-\alpha)t}},$$

and

$$\sqrt{\frac{A_t}{A_{t-1}}} - 1 = \sqrt{\frac{1-\alpha^t}{(1-\alpha^{t-1})\alpha}} \le \sqrt{\frac{1+\alpha}{\alpha}}.$$

Then, we get

$$\sum_{t=1}^{T} \eta_t \sqrt{\frac{a_t}{A_t}} \le 2\eta\sqrt{T}, \quad \sum_{t=1}^{T} w_t \left(\sqrt{\frac{A_t}{A_{t-1}}} - 1\right) = \mathcal{O}\left(\sqrt{T}\right), \quad \text{and} \quad C(T) = \mathcal{O}(1).$$

Combining the results with Proposition 5, we complete the proof. $\qquad\square$

### F.3 PROOF OF COROLLARY 2

*Proof.* As $\|g_{t,\mathcal{G}_b}\|_2^2/d_b \leq G_b^2$, then we have

$$\tilde{v}_{t,b} = (v_{t-1,b} + a_t \mathbf{E}_t[\|g_{t,\tilde{\mathcal{G}}_b}\|_2^2]/d_b)/A_t \leq G_b^2,$$

and therefore $\bar{v}_{T,B} \equiv \max_{1\leq t\leq T} \mathbf{E}\left[\max_b \tilde{v}_{t,b}\right] \leq \max_b G_b^2$. Arranging the terms in $\tilde{C}(T)$, we obtain

$$
\begin{aligned}
&\tilde{C}(T) \\
&= \frac{\sqrt{2\left(\max_b G_b^2 + \epsilon^2\right)}}{\eta_T T}\Bigg[\frac{2C_2}{(1-\beta)C_1}[F(\theta_1) - F(\theta_*)] \\
&\quad + \frac{2C_2^2}{C_1^2\sqrt{C_a}(1-\sqrt{\rho})(1-\beta)}\Bigg[\Bigg[\frac{\beta}{\sqrt{C_a}(1-\rho)}\sum_{t=2}^{T} w_t\left(\sqrt{\frac{A_t}{A_{t-1}}} - 1\right) + \frac{2C_3^2 C_2}{C_1}\omega\sum_{t=1}^{T}\eta_t\sqrt{\frac{a_t}{A_t}}\Bigg]\sum_{b=1}^{B}\sigma_b d_b \\
&\quad + \frac{LC_2^3 w_1^2}{C_1^3 C_a(1-\sqrt{\rho})^2}\sum_{b=1}^{B} d_b\log\left(\frac{\sigma_b^2}{\epsilon^2} + 1\right) + \frac{2C_3^2 C_2 w_1}{C_1}\sum_{b=1}^{B}\sigma_b d_b\log\left(\frac{\sigma_b^2}{\epsilon^2} + 1\right) \\
&\quad + \frac{LC_2^3 w_1 d\omega}{C_1^3 C_a(1-\sqrt{\rho})^2}\sum_{t=1}^{T}\eta_t\sqrt{\frac{a_t}{A_t}}\Bigg]\Bigg].
\end{aligned}
$$

When $B = d$, we have

$$
\begin{aligned}
&\tilde{C}_d(T) \\
&= \frac{\sqrt{2\left(\max_b \max_{i\in\tilde{\mathcal{G}}_b} G_i^2 + \epsilon^2\right)}}{\eta_T T}\Bigg[\frac{2C_2}{(1-\beta)C_1}[F(\theta_1) - F(\theta_*)] \\
&\quad + \frac{2C_2^2}{C_1^2\sqrt{C_a}(1-\sqrt{\rho})(1-\beta)}\Bigg[\Bigg[\frac{\beta}{\sqrt{C_a}(1-\rho)}\sum_{t=2}^{T} w_t\left(\sqrt{\frac{A_t}{A_{t-1}}} - 1\right) + \frac{2C_3^2 C_2}{C_1}\omega\sum_{t=1}^{T}\eta_t\sqrt{\frac{a_t}{A_t}}\Bigg]\sum_{i=1}^{d}\sigma_i \\
&\quad + \frac{LC_2^3 w_1^2}{C_1^3 C_a(1-\sqrt{\rho})^2}\sum_{i=1}^{d}\log\left(\frac{\sigma_i^2}{\epsilon^2} + 1\right) + \frac{2C_3^2 C_2 w_1}{C_1}\sum_{i=1}^{d}\sigma_i\log\left(\frac{\sigma_i^2}{\epsilon^2} + 1\right) \\
&\quad + \frac{LC_2^3 w_1 d\omega}{C_1^3 C_a(1-\sqrt{\rho})^2}\sum_{t=1}^{T}\eta_t\sqrt{\frac{a_t}{A_t}}\Bigg]\Bigg] \\
&= \frac{\sqrt{2\left(\max_b \max_{i\in\tilde{\mathcal{G}}_b} G_i^2 + \epsilon^2\right)}}{\eta_T T}\Bigg[\frac{2C_2}{(1-\beta)C_1}[F(\theta_1) - F(\theta_*)] \\
&\quad + \frac{2C_2^2}{C_1^2\sqrt{C_a}(1-\sqrt{\rho})(1-\beta)}\Bigg[\Bigg[\frac{\beta}{\sqrt{C_a}(1-\rho)}\sum_{t=2}^{T} w_t\left(\sqrt{\frac{A_t}{A_{t-1}}} - 1\right) + \frac{2C_3^2 C_2}{C_1}\omega\sum_{t=1}^{T}\eta_t\sqrt{\frac{a_t}{A_t}}\Bigg]\sum_{b=1}^{\tilde{B}}\sum_{i\in\tilde{\mathcal{G}}_b}\sigma_i \\
&\quad + \frac{LC_2^3 w_1^2}{C_1^3 C_a(1-\sqrt{\rho})^2}\sum_{b=1}^{\tilde{B}}\sum_{i\in\tilde{\mathcal{G}}_b}\log\left(\frac{\sigma_i^2}{\epsilon^2} + 1\right) + \frac{2C_3^2 C_2 w_1}{C_1}\sum_{b=1}^{\tilde{B}}\sum_{i\in\tilde{\mathcal{G}}_b}\sigma_i\log\left(\frac{\sigma_i^2}{\epsilon^2} + 1\right) \\
&\quad + \frac{LC_2^3 w_1 d\omega}{C_1^3 C_a(1-\sqrt{\rho})^2}\sum_{t=1}^{T}\eta_t\sqrt{\frac{a_t}{A_t}}\Bigg]\Bigg].
\end{aligned}
$$

Substituting $\quad r_1 \quad := \quad \dfrac{\sum_{b=1}^{\tilde{B}} \sum_{i \in \tilde{\mathcal{G}}_b} \log\left(\sigma_i^2/\epsilon^2+1\right)}{\sum_{b=1}^{\tilde{B}} d_b \log\left(\sigma_b^2/\epsilon^2+1\right)}, \quad r_2 \quad := \quad \dfrac{\sum_{b=1}^{\tilde{B}} \sum_{i \in \tilde{\mathcal{G}}_b} \sigma_i}{\sum_{b=1}^{\tilde{B}} \sigma_b d_b} \quad$ and $\quad r_3 \quad :=$

$\dfrac{\sum_{b=1}^{\tilde{B}} \sum_{i \in \tilde{\mathcal{G}}_b} \sigma_i \log\left(\sigma_i^2/\epsilon^2+1\right)}{\sum_{b=1}^{\tilde{B}} \sigma_b d_b \log\left(\sigma_b^2/\epsilon^2+1\right)}$, we get

$$
\begin{aligned}
&\tilde{C}_d(T) \\
&= \frac{\sqrt{2\left(\max_b \max_{i \in \tilde{\mathcal{G}}_b} G_i^2 + \epsilon^2\right)}}{\eta_T T} \left[ \frac{2C_2}{(1-\beta)C_1}[F(\theta_1) - F(\theta_*)] \right. \\
&\quad + \frac{2C_2^2}{C_1^2 \sqrt{C_a}(1-\sqrt{\rho})(1-\beta)} \left[ \left[ \frac{\beta}{\sqrt{C_a(1-\rho)}} \sum_{t=2}^{T} w_t \left( \sqrt{\frac{A_t}{A_{t-1}}} - 1 \right) + \frac{2C_3^2 C_2}{C_1} \omega \sum_{t=1}^{T} \eta_t \sqrt{\frac{a_t}{A_t}} \right] r_2 \sum_{b=1}^{\tilde{B}} \sigma_b d_b \right. \\
&\quad + \frac{LC_2^3 w_1^2}{C_1^3 C_a (1-\sqrt{\rho})^2} r_1 \sum_{b=1}^{\tilde{B}} d_b \log\left(\frac{\sigma_b^2}{\epsilon^2} + 1\right) + \frac{2C_3^2 C_2 w_1}{C_1} r_3 \sum_{b=1}^{\tilde{B}} \sigma_b d_b \log\left(\frac{\sigma_b^2}{\epsilon^2} + 1\right) \\
&\quad \left. \left. + \frac{LC_2^3 w_1 d\omega}{C_1^3 C_a (1-\sqrt{\rho})^2} \sum_{t=1}^{T} \eta_t \sqrt{\frac{a_t}{A_t}} \right] \right] \\
&\geq \min(1, r_{\min}) \sqrt{\frac{\max_b \max_{i \in \tilde{\mathcal{G}}_b} G_i^2 + \epsilon^2}{\max_b G_b^2 + \epsilon^2}} C_{\tilde{B}}(T),
\end{aligned}
$$

where $r_{\min} = \min(r_1, r_2, r_3)$. Then, $\frac{\tilde{C}_d(T)}{\tilde{C}_{\tilde{B}}(T)} \geq \min(1, r_{\min}) \sqrt{\frac{\max_b \max_{i \in \tilde{\mathcal{G}}_b} G_i^2 + \epsilon^2}{\max_b G_b^2 + \epsilon^2}}$. When $B = \tilde{B}$, we assume that Assumption 4 is tight in the sense that $\sigma_b^2 \leq \frac{1}{d_b} \sum_{i \in \tilde{\mathcal{G}}_b} \sigma_i^2$,[5] Thus, $r_{\min}$ can be larger than 1 as $\sigma_b^2 \leq \frac{1}{d_b} \sum_{i \in \tilde{\mathcal{G}}_b} \sigma_i^2$. Corollary 2 then indicates that blockwise adaptive stepsize will lead to improvement if $\sqrt{(\max_b \max_{i \in \tilde{\mathcal{G}}_b} G_i^2 + \epsilon^2)/(\max_b G_b^2 + \epsilon^2)} > \frac{1}{r_{\min}}$. Similarly, assume that the upper bound $G_b$ is tight so that $G_b^2 \leq \frac{1}{d_b} \sum_{i \in \tilde{\mathcal{G}}_b} G_i^2$. Thus, $\max_b \max_{i \in \tilde{\mathcal{G}}_b} G_i^2 \geq \max_b G_b^2$, and the above condition is likely to hold when $r_{\min}$ is close to 1 or is larger than 1. From the definitions of $r_1$, $r_2$ and $r_3$, we can see that they get close to 1 or is larger than 1 when $\{\sigma_i^2\}_{i \in \tilde{\mathcal{G}}_b}$ have sufficiently low variability. $\qquad \square$

### F.4 PROOF OF PROPOSITION 4

*Proof.* As function is $\tilde{\gamma}$-Lipschitz, we have the following result:

$$
\sup_z \mathbf{E}_M[f(M(S); z) - f(M(S'); z)] \leq \tilde{\gamma} \mathbf{E}_M[\|M(S) - M(S')\|_2].
$$

Therefore, we can consider bounding $\mathbf{E}_M[\|M(S) - M(S')\|_2]$. Let $\beta_t = 0$ for all $t$.

$$
\begin{aligned}
\theta_{t+1} &= \theta_1 - \sum_{k=1}^{t} \eta_k H_k^{-1} m_k \\
&= \theta_1 - \sum_{k=1}^{t} \eta_k H_k^{-1} g_k \\
&= \theta_1 - \sum_{k=1}^{t} \eta_k H_k^{-1} \nabla f(\theta_k; z_{i_k}),
\end{aligned}
$$

---

[5]Note that $\frac{1}{d_b} \mathbf{E}_t[\|g_{t,\tilde{\mathcal{G}}_b}\|_2^2] = \frac{1}{d_b} \sum_{i \in \tilde{\mathcal{G}}_b} \mathbf{E}_t[g_{t,i}^2] \leq \frac{1}{d_b} \sum_{i \in \tilde{\mathcal{G}}_b} \sigma_i^2$. On the other hand, $\frac{1}{d_b} \mathbf{E}_t[\|g_{t,\tilde{\mathcal{G}}_b}\|_2^2] \leq \sigma_b^2$. Thus, this bound is tight in the sense that $\sigma_b^2 \leq \frac{1}{d_b} \sum_{i \in \tilde{\mathcal{G}}_b} \sigma_i^2$.

where $i_k \in [n]$ is the example index selected at iteration $k$. Then, we can bound $\Delta_{t+1} = \|\theta_{t+1} - \theta'_{t+1}\|_2$ as follows

$$
\begin{aligned}
\mathbf{E}[\Delta_{t+1}] &= \mathbf{E}[\|\theta_{t+1} - \theta'_{t+1}\|_2] \\
&= \mathbf{E}[\|\theta_1 - \theta'_1 - \sum_{k=1}^{t} \eta_k H_k^{-1} \nabla f(\theta_k; z_{i_k}) + \sum_{k=1}^{t} \eta_k H_k^{'-1} \nabla f(\theta'_k; z'_{i_k})\|_2] \\
&\leq \mathbf{E}[\|\theta_1 - \theta'_1\|_2] + \sum_{k=1}^{t} \eta_k \mathbf{E}[\|H_k^{-1} \nabla f(\theta_k; z_{i_k}) - H_k^{'-1} \nabla f(\theta'_k; z'_{i_k})\|_2] \\
&= \sum_{k=1}^{t} \eta_k \mathbf{E}[\|H_k^{-1} \nabla f(\theta_k; z_{i_k}) - H_k^{'-1} \nabla f(\theta'_k; z'_{i_k})\|_2]. \tag{39}
\end{aligned}
$$

Note that $z_{i_k} = z'_{i_k}$ with probability $1 - 1/n$. Then, we can bound each term $\mathbf{E}[\|H_k^{-1} \nabla f(\theta_k; z_{i_k}) - H_k^{'-1} \nabla f(\theta'_k; z'_{i_k})\|_2]$ as follows

$$
\begin{aligned}
&\mathbf{E}[\|H_k^{-1} \nabla f(\theta_k; z_{i_k}) - H_k^{'-1} \nabla f(\theta'_k; z'_{i_k})\|_2] \\
&\leq \frac{2}{n} \mathbf{E}[\|H_k^{-1} \nabla f(\theta_k; z_{i_k})\|_2] + \left(1 - \frac{1}{n}\right) \mathbf{E}[\|H_k^{-1} \nabla f(\theta_k; z_{i_k}) - H_k^{'-1} \nabla f(\theta'_k; z_{i_k})\|_2] \\
&\leq \frac{2}{n} \mathbf{E}[\|H_k^{-1} \nabla f(\theta_k; z_{i_k})\|_2] + \left(1 - \frac{1}{n}\right) \mathbf{E}[\|H_k^{-1} \nabla f(\theta_k; z_{i_k}) - H_k^{'-1} \nabla f(\theta_k; z_{i_k})\|_2] \\
&\quad + \left(1 - \frac{1}{n}\right) \mathbf{E}[\|H_k^{'-1} \nabla f(\theta_k; z_{i_k}) - H_k^{'-1} \nabla f(\theta'_k; z_{i_k})\|_2]. \tag{40}
\end{aligned}
$$

The second term is bounded as

$$
\begin{aligned}
&\mathbf{E}[\|H_k^{-1} \nabla f(\theta_k; z_{i_k}) - H_k^{'-1} \nabla f(\theta_k; z_{i_k})\|_2] \\
&\leq \mathbf{E}[\|H_k^{-1} - H_k^{'-1}\|_2 \|\nabla f(\theta_k; z_{i_k})\|_2] \\
&\leq \tilde{\gamma} \mathbf{E}[\|H_k^{-1} - H_k^{'-1}\|_2] \\
&= \tilde{\gamma} \mathbf{E}\left[\max_b \left|\frac{1}{\sqrt{\hat{v}_{k,b}} + \epsilon} - \frac{1}{\sqrt{\hat{v}'_{k,b}} + \epsilon}\right|\right].
\end{aligned}
$$

We expand the third term of (40) as

$$
\begin{aligned}
&\mathbf{E}[\|H_k^{'-1} \nabla f(\theta_k; z_{i_k}) - H_k^{'-1} \nabla f(\theta'_k; z_{i_k})\|_2] \\
&\leq \mathbf{E}[\|H_k^{'-1}\|_2 \|\nabla f(\theta_k; z_{i_k}) - \nabla f(\theta'_k; z_{i_k})\|_2] \\
&\leq L \mathbf{E}[\|H_k^{'-1}\|_2 \|\theta_k - \theta'_k\|_2] \\
&\leq L \mathbf{E}\left[\frac{1}{\sqrt{\min_b \hat{v}_{k,b}} + \epsilon} \|\theta_k - \theta'_k\|_2\right] \\
&= L \mathbf{E}\left[\frac{1}{\sqrt{\min_b \hat{v}_{k,b}} + \epsilon} \Delta_k\right].
\end{aligned}
$$

Substituting the above results into (40) and combining with (39), we obtain

$$
\begin{aligned}
\mathbf{E}[\Delta_{t+1}] &\leq \frac{2}{n} \sum_{k=1}^{t} \eta_k \mathbf{E}[\|H_k^{-1} \nabla f(\theta_k; z_{i_k})\|_2] \\
&\quad + \left(1 - \frac{1}{n}\right) \tilde{\gamma} \sum_{k=1}^{t} \eta_k \mathbf{E}\left[\max_b \left|\frac{1}{\sqrt{\hat{v}_{k,b}} + \epsilon} - \frac{1}{\sqrt{\hat{v}'_{k,b}} + \epsilon}\right|\right] \\
&\quad + \left(1 - \frac{1}{n}\right) L \sum_{k=1}^{t} \eta_k \mathbf{E}\left[\frac{1}{\sqrt{\min_b \hat{v}_{k,b}} + \epsilon} \Delta_k\right].
\end{aligned}
$$

Note that if $w_t = \eta_t/\sqrt{a_t/A_t}$ is "almost" non-increasing w.r.t. another non-increasing sequence $\{z_t\}$ and positive constant $C_1$ and $C_2$, then $w_t^2$ is also "almost" non-increasing w.r.t. another non-increasing sequence $\{z_t^2\}$ and positive constant $C_1^2$ and $C_2^2$. Using Lemma 8 with $C = I$, we have

$$\sum_{k=1}^{t} \eta_k \mathbf{E}[\|H_k^{-1} \nabla f(\theta_k; z_{i_k})\|_2]$$

$$\leq \quad \sqrt{t} \sqrt{\sum_{k=1}^{t} \eta_k^2 \mathbf{E}[\|H_k^{-1} \nabla f(\theta_k; z_{i_k})\|_2^2]}$$

$$= \quad \sqrt{t} \sqrt{\sum_{k=1}^{t} \eta_k^2 \sqrt{\frac{A_k}{a_k}} \mathbf{E}\left[\sqrt{\frac{a_k}{A_k}} \|H_k^{-1} \nabla f(\theta_k; z_{i_k})\|_2^2\right]}$$

$$\leq \quad \sqrt{t} \sqrt{\frac{C_2^2}{C_1^2}\left[w_1^2 \sum_{b=1}^{B} d_b \log\left(\frac{\sigma_b^2}{\epsilon^2} + 1\right) + d\sum_{k=1}^{t} \eta_k^2 \frac{A_k}{A_{k-1} + a_1}\right]}$$

$$\leq \quad \frac{C_2}{C_1} \sqrt{\left[w_1^2 \sum_{b=1}^{B} d_b \log\left(\frac{\sigma_b^2}{\epsilon^2} + 1\right) + d\omega \sum_{k=1}^{t} \eta_k^2\right] t}.$$

Then, we get

$$\mathbf{E}[\Delta_{t+1}] \quad \leq \quad \frac{2C_2}{nC_1}\sqrt{\left[w_1^2 \sum_{b=1}^{B} d_b \log\left(\frac{\sigma_b^2}{\epsilon^2} + 1\right) + d\omega \sum_{k=1}^{t} \eta_k^2\right] t}$$

$$+ \left(1 - \frac{1}{n}\right)\tilde{\gamma}\sum_{k=1}^{t} \eta_k \mathbf{E}\left[\max_b \left|\frac{1}{\sqrt{\hat{v}_{k,b}} + \epsilon} - \frac{1}{\sqrt{\hat{v}'_{k,b}} + \epsilon}\right|\right]$$

$$+ \left(1 - \frac{1}{n}\right) L \sum_{k=1}^{t} \eta_k \mathbf{E}\left[\frac{1}{\sqrt{\min_b \hat{v}_{k,b}} + \epsilon}\Delta_k\right].$$

As the bound holds for any $S, S', z$, it also holds for its supreme. □

