# OpenReview forum: "Blockwise Adaptivity:  Faster Training and Better Generalization in Deep Learning"
_ICLR.cc/2020/Conference — Reject_

### Official Review · AnonReviewer1 · 2019-10-21
**Official Blind Review #1**

**Rating:** 3

**Review:**

This paper proposes blockwise adaptivity.  We divide the parameters into blocks, for example in a linear threshold unit the bias term is in a bias term block while the input weights are in an input weight block.  We then average the square norm of the gradients over each block and use the same adaptation based on this average square norm for all parameters in the block.  theoretical and experimental results are given.

The idea of assigning different learning rates to different types of parameters is old.  Extending this idea to blockwise adaptation is natural and intuitively I would expect this to be an improvement on Adam.  However, I did not find this paper very compelling.  First, I believe that the theoretical results are a straightforward adaptation of know methods.  Second, and more significantly, comparing optimizers empirically is very tricky and I am not convinced that the experiments described here are convincing.  In particular the performance of optimizers is very sensitive to the tuning of hyper-parameters.  I would need to be convinced that the hyper-parameter tuning is sufficient.  Grid search is very inefficient compared to random or quasi-random methods.  Adam has four hyper-parameters and gird search over four parameters is very difficult.  For vision applications of Adam joint tuning of the learning rate and the epsilon parameter is critical --- these parameters are coupled.  It seems extremely likely to me that the move to blockwise adaptation has a profound effect on the optimal value of epsilon.  A thorough investigation of epsilon tuning is needed to demonstrate the value of blockwise adaptation in vision applications.

Postscript:

I still feel that the theoretical analysis provides little to no evidence of in-practice value of the method.  In general I find that theorems guaranteeing getting stuck on a flat plateau to be not very exciting.  What about the exploration goal of local search or mcmc?  In the absence of meaningful theory, the empirical results are what matter.

I have read the response and am not convinced by the comments on optimizing epsilon.  I still believe that empirical claims about optimizers require extraordinary do-diligence in hyper-parameter optimization of both the proposed method and the baselines.

**Experience Assessment:**

I have read many papers in this area.

**Review Assessment: Checking Correctness Of Derivations And Theory:**

I assessed the sensibility of the derivations and theory.

**Review Assessment: Checking Correctness Of Experiments:**

I assessed the sensibility of the experiments.

**Review Assessment: Thoroughness In Paper Reading:**

I made a quick assessment of this paper.

---

> ### Author Response · Authors · 2019-11-15
> **Response to Reviewer 1**
>
>
> "The idea of assigning different learning rates to different types of parameters is old."
>
> - Indeed, there are existing works on assigning different learning rates to different layers (Singh et al., 2015; You et al., 2017; Yu et al., 2017; Zhou et al., 2018). However, they are heuristic and do not have convergence analysis and uniform stability bound (there are no known uniform stability bound for coordinate-wise adaptive methods as well).
>
> "First, I believe that the theoretical results are a straightforward adaptation of know methods."
>
> - Compared to the existing heuristic layer-wise methods (Singh et al., 2015; You et al., 2017; Yu et al., 2017; Zhou et al., 2018), our work is the first one that shows that convergence analysis on optimizing non-convex objectives. By exploiting blockwise structure, our bound with $B=d$ can be $\mathcal{O}(\sqrt{d})$ better than the existing bound for coordinate-wise adaptive method (Zou et al., 2019). Besides, we give new results on what solutions the blockwise adaptive method finds for optimizing underdetermined least square and its nonlinear variant, and we are also the first one that gives uniform stability bound for adaptive gradient method.
>
> "I would need to be convinced that the hyper-parameter tuning is sufficient.  Grid search is very inefficient compared to random or quasi-random methods.  Adam has four hyper-parameters and gird search over four parameters is very difficult."
>
> - Grid search has been widely adopted in many existing works such as (Kingma \& Ba, 2015; Reddi et al., 2018; Bernstein et al., 2018; Zou et al., 2019; Luo et al., 2019). The details of search range and the optimal value are given in Appendix A. As can been seen, BAGM and Adam share the same optimal learning rate and momentum parameter. Thus, for most of cases, we can simply use the hyper-parameters tuned for Adam.
> - Tuning all the hyper-parameters are impractical and computationally expensive. Thus, given that the optimal $\epsilon$ has been given in (Zaheer et al., 2018) for Adam, we only tune other two most important hyper-parameters, namely, learning rate and first-moment momentum value. So the hyper-parameter tuning of Adam and BAGM do not come with higher cost than NAG.
>
> "It seems extremely likely to me that the move to blockwise adaptation has a profound effect on the optimal value of epsilon.  A thorough investigation of epsilon tuning is needed to demonstrate the value of blockwise adaptation in vision applications."
>
> - In our experiments, we use $\epsilon=10^{-3}$, which is suggested in (Zaheer et al., 2018) for coordinate-wise adaptive methods. In (Zaheer et al., 2018), they show that the generalization gets better when $\epsilon=10^{-3}$ compared to $\epsilon=10^{-8}$.
> - Our results show that, given that $\epsilon$ was tuned for coordinate-wise adaptive methods, our method can further improve the generalization over Adam.
> - Algorithmically, the proposed estimate of the second moment is a simple averaging, which does not change the scale of $\hat{v}_t$. This can be confirmed by the optimal learning rate (obtained by grid search) in Tables 4, 5 \& 7, which show that the optimal learning rates are the same for BAGM and Adam. If blockwise adaptation had an profound effect on the optimal value of $\epsilon$, we would have obtained different optimal learning rates. However, our results contradict this hypothesis.

---

### Official Review · AnonReviewer2 · 2019-10-24
**Official Blind Review #2**

**Rating:** 6

**Review:**

In this paper, the authors propose a generalization of AdaGrad, called BAG, that operates on blocks of parameters instead of each individual parameter. The authors also propose a momentum version of BAG called BAGM. Convergence rate results are proved for the algorithm, and some uniform stability results show situations where BAG would generalize better than previous adaptive gradient methods.

Overall I found the paper interesting, although I found the paper very dense and hard to read. The paper would be much easier to read if an effort was made to present the theoretical results in slightly simplified forms. In addition to this, I have a couple more concerns about the paper, which I list below:

1. I am a bit unsure about the what the uniform stability results add to the paper.
a) While Proposition 1 is interesting, is it relevant for the BAG algorithm since it considers a layer-wise training process.
b) Proposition 2 is also a bit confusion to me. How does generalization get worse as the number of blocks approaches the number of parameters (i.e., gets closer to AdaGrad)?
c) Proposition 4 looks interesting, but hard to interpret the way it is presented now.

2. Is the better generalization performance observed for BAGM (as well as Adam) simply due to the larger epsilon value used in the experiments? Would such an epsilon value simply turn off the adaptivity as the iterates gets closer to the minimizer (when the gradients are small relative to epsilon), and effectively do SGD? How does performance vary with epsilon?

==================================

Edit after rebuttal:
I thank the authors for their response. Some of the other authors have expressed a number of concerns about this paper's theoretical and empirical contributions, and I am not raising my score.


**Experience Assessment:**

I have published one or two papers in this area.

**Review Assessment: Checking Correctness Of Derivations And Theory:**

I assessed the sensibility of the derivations and theory.

**Review Assessment: Checking Correctness Of Experiments:**

I carefully checked the experiments.

**Review Assessment: Thoroughness In Paper Reading:**

I read the paper at least twice and used my best judgement in assessing the paper.

---

> ### Author Response · Authors · 2019-11-15
> **Response to Reviewer 2**
>
>
> "While Proposition 1 is interesting, is it relevant for the BAG algorithm since it considers a layer-wise training process."
>
> - We agree that it is not the same as that used in the proposed BAG(M) algorithm. What we want to demonstrate there is that blockwise adaptivity can be advantageous over coordinate-wise adaptivity in that setup. This motivates us to further exploit the optimization (Corollary 2) and generalization (Proposition 4) of blockwise adaptivity in end-to-end training.
>
> "Proposition 2 is also a bit confusion to me. How does generalization get worse as the number of blocks approaches the number of parameters (i.e., gets closer to AdaGrad)?"
>
> - Proposition 2 does not show that generalization gets worse as the number of block gets larger. What it shows is that BAG finds the minimum $\ell_2$-norm solution in each subspace induced by the group structure when each data sub-matrix has full row rank. Wilson et al., 2017 show that coordinate-wise adaptive methods converge to solutions with low $\ell_{\infty}$ norm, and such solutions can generalize arbitrarily poorly.
>
> "Proposition 4 looks interesting, but hard to interpret the way it is presented now."
>
> - We will make it more precise in the final version.
>
> "Is the better generalization performance observed for BAGM (as well as Adam) simply due to the larger epsilon value used in the experiments? Would such an epsilon value simply turn off the adaptivity as the iterates gets closer to the minimizer (when the gradients are small relative to epsilon), and effectively do SGD? How does performance vary with epsilon?"
>
> - In our experiments, we use $\epsilon=10^{-3}$, which is suggested in (Zaheer et al., 2018) for coordinate-wise adaptive methods. Zaheer et al., 2018 show how the generalization varies between $\epsilon=10^{-3}$ and $\epsilon=10^{-8}$.
> - larger epsilon value indeed improves the generalization for adaptive methods, which has been empirically verified in (Zaheer et al., 2018) and is theoretically demonstrated in Proposition 4. We agree that it is possible that epsilon can dominate the denominator when gradients are relatively small. However, our experiments also show that blockwise adaptivity can further improve upon Adam and achieve better generalization.

---

### Official Review · AnonReviewer3 · 2019-10-25
**Official Blind Review #3**

**Rating:** 1

**Review:**

The paper proposes adaptive gradient approaches where the step-size is not determined on the per-coordinate basis but rather for blocks of coordinates. Theoretical results are presented in terms of regret in online convex optimization, regarding convergence in non-convex optimization,  and with respect to uniform stability and generalization. These indicate that under certain conditions adaptivity at the block level could outperform coordinate-wise adaptivity. The approach is evaluated against alternatives on simulated and real-world problems.

The paper considers an important topic, which has been the object of many related studies. Though the proposed approach is interesting, there are several issues with the present manuscript that would warrant significant revision.
- Though the discussion in section 3 aims at motivating the use of block-wise adaptivity, it is quite confusing. Indeed the problem considered in that section is different from the setup eventually considered.  Moreover the paper claims that it is more general than the many previous work on layer-wise adaptation. However, the example considered here does consider layers.
- The paper organization could be improved significantly. BAG is presented followed by a regret analysis in convex optimization. Then BAGM is presented followed by the theory on non-convex optimization. It might be more effective to first present the algorithms BAG and BAGM and then have a theory section.  It would also be good to emphasize that the regret analysis is solely for convex optimization, which makes it much less relevant in the context of deep learning.
- The claimed superiority of block-wise adaptivity in the theory relies heavily on an assumption relying on tightness and closeness of  upper bounds on gradient magnitude /  gradient second moment in each block.  As this assumption is crucial to the results, its validity in practice should be more thoroughly investigated, beyond the small study relegated in the appendix  B. It would also be important to assess what happens when the assumption breaks down.
- The aforementioned issue is somewhat also related to the choice of design for the blocks. Some choices might help, some might hurt based on the unknown structure of the data. This is barely touched upon in section 5.1. as it is noted that the more sever the mismatch the worse the results.
- The empirical evaluation needs to be more comprehensive in terms of comparison methods. Among others it would be important to assess performance against previously proposed layer wise stepwise approaches mentioned in the introduction.
- These related approaches should also be discussed more deeply and contrasted against qualitatively.
- The empirical results are not substantially superior. It is also quite disappointing to see that the training, testing and generalization curves on Figure 2 are quite similar with NAG and Adam and do not exhibit less instability etc.

**Experience Assessment:**

I have published in this field for several years.

**Review Assessment: Checking Correctness Of Derivations And Theory:**

I assessed the sensibility of the derivations and theory.

**Review Assessment: Checking Correctness Of Experiments:**

I carefully checked the experiments.

**Review Assessment: Thoroughness In Paper Reading:**

I read the paper thoroughly.

---

> ### Author Response · Authors · 2019-11-15
> **Response to Reviewer 3**
>
>
> "Indeed the problem considered in that section is different from the setup eventually considered.  Moreover the paper claims that it is more general than the many previous work on layer-wise adaptation. However, the example considered here does consider layers. "
>
> - We agree that it is not the same as that used in the proposed algorithm. What we want to demonstrate there is that blockwise adaptivity can be advantageous over coordinate-wise adaptivity in that setup. This motivates us to further exploit the optimization (Corollary 2) and generalization (Proposition 4) of blockwise adaptivity in end-to-end training.
> - In this example, we want to explain why layer-wise adaption is better. For the proposed algorithms BAG(M), we consider more general cases such as B.2, B.3 and B.4, which have finer granularity.
> -As it may be confusing to the reader, we will remove this section in the final version.
>
> "It might be more effective to first present the algorithms BAG and BAGM and then have a theory section..."
>
> - Thank you for your suggestion. We will revise it in the final version.
>
> "Some choices might help, some might hurt based on the unknown structure of the data. This is barely touched upon in section 5.1. as it is noted that the more sever the mismatch the worse the results. "
>
> - The experiments on Section 5.1 serves to verify Corollary 1. Indeed, for linear model, it is difficult to construct proper blocks without prior knowledge. However, due to the natural layer-wise structure of deep learning models, we have some intuition on which parameters are likely to have similar gradients in expectation (this is because the theories suggest to group as many gradients with similar magnitudes in expectation as possible). For instance, for image classification, gradients in each parameter vector/matrix/tensor (i.e., B.1) are likely to have similar gradient upper bound in expectation. For language modeling task, as word frequency varies a lot, we expect that gradients in each word embedding (i.e., B.2) are likely to have similar gradients. Our experiments also validate such hypotheses as B.1 and B.2 achieve the best performance for image classification and language modeling, respectively.
>
> "The claimed superiority of block-wise adaptivity in the theory relies heavily on an assumption relying on tightness and closeness of  upper bounds on gradient magnitude/gradient second moment in each block..."
>
> - When we make Assumptions 2, 4$\&$8, constants $D_b$, $\sigma_b$ and $G_b$ are already the unknown "\textit{smallest}" (tight) constants. Given that constants $D_b$, $\sigma_b$ and $G_b$ are the smallest ones, the corresponding inequalities $D_b^2 \leq \sum_{i \in \mathcal{G}_b}D_i^2$ and $\sqrt{\frac{\max_b\max_{i\in\tilde{\mathcal{G}}_b} G_{i}^2 + \epsilon^2}{\max_b G_b^2 + \epsilon^2}} \geq 1$ hold.
> - We add extra experiment to estimate$\sqrt{\frac{\bar{v}_{T,d} + \epsilon^2}{\bar{v}_{T,
> \tilde{B}} + \epsilon^2}}$and $\min(r_1, r_2, r_3)$ for B.2, B.3 and B.4. The numbers are shown in Appendix B of the revision.
>
> "assess performance against previously proposed layer wise stepwise approaches mentioned in the introduction. "
>
> - As requested, we add comparison to LARS (You et al., 2017), the results on CIFAR-10 and Wikitext-2 are: LARS:  $6.60 \pm 0.17$ on ResNet56, $6.15 \pm 0.21$ on ResNet110, and $65.38 \pm  0.08$ on AWD-LSTM. LARS improves over NAG and Adam. However, it is still worse than BAGM-B.1 and BAGM-B.4 on CIFAR-10, and is worse than BAGM-B.2 on training AWD-LSTM.  For the experiment on ImageNet, we have tried a number of hyper-parameters, but none of them allow LARS to converge.
>
> "These related approaches should also be discussed more deeply and contrasted against qualitatively."
>
> - These approaches are heuristic and do not have convergence analysis. Algorithmically, these layer-wise methods consider incorporating gradient normalization into the learning rate, while our method averages the estimate of the second moments of each block, which is more related to coordinate-wise adaptive methods.
>
> "The empirical results are not substantially superior... "
>
> - Improvements on these competitive datasets are usually not very substantial. Here, we compare our improvements with those of other papers (that are not addressing the same problem as ours) on the same dataset.  For example, on ImageNet, BAGM-B.1 has 0.15% better accuracy than NAG; while in Table 8, "Wide residual networks",  changing from ResNet152 to WRN-50-2-bottleneck achieves a similar 0.26% improvement. On language modeling, BAGM-B.2 reduces perplexity by 0.46 compared to NAG; while in Table 2, (Merity et al., 2017), changing from a 2-layer skip connection LSTM to AWD-LSTM only improves perplexity by 0.1. On the other hand, the test error of BAGM-B.1 on ResNet56 is lower than those of NAG and Adam on ResNet110.
> - From Tables 1,2&3, it can be seen that standard deviation of all methods are already very small relative to the error rate (%).

---

### Decision · Program_Chairs · 2019-12-19

**Decision:**

Reject

**Comment:**

The authors propose an adaptive block-wise coordinate descent method and claim faster convergence and lower generalization error. While the reviewers agreed that this method may work well in practice, they had several concerns about the relevance of the theory and strength of the empirical results. After considering the author responses, the reviewers have agreed that this paper is not yet ready for publication.